# DS-STAR: Data Science Agent for Solving Diverse Tasks across Heterogeneous Formats and Open-Ended Queries

**Jaehyun Nam**                                                                          *jaehyunnam@google.com*
*Google Cloud AI Research*

**Jinsung Yoon**                                                                         *jinsungyoon@google.com*
*Google Cloud AI Research*

**Jiefeng Chen**                                                                           *jiefengc@google.com*
*Google Cloud AI Research*

**Rajarishi Sinha**                                                                          *sinharaj@google.com*
*Google Cloud AI Research*

**Jinwoo Shin**                                                                         *jinwoos@kaist.ac.kr*
*KAIST AI*

**Tomas Pfister**                                                                             *tpfister@google.com*
*Google Cloud AI Research*

**Reviewed on OpenReview:** *https://openreview.net/forum?id=Yz3ZPLzYaU*

## Abstract

While large language models (LLMs) have shown promise in automating data science, existing agents often struggle with the complexity of real-world workflows that require exploring multiple sources and synthesizing open-ended insights. In this paper, we introduce DS-STAR, a specialized agent to bridge this gap. Unlike prior approaches, DS-STAR is designed to (1) seamlessly process and integrate data across diverse, heterogeneous formats, and (2) move beyond simple QA to generate comprehensive research reports for open-ended queries. Extensive evaluation shows that DS-STAR achieves state-of-the-art performance on four benchmarks: DABStep, DABStep-Research, KramaBench, and DA-Code. Most notably, it significantly outperforms existing baseline models especially in hard-level QA tasks requiring multi-file processing, and generates high-quality data science reports that are preferred over the best baseline model in over 88% of cases. Code is available at https://github.com/google-research/ds-star.

## 1 Introduction

Data science transforms raw data into actionable insights and is critical to solving real-world problems (De Bie et al., 2022; Hassan et al., 2023); indeed, businesses rely heavily on these insights to make strategic decisions (Sun et al., 2018; Sarker, 2021). However, the data science workflow is inherently complex, requiring deep expertise across diverse fields such as computer science and statistics (Zhang et al., 2023; Hong et al., 2024). This workflow entails a series of labor-intensive tasks (Zhang et al., 2024; Martin Iglesias, 2025), ranging from understanding distributed document to executing intricate data processing and statistical analyses. To simplify this demanding process, recent research has explored deploying large language models (LLMs) as autonomous data science agents (Hong et al., 2024). These agents aim to translate natural language queries directly into functional code that generate answers (Ouyang et al., 2025; Yang et al., 2025).

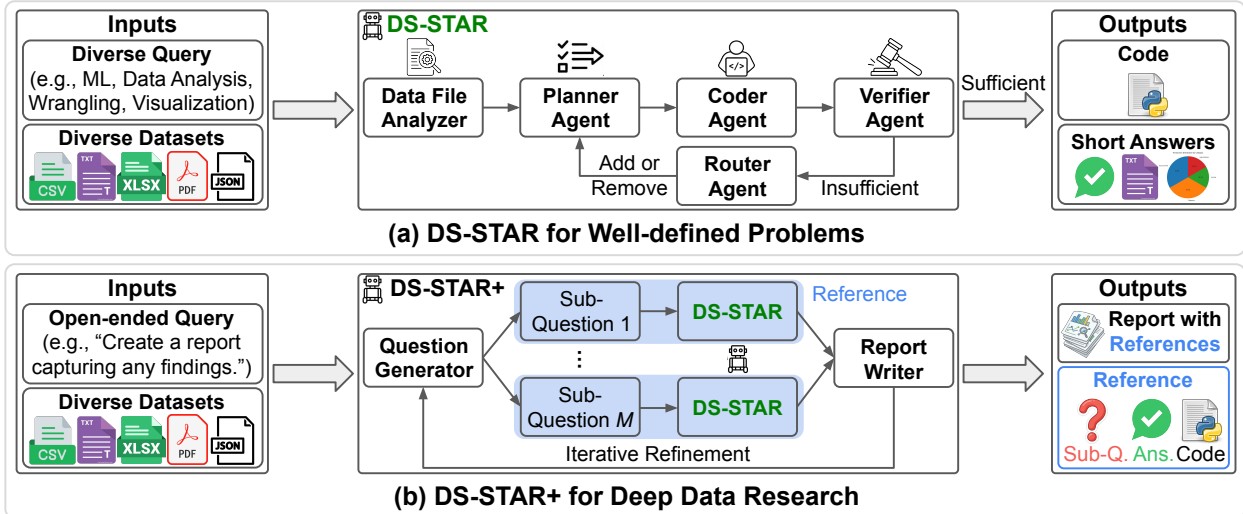

Figure 1: **Overview.** DS-STAR is capable to process data files in various formats (*e.g.*, csv, txt, xlsx, md). (a) DS-STAR is designed to accomplish user queries, such as data analysis for extracting useful insights or predictive tasks using machine learning, by writing code scripts (*e.g.*, Python) and answering based on their execution output. The output may include a trained model for prediction tasks, processed databases, text-formatted answers, visual charts, and more. (b) Building upon this core capability, we develop DS-STAR+ for deep data research, which addresses open-ended queries. DS-STAR+ generates a comprehensive data science report where each statement is grounded on the generated sub-questions and their corresponding code solutions obtained from DS-STAR. Here, DS-STAR+ utilizes the data file analyzer of DS-STAR to obtain the data file description and uses this to generate sub-questions for the open-ended query.

Despite these advancements, existing agents face two major limitations that hinder their application in real-world scenarios. First, they predominantly focus on well-structured data, such as relational databases composed of CSV files (Li et al., 2024; Pourreza et al., 2024), neglecting the wealth of information available in the heterogeneous data formats encountered in real-world scenarios (*e.g.*, JSON, unstructured text, and markdown). Second, current agents are primarily designed for specific, well-defined queries, such as distinct data analysis, machine learning, or visualization tasks, where an exact answer exists (Hong et al., 2024; Huang et al., 2024). They often struggle with open-ended exploration (Zhang et al., 2025), where a user seeks a broad investigation of the data and a comprehensive report of finding, rather than a single numerical result.

In this paper, we propose *DS-STAR*, a novel agentic framework designed to tackle any data-related task, addressing both heterogeneity and open-endedness. DS-STAR distinguishes itself through two key capabilities: it operates on data with diverse, heterogeneous formats (*e.g.*, csv, txt, xlsx, md), and it extends beyond simple query answering to generate comprehensive reports for open-ended exploratory questions.

To achieve this, we first build a module for answering well-defined questions (see Figure 1(a)). This module operates in two stages. First, to ensure adaptability across diverse data types, DS-STAR automatically analyzes all files in a directory, generating a textual summary of their structure and content. This summary then serves as a crucial context of DS-STAR's approach to the task. Second, DS-STAR enters a core loop of planning, implementation, and verification. It begins by formulating a high-level plan, implementing it as a code script, and then utilizes an LLM-based judge (Zheng et al., 2023; Gu et al., 2024) to explicitly evaluate whether the plan is sufficient to solve the problem. Crucially, rather than generating a complete plan at once, the agent operates sequentially, reviewing intermediate results before proceeding. If verification fails, the agent refines its plan, repeating the cycle until a satisfactory solution is verified.

Building upon this core capability, we expand DS-STAR to address open-ended objectives, *i.e.*, DS-STAR+. Given an open-ended query (see Figure 1(b)), DS-STAR+ decomposes the query into a series of well-defined questions. The agent then solves these sub-questions using DS-STAR. Finally, grounding its reasoning in these question-answer pairs, DS-STAR+ generates a comprehensive data science report.

We verify DS-STAR's effectiveness through rigorous evaluations on various data science benchmarks. We utilize DABStep (Martin Iglesias, 2025), KramaBench (Lai et al., 2025), and DA-Code (Huang et al., 2024) to assess performance on complex data analysis tasks (*e.g.*, data wrangling, machine learning, visualization), and DABStep-Research (Zhang et al., 2025) to evaluate open-ended report generation. All benchmarks involve multiple data sources and formats. Experimental results demonstrate that DS-STAR significantly outperforms state-of-the-art methods across all scenarios. Specifically, compared to the best alternative, DS-STAR improves accuracy from 41.0% to 45.2% on the DABStep benchmark, from 39.8% to 44.7% on KramaBench, and from 37.0% to 38.5% on DA-Code. Furthermore, reports generated by DS-STAR+ are preferred over those from DeepAnalyze (Zhang et al., 2025) in over 88% of cases according to LLM judge.

## 2 Related work

**LLM agents.** Recent advancements in LLMs have spurred significant research into autonomous agents designed to tackle complex and long-horizon tasks. A key strategy employed by these multi-agent systems is the autonomous decomposition of a main objective into a series of smaller, manageable sub-tasks, which can be delegated to sub-agents. General-purpose agents, such as ReAct (Yao et al., 2023), HuggingGPT (Shen et al., 2023), and OpenHands (Wang et al., 2024), utilize external tools to reason, plan, and act across a wide range of problems. Building on these foundational capabilities, research has increasingly focused on specialized domains such as Voyager (Wang et al., 2023) for navigating the Minecraft environment and AlphaCode (Li et al., 2022) for performing code generation. Furthermore, DS-Agent (Guo et al., 2024), AIDE (Jiang et al., 2025), and MLE-STAR (Nam et al., 2025) are tailored specifically for machine learning engineering. In a similar vein, our work is a specialized LLM agent for data science tasks.

**Data science agents.** Recent research has increasingly focused on developing data science agents that leverage the coding and reasoning capabilities of LLMs (Jimenez et al., 2024; Jiang et al., 2025). Initial efforts utilized general-purpose frameworks such as ReAct (Yao et al., 2023) and AutoGen (Wu et al., 2023). Building on these pioneering approaches, specialized agents like DA-Agent (Huang et al., 2024) have emerged, designed to autonomously generate code-based solutions for data analysis and visualization (Hu et al., 2024; Jing et al., 2025). A prominent example in this domain is Data Interpreter (Hong et al., 2024), which employs a graph-based method to decompose primary tasks into a series of manageable sub-tasks; the resulting task graph is dynamically refined based on execution feedback. However, a critical limitation of this approach is its reliance on successful code execution as the sole proxy for correctness. This often leads to sub-optimal plans, as execution success does not guarantee logical accuracy or alignment with user intent. To mitigate this, DS-STAR introduces a verification mechanism that employs an LLM-based judge to assess solution quality beyond mere executability. More recently, DeepAnalyze (Zhang et al., 2025) proposed a training paradigm, fine-tuning the DeepSeek-R1-0528-Qwen3-8B model with their custom DataScience-Instruct-500K dataset. While effective, fine-tuning-based approaches face significant scalability challenges: they require substantial retraining costs to adapt to new domains and cannot immediately benefit from the rapid advancements of foundation models. Furthermore, they are often precluded from leveraging the superior reasoning capabilities of proprietary LLMs (*e.g.*, Gemini), which consistently outperform open-source models on complex reasoning benchmarks. In this paper, we propose a multi-agent system built upon proprietary commercial models to fully leverage their strong reasoning and coding capabilities without the constraints of fine-tuning.

**Text-to-SQL.** The task of Text-to-SQL aims to translate natural language questions into executable SQL queries (Li & Jagadish, 2014; Li et al., 2023). Early approaches relied on sequence-to-sequence architectures that jointly encoded user queries and database schemas (Cao et al., 2021; Choi et al., 2021). With the advent of LLMs, research shifted toward prompt engineering (Pourreza & Rafiei, 2023), followed by the development of sophisticated multi-step pipelines incorporating schema linking, self-correction, and self-consistency to enhance accuracy (Maamari et al., 2024; Pourreza et al., 2024; Talaei et al., 2024). While our proposed method, DS-STAR, retains the capability to generate SQL, particularly advantageous for querying large-scale structured relational databases, it fundamentally broadens the scope by adopting Python as the target language. By framing the data science task as a Text-to-Python task and introducing a novel data file summarization mechanism, DS-STAR significantly expands applicability beyond traditional Text-to-SQL, effectively handling diverse data formats including JSON, Markdown, and unstructured text.

**LLM as a verifier.** Our work builds upon the literature on iterative refinement and self-correction. Reflexion (Shinn et al., 2023) introduced a framework for agents to learn from linguistic feedback through trial and error, while Self-Refine (Madaan et al., 2023) demonstrated that LLMs can improve their own outputs by generating and acting upon multi-step feedback. More recently, TextGrad (Yuksekgonul et al., 2024) proposed treating textual feedback as gradients to optimize system components. Our verifier agent differentiates itself by specifically evaluating the sufficiency of a dynamically evolving analysis plan grounded in intermediate code execution results across heterogeneous data formats, enabling the agent to navigate complex, open-ended data science tasks where ground-truth labels are absent.

## 3 DS-STAR

In this section, we introduce DS-STAR, a novel data science agent that is designed to automate end-to-end data science workflows. Section 3.1 first introduces a module for well-defined queries that yield concise answers. Subsequently, in Section 3.2, we propose DS-STAR+, which extends this core functionality of DS-STAR and generates comprehensive research reports that address open-ended queries. Implementation details on algorithms are in Appendix C, and used prompts are in Appendix K and Appendix L.

### 3.1 DS-STAR for well-defined queries

**Problem setup.** Our goal is to automatically generate a code solution $s$ (*e.g.*, a Python or SQL script) that answers query $q$ using a given data files $\mathcal{D}$. We formulate this as a search problem over the space of all possible scripts, denoted by $\mathcal{S}$. The quality of any script $s \in \mathcal{S}$ is evaluated by a scoring function $h(s)$, which measures the correctness of its output, $s(\mathcal{D})$, against a ground-truth answer (*e.g.*, accuracy) where $s(\mathcal{D})$ represents the output generated by executing $s$, which loads the data $\mathcal{D}$ to produce an answer. Therefore, the objective is to identify the $s^*$ that is:

$$s^* = \arg\max_{s \in \mathcal{S}} h(s(\mathcal{D})). \tag{1}$$

In this paper, we propose a multi-agent framework, $\mathcal{A}$, designed to process the query $q$ and data files $\mathcal{D}$, which may be numerous and in heterogeneous formats. This framework is composed of $n$ specialized LLM agents, $\{\mathcal{A}_i\}_{i=1}^n$, where each agent possesses a distinct functionality, as detailed in the subsequent sections.

**Analyzing data files.** To effectively interact with the given data files, DS-STAR first requires a comprehensive understanding of its contents and structure. We achieve this by generating a concise, analytical description for each data file. This initial analysis informs all subsequent actions taken by the agent. Conventional frameworks for data science often rely on displaying a few sample rows from structured files, such as CSVs. However, this approach is fundamentally limited to structured data and fails for unstructured formats where the concept of a 'row' is ill-defined. To overcome this limitation, we introduce a more general mechanism as follows. For each data file $\mathcal{D}_i \in \mathcal{D}$ we employ an analyzer agent, $\mathcal{A}_{\texttt{analyzer}}$, to generate a Python script, $s_{\texttt{desc}}^i$. This script is designed to correctly parse and load the data file $\mathcal{D}_i$ and then extract its essential properties. For structured data, this might include column names and data types; for unstructured data, it could be file metadata, text summaries. The resulting analytical description, $d_i$, is captured directly from the script's execution output (see Appendix G and H). This process, which can be parallelized, is denoted as:

$$d_i = \texttt{exec}(s_{\texttt{desc}}^i), \ s_{\texttt{desc}}^i = \mathcal{A}_{\texttt{analyzer}}(\mathcal{D}_i). \tag{2}$$

**Initialization for planning.** After generating analytic descriptions of the $N$ data files, DS-STAR begins the solution generation process. First, a planner agent, $\mathcal{A}_{\texttt{planner}}$, generates an initial high-level executable step $p_0$ (*e.g.*, loading a data file) using the query $q$ and obtained data descriptions:

$$p_0 = \mathcal{A}_{\texttt{planner}}(q, \{d_i\}_{i=1}^N). \tag{3}$$

**Plan implement and execution.** Then, DS-STAR enters an execution and verification cycle. First, $\mathcal{A}_{\texttt{coder}}$ implements $p$ into code $s$. The execution of this code yields a new observation $r = \texttt{exec}(s)$. With this $r$, our verier agent $\mathcal{A}_{\texttt{verifier}}$ is invoked to assess if the current plan is sufficient.

**Plan verification.** The main challenge in data science tasks is guiding the refinement of a solution, as determining its correctness is often non-trivial, since there are no ground-truth label. To address this, we use an LLM as a judge to assess whether the current plan is sufficient for the user's query. Our approach introduces a verifier agent $\mathcal{A}_{\tt verifier}$. At any given round $k$ in the problem-solving process, this agent evaluates the state of the solution, *i.e.*, whether the plan is sufficient to solve the problem. The evaluation is based on the cumulative plan $p = \{p_0, \cdots, p_k\}$, the user's query $q$, the current solution code $s_k$, which is an implementation of the cumulative plan, and its execution result $r_k$. This is denoted as follows:

$$v = \mathcal{A}_{\tt verifier}(p, q, s_k, r_k). \tag{4}$$

Here, the output $v$ is a binary variable: `sufficient` or `insufficient`. While our method focuses primarily on comparing the plan and query, we found that providing $s_k$ and its execution output $r_k$ as additional conditions to $\mathcal{A}_{\tt verifier}$ improves overall performance (see Table 8; even without providing these conditions, the performance degradation of DS-STAR is minimal). We hypothesize that performance is consistently improved because providing these conditions allows $\mathcal{A}_{\tt verifier}$ to evaluate whether $s_k$ is properly implemented and whether $r_k$ contains the information necessary to fully address the query.

**Plan refinement.** If the verifier agent $\mathcal{A}_{\tt verifier}$ determines that the current plan is insufficient to solve the user's query $q$, DS-STAR must decide how to proceed. Such insufficiency could arise because the plan is merely incomplete and requires additional steps, or because it contains erroneous steps that invalidate the approach. To resolve this, DS-STAR employs a router agent, $\mathcal{A}_{\tt router}$, which decides whether to append a new step or to correct an existing one. The router's decision $w$ is generated as follows, where $p = \{p_0, \cdots, p_k\}$ is the current cumulative plan: $w = \mathcal{A}_{\tt router}(p, q, r_k, \{d_i\}_{i=1}^N)$.

The output $w$ is either the token `Add Step` or an index $l \in \{1, \cdots, k\}$. If $w$ is `Add Step`, $\mathcal{A}_{\tt router}$ has determined the plan is correct but incomplete. In this case, we retain the plan $p$ and proceed to generate the next step. On the other hand, if $w = l$, $\mathcal{A}_{\tt router}$ has identified $p_l$ as erroneous. In this case, we backtrack by truncating the plan to $p \leftarrow \{p_0, \cdots, p_{l-1}\}$. Here, we deliberately choose to truncate and regenerate through the LLM's random sampling, rather than directly correcting $p_l$, since our empirical finding has shown that revising a specific incorrect step often leads to an overly complex replacement, therefore frequently flagged again by $\mathcal{A}_{\tt router}$ in a next iteration. Following the decision from the $\mathcal{A}_{\tt router}$, our agent proceeds with an updated plan $p = \{p_0, \cdots, p_{k'}\}$, where $k' = k$ or $k' = l - 1$. Then DS-STAR generates a subsequent step:

$$p_{k'+1} = \mathcal{A}_{\tt planner}(p, q, r_k, \{d_i\}_{i=1}^N). \tag{5}$$

Notably, the planner agent $\mathcal{A}_{\tt planner}$ is conditioned on the last execution result, $r_k$, enabling it to generate a step that attempts to resolve the previously identified insufficiency. Once the new step $p_{k'+1}$ is defined, the plan is updated to: $p \leftarrow \{p_0, \cdots, p_{k'}, p_{k'+1}\}$. This entire iteration—planning, coding, executing, verifying, and routing—is repeated until $\mathcal{A}_{\tt verifier}$ returns a `sufficient` or a maximum number of iterations is reached.

**Additional module: Debugging agent.** When a Python script $s$ fails during execution, it generates an error traceback $\mathcal{T}_{\tt bug}$. To automatically debug the script, DS-STAR employs a debugging agent, $\mathcal{T}_{\tt debugger}$. First, when generating $\{d_i\}_{i=1}^N$ using $s_{\tt desc}$ obtained from $\mathcal{A}_{\tt analyzer}$, $\mathcal{A}_{\tt debugger}$ iteratively update the script using only the traceback: $s_{\tt desc} \leftarrow \mathcal{A}_{\tt debugger}(s_{\tt desc}, \mathcal{T}_{\tt bug})$.

Secondly, once DS-STAR obtains $\{d_i\}_{i=1}^N$, $\mathcal{A}_{\tt debugger}$ utilizes such information when generating a solution script $s$. Our key insight is that tracebacks alone are often insufficient for resolving errors in data-centric scripts, while $\{d_i\}_{i=1}^N$ might include critical metadata such as column headers in a CSV file, sheet names in an Excel workbook, or database schema information. Therefore, $\mathcal{A}_{\tt debugger}$ generates a corrected script, $s$ , by conditioning on the original script s, the error traceback $\mathcal{T}_{\tt bug}$ , and this rich data context $\{d_i\}_{i=1}^N$:

$$s \leftarrow \mathcal{A}_{\tt debugger}(s, \mathcal{T}_{\tt bug}, \{d_i\}_{i=1}^N). \tag{6}$$

**Additional module: Retriever.** A potential scalability challenge arises when the number of data files $N$ is large (*i.e.*, $N > 100$). In such cases, all descriptions $\{d_i\}_{i=1}^N$ cannot be prompted within the predefined context length of LLMs. To address this, we employ a retrieval mechanism that leverages a pre-trained embedding model (Nie et al., 2024). Specifically, we identify the top-$K$ most relevant data files, which will be provided as context to the LLM, by computing the cosine similarity between the embedding of the user's query and the embedding of each description.

### 3.2 DS-STAR for deep data research

In real-world scenarios, users frequently pose open-ended queries—such as "Analyze the data to create a report capturing any relevant findings"—which require data science agents to synthesize a comprehensive report rather than simply generating code. In this section, we extend the core module of DS-STAR to address such deep research tasks (see Figure 1(b) for an overview).

**Gathering evidence via sub-question generation.** To construct a detailed report, we first decompose the high-level query into specific analytical sub-questions. Our key insight is that while open-ended queries are difficult to address directly from raw data, they become manageable when broken down into concrete analytical tasks. Formally, we provide the data descriptions $\{d_i\}_{i=1}^N$ and the open-ended query $q$ to the sub-question generator $\mathcal{A}_{\texttt{generator}}$. This module generates a set of analysis questions $\{f_i^0\}_{i=1}^{M_0}$ designed to elicit the information necessary for a high-quality report, ensuring each question is answerable given the available data:

$$\{f_i^0\}_{i=1}^{M_0} = \mathcal{A}_{\texttt{generator}}(q, \{d_i\}_{i=1}^N). \tag{7}$$

Here, the number of questions $M_0$ is determined dynamically by the model. Following generation, we utilize the core module of DS-STAR (see Section 3.1) to obtain the answer $a_i$ for each generated question:

$$a = \texttt{DS-STAR}(f, \mathcal{D}). \tag{8}$$

**Compiling the research report.** We utilize the resulting pairs of sub-questions and answers, $\{f_i^0, a_i^0\}_{i=1}^{M_0}$, as grounded evidence to guide the report generator, $\mathcal{A}_{\texttt{writer}}$. To ensure hallucination-free generation, we explicitly instruct $\mathcal{A}_{\texttt{writer}}$ to cite specific sub-questions as evidence for its claims. The report $R$ is generated:

$$R = \mathcal{A}_{\texttt{writer}}(q, \{f_i^0, a_i^0\}_{i=1}^{M_0}). \tag{9}$$

**Iterative refinement.** Beyond single-pass generation, we also introduce a refinement strategy to iteratively enhance report quality. At round $K$, we re-engage $\mathcal{A}_{\texttt{generator}}$, providing the current report $R$ as context. The agent then identifies gaps in the current draft and generates supplementary sub-questions $\{f_i^K\}_{i=1}^{M_K}$:

$$\{f_i^K\}_{i=1}^{M_K} = \mathcal{A}_{\texttt{generator}}(q, \{d_i\}_{i=1}^N, R). \tag{10}$$

After obtaining the new answers $\{a_i^K\}_{i=1}^{M_K}$ via DS-STAR, we prompt $\mathcal{A}_{\texttt{writer}}$ to update the original report by integrating this additional evidence:

$$R \leftarrow \mathcal{A}_{\texttt{writer}}(q, R, \{f_i^K, a_i^K\}_{i=1}^{M_K}). \tag{11}$$

## 4 Experiments on well-defined queries

In this section, we evaluate DS-STAR's capability on well-defined data science tasks where ground truth answer exist. In Section 4.1, we utilize three challenging data science benchmarks, *i.e.*, DABStep (Martin Iglesias, 2025), KramaBench (Lai et al., 2025), and DA-Code (Huang et al., 2024). In Section 4.2, we perform a detailed ablation study to understand the source of gains.

**Common setup.** DS-STAR runs for a maximum of 20 rounds per task with Gemini-2.5-Pro as the base LLM. Additionally, we employ a finalizer agent, which takes formatting guidelines (*e.g.*, rounding to two decimal places) and generates the final code solution (see Appendix K).

### 4.1 Main results of DS-STAR

**DABStep.** We evaluate DS-STAR on the DABStep benchmark (Martin Iglesias, 2025), which is designed to mirror real-world data analysis tasks requiring the processing of seven diverse data files, including formats like JSON, Markdown, and CSV. The benchmark features two difficulty levels; its hard-level tasks are particularly demanding, necessitating the analysis of multiple data files. Statistics and the full execution logs of DS-STAR are provided in Appendix D and J, respectively.

Table 1: **Main results from DABStep.** We report accuracy (%) on easy-level and hard-level tasks. All results are taken from the DABStep leaderboard (Martin Iglesias, 2025) and DeepAnalyze paper (Zhang et al., 2025), except for the model marked with †. The highest scores are shown in **bold**. For DS-STAR with Gemini-2.5-Pro, we report the mean and standard deviation for three random seeds.

| Framework | Model | Easy | Hard |
|---|---|---|---|
| Model-only | Gemini-2.5-Pro | 66.67 | 12.70 |
| | o4-mini | 76.39 | 14.55 |
| | DeepAnalyze-8B | 70.83 | 32.80 |
| ReAct | Claude-4-Sonnet | 81.94 | 19.84 |
| | Gemini-2.5-Pro† | 69.44 | 10.05 |
| AutoGen | Gemini-2.5-Pro† | 59.72 | 10.32 |
| Data Interpreter | Gemini-2.5-Pro† | 72.22 | 3.44 |
| DA-Agent | Gemini-2.5-Pro† | 68.06 | 22.49 |
| Open Data Scientist | DeepSeek-V3 | 84.72 | 16.40 |
| I2I-Agent | Claude-3.5-Sonnet | 80.56 | 28.04 |
| Amity DA Agent | Gemini-2.5-Pro | 80.56 | 41.01 |
| Claude Code | Claude-4.5-Sonnet† | 84.72 | 41.80 |
| Gemini CLI | Gemini-2.5-Pro† | 76.39 | 27.78 |
| | Gemini-3-Flash† | **88.89** | 42.33 |
| **DS-STAR (Ours)** | Gemini-2.5-Flash† | 80.56 | 29.37 |
| | Gemini-2.5-Pro† | $\textbf{88.89}_{\pm 1.39}$ | $45.50_{\pm 1.22}$ |
| | Gemini-3.1-Pro-Preview† | 87.50 | **54.23** |

As shown in Table 1, demonstrate that DS-STAR significantly outperforms all baselines. For instance, integrating DS-STAR with Gemini-2.5-Pro boosts the hard-level accuracy from 12.70% to 45.24%, an absolute improvement of over 32 percentage points. Notably, the DS-STAR using Gemini-2.5-Pro substantially surpasses other commercial agents like Open Data Scientist, Mphasis-I2I-Agents, and Amity DA Agent across both easy and hard difficulty levels. In addition, while Claude Code and Gemini CLI demonstrate impressive performance, it still falls short of our DS-STAR. Specifically, DS-STAR achieves higher accuracy despite using an older generation foundation model (Gemini-2.5-Pro) compared to Claude Code (Claude-4.5-Sonnet); note that ReAct framework with Gemini-2.5-Pro shows lower performance compared to ReAct with Claude-4-Sonnet. Moreover, with the same base model (Gemini-2.5-Pro), we found that Gemini CLI is much worse than DS-STAR. This experiment demonstrates that while general-purpose coding agents are powerful, specialized data science agent architectures provide distinct, practical advantages for data science tasks, thereby validating our core contribution.

**KramaBench.** We additionally validate DS-STAR on Kramabench (Lai et al., 2025), a benchmark that requires data discovery, *i.e.*, selecting the correct data files from a vast data lake to address a user's query. For example, the Astronomy domain includes over 1,500 files, while only a small subset is relevant to any single query (see Appendix D for the detailed statistics). To navigate this, DS-STAR integrates a retrieval module that identifies the top 100 candidate files (if the total data is less than 100, we fully utilized all the data for the task) based on embedding similarity between the user's query and data descriptions, computed with Gemini-Embedding-001 (Lee et al., 2025).

As shown in Table 2, DS-STAR demonstrates a substantial performance gain. Specifically, DS-STAR achieves an accuracy of 44.69%, significantly outperforming the 39.79% score of the state-of-the-art DA-Agent. Moreover, to isolate the impact of data retrieval, we also evaluated performance in an oracle setting, assuming all relevant data for a task is already provided. Under these ideal conditions, DS-STAR's accuracy increases by 8 percentage points. In contrast, other agentic frameworks like ReAct show only a marginal performance gain. This large gap highlights that while our current retrieval method is effective, advanced data discovery is a promising direction for unlocking the full potential of DS-STAR.

Table 2: **Main results from KramaBench.** All results are taken from the original paper (Lai et al., 2025), except the results with Gemini-2.5-Pro. The highest scores are shown in **bold**.

| Framework | Model | Domains | | | | | | |
| | | Archaeology | Astronomy | Biomedical | Environment | Legal | Wildfire | **Total** |
|---|---|---|---|---|---|---|---|---|
| *Original experimental setting for which the relevant data must be retrieved.* | | | | | | | | |
| Model-only | o3 | 25.00 | 1.73 | 3.50 | 1.35 | 3.35 | 24.87 | 9.64 |
| | GPT-4o | 0.00 | 1.41 | 1.98 | 0.45 | 1.46 | 1.45 | 1.62 |
| | Claude-3.5-Sonnet | 16.67 | 1.62 | 2.87 | 1.17 | 7.33 | 13.63 | 7.45 |
| DS-GURU | o3 | 25.00 | 3.53 | 8.95 | 19.60 | 13.89 | 50.73 | 22.08 |
| | GPT-4o | 16.67 | 2.76 | 8.97 | 2.60 | 2.80 | 17.18 | 8.28 |
| | Claude-3.5-Sonnet | 16.67 | 1.52 | 1.96 | 11.21 | 7.01 | 39.16 | 14.35 |
| ReAct | Gemini-2.5-Pro | 16.67 | 4.77 | 3.69 | 26.17 | 41.95 | 51.40 | 30.31 |
| AutoGen | Gemini-2.5-Pro | 16.67 | 4.39 | 7.25 | 19.38 | 26.38 | 41.76 | 22.83 |
| Data Interpreter | Gemini-2.5-Pro | **41.67** | 12.72 | 28.05 | 9.87 | 30.04 | 59.67 | 31.32 |
| DA-Agent | Gemini-2.5-Pro | **41.67** | **15.52** | 12.59 | 42.64 | 39.73 | 61.61 | 39.79 |
| **DS-STAR (Ours)** | Gemini-2.5-Pro | 25.00 | 12.09 | **43.74** | **46.75** | **49.64** | **65.94** | **44.69** |
| *Oracle experimental setting with relevant data already provided.* | | | | | | | | |
| ReAct | Gemini-2.5-Pro | 25.00 | 6.36 | 3.69 | 30.70 | 46.93 | 51.69 | 33.82 |
| AutoGen | Gemini-2.5-Pro | 25.00 | 5.29 | 26.03 | 23.46 | 39.28 | 49.41 | 31.77 |
| Data Interpreger | Gemini-2.5-Pro | **41.67** | 13.29 | 28.39 | 13.09 | 32.97 | 63.13 | 33.57 |
| DA-Agent | Gemini-2.5-Pro | **41.67** | 15.77 | 44.26 | 44.55 | 56.42 | 65.92 | 48.61 |
| **DS-STAR (Ours)** | Gemini-2.5-Pro | 25.00 | **19.08** | **55.24** | **58.80** | **59.66** | **70.18** | **52.55** |

Table 3: **Main results from DA-Code.** All results are taken from the original paper (Huang et al., 2024), except the results with Gemini-2.5-Pro. The highest scores are shown in **bold**.

| Framework | Model | Score | | | | | | |
| | | Data Wrangling | ML | EDA | Easy | Medium | Hard | **Total** |
|---|---|---|---|---|---|---|---|---|
| DA-Agent | Mixtral-8x22B | 14.8 | 31.6 | 10.2 | 17.6 | 16.8 | 8.6 | 15.4 |
| | DeepSeek-Coder-V2.5 | 25.1 | 34.1 | 14.7 | 32.8 | 18.7 | 14.1 | 20.7 |
| | Qwen-2.5-72B | 24.9 | 41.8 | 15.4 | 31.9 | 19.4 | 22.3 | 22.6 |
| | Claude-3-Opus | 29.3 | 46.8 | 20.7 | 44.7 | 23.8 | 19.0 | 27.6 |
| | GPT-4 | 30.4 | 48.4 | 24.6 | 45.4 | 27.8 | 23.4 | 30.5 |
| | Gemini-2.5-Pro | **34.8** | 57.2 | 31.1 | **50.0** | 34.2 | 32.0 | 37.0 |
| ReAct | Gemini-2.5-Pro | 14.7 | 31.2 | 22.2 | 32.0 | 17.9 | 25.9 | 22.5 |
| AutoGen | Gemini-2.5-Pro | 25.6 | 51.6 | 25.6 | 38.7 | 28.5 | 29.6 | 30.8 |
| **DS-STAR (Ours)** | Gemini-2.5-Pro | 30.4 | **57.3** | **34.8** | 48.9 | **35.2** | **37.1** | **38.5** |

**DA-Code.** To evaluate the generalization capabilities of DS-STAR, we leverage the DA-Code dataset (Huang et al., 2024), which encompasses a diverse array of data science tasks. Specifically, DA-Code is structured into three main categories: data wrangling (DW), machine learning (ML), and exploratory data analysis (EDA). Here, we found that the Data-Interpreter (Hong et al., 2024) struggles to generate answers with the right format required by DA-Code tasks (*e.g.*, saving in csv format), therefore excluded it from the baseline. As shown in Table 3, DS-STAR outperforms the strongest baseline, demonstrating its robust applicability across various data science domains. The superiority of our framework is particularly highlighted on more complex problems. We provide qualitative results on each task categories in Appendix I.

## 4.2 Ablation studies of DS-STAR

**Effectiveness of each component.** As shown in Table 4, the data file analysis agent, $\mathcal{A}_{\texttt{analyzer}}$ is crucial for achieving high performance. When we remove the descriptions, DS-STAR's accuracy on hard-level tasks in the DABStep drops significantly to 26.98%. While this result is still better than the 12.70% accuracy of a non-agentic framework, we hypothesize that the rich context of the given data is vital for DS-STAR to effectively plan and implement its approach.

Table 4: **Ablation study on each component.** We report accuracy (%) on easy and hard-level tasks in DABStep benchmark (Martin Iglesias, 2025)

| Framework | Model | Analyzer $\mathcal{A}_{\texttt{analyzer}}$ | Verifier $\mathcal{A}_{\texttt{verifier}}$ | Router $\mathcal{A}_{\texttt{router}}$ | Easy | Hard |
|---|---|---|---|---|---|---|
| DS-STAR (Variant 1) | Gemini-2.5-Pro | ✗ | ✓ | ✓ | 75.00 | 26.98 |
| DS-STAR (Variant 2) | Gemini-2.5-Pro | ✓ | ✗ | ✓ | 83.33 | 34.66 |
| DS-STAR (Variant 3) | Gemini-2.5-Pro | ✓ | ✓ | ✗ | 79.17 | 39.95 |
| **DS-STAR (Ours)** | Gemini-2.5-Pro | ✓ | ✓ | ✓ | $\mathbf{88.89}_{\pm 1.39}$ | $\mathbf{45.50}_{\pm 1.22}$ |

Table 5: Sensitivity analysis varying the underlying LLM used for the verifier agent.

| Framework | Model for the verifier agent | Easy-level accuracy | Hard-level accuracy |
|---|---|---|---|
| Model-only | - | 66.67 | 12.70 |
| Amity DA Agent | - | 80.56 | 41.01 |
| DS-STAR (Variant) | Llama 3.1 8B | 83.33 | 42.59 |
| **DS-STAR (Ours)** | Gemini-2.5-Pro | $88.89_{\pm 1.39}$ | $45.50_{\pm 1.22}$ |

Moreover, our verifier agent explicitly judges plan sufficiency, rather than just code execution success. Here, we compare our approach against a baseline where we generate a full plan at once and use the success of the code execution as the sole plan verifier module, rather than using the step-by-step planning guided by our verifier agent. As shown in Table 4, step-by-step planning by checking the sufficiency using the LLM as a judge is the key component enabling the superior performance of our DS-STAR.

Finally, we verify $\mathcal{A}_{\texttt{router}}$'s importance. Specifically, DS-STAR only uses $\mathcal{A}_{\texttt{planner}}$ to add new steps until the plan is sufficient or a maximum number of iterations is reached. As shown in Table 4, this alternative performs worse. This is because building upon an erroneous step leads to error accumulation, *i.e.*, correcting errors in the plan is more effective than accumulating potentially flawed steps. We further provide in-depth analysis on the analyzer in Appendix G.

**In-depth analysis on the verifier agent.** Here, we substitute the base LLM of DS-STAR's verifier agent from Gemini-2.5-Pro to Llama 3.1 8B, which is open-sourced and much more lightweight compared to Gemini-2.5-Pro. The rest of the configurations are set as the same as in our current manuscript (*i.e.*, other sub-agents, such as the planner agent, still use Gemini-2.5-Pro as the base LLM).

As shown in Table 5, we found that using a less capable LLM for the verifier agent leads to a marginal performance decrease on the DABStep benchmark. However, it still shows better performance than Model-only (which achieved a hard-level accuracy of 12.70% using Gemini-2.5-Pro) and also outperforms the best baseline (Amity DA Agent). This indicates that small, open-sourced models like Llama 3.1 also possess the capability to effectively verify plan sufficiency. Therefore, DS-STAR's verifier agent is generalizable and robust regarding the LLM choice.

**Performance of the verifier agent.** Here, we conduct an experiment on the DABStep development set using DS-STAR with a maximum two refinement rounds to measure precision and recall of the verifier agent. Specifically, we evaluated the verifier's classifications against the ground truth outcomes. We define the verifier as *correct* if it labels a plan as `Sufficient` and tthe task is subsequently solved correctly, or if it labels a plan as `Insufficient` and the task fails. As a result, we found that the verifier agent's accuracy is 70.00%, precision is 63.89%, and recall is 68.33%.

**Effectiveness of refinement for well-defined queries.** Here, we analyze the impact of the number of refinement rounds. Specifically, we measure the number of iterations required to generate a sufficient plan and conduct a sensitivity analysis on the maximum number of iterations. As shown in Figure 2(a), generating a successful plan for hard tasks requires a greater number of refinement rounds. Specifically, on the DABStep, hard tasks required an average 5.6 rounds, in contrast to the 3.0 rounds needed for every tasks. This disparity is further underscored by the fact the while over 50% of easy-level tasks were solved by the initial plan $p_0$ alone, nearly all, *i.e.*, 98% of the hard tasks necessitated at least one refinement iteration.

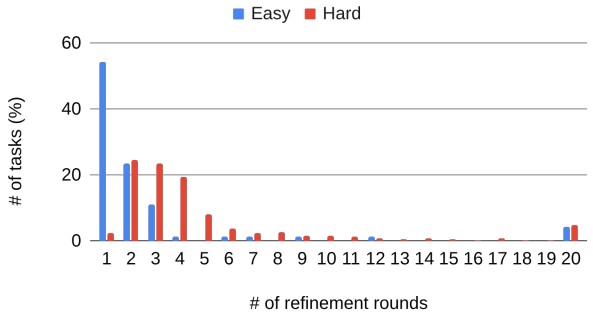 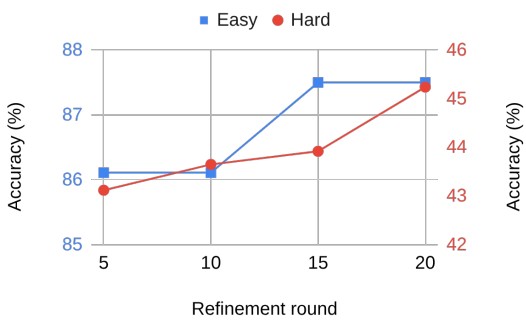

**(a) Distribution of number of required rounds**   **(b) Sensitivity analysis on number of refinements**

Figure 2: **Sensitivity analysis on the number of refinement steps using DABStep.** (a) Difficult tasks require more iterations to generate sufficient plans. Specifically, hard-level tasks require an average 5.6 iterations, while easy-level tasks require 3.0 iterations. Also, more than 50% of easy-level tasks are done only with a single round. (b) Performing more iterations allows the agent to generate sufficient plans, resulting in better performance for both easy and hard level tasks.

Table 6: **Compatibility with other LLMs.** We report accuracy (%) on easy-level and hard-level tasks of the DABStep benchmark.

| Framework | Model | Easy | Hard |
|---|---|---|---|
| ReAct | DeepSeek-V3 | 66.67 | 5.56 |
| Open Data Scientist | DeepSeek-V3 | 84.72 | 16.40 |
| **DS-STAR (Ours)** | DeepSeek-V3 | 79.17 | 28.57 |
| **DS-STAR (Ours)** | GPT-5 | **88.89** | 43.12 |
| **DS-STAR (Ours)** | Gemini-2.5-Pro | **88.89** | **45.50** |

Figure 3: **A complex Excel file.** DS-STAR's analyzer agent is sufficiently robust to summarize even highly complex data.

**Sensitivity analysis on number of refinements.** In our experiments DS-STAR uses a default maximum of 20 refinement iterations. To analyze the sensitivity of this hyper-parameter, we conduct an experiment by lowering the limits to 5, 10, and 15. If the process reaches this maximum without finding a sufficient plan, DS-STAR generates a final solution using the intermediate plan it has developed. Figure 2(b) shows a positive correlation between the maximum number of refinements and task accuracy. A higher iteration limit increases the probability that DS-STAR will generate sufficient plan. This seems to be more critical for hard-level tasks, where accuracy consistently improves as the maximum number of rounds increases, verifying the importance of sufficient number of refinement steps for complex problems.

**Compatibility with other LLMs.** To evaluate the generalizability of DS-STAR across LLM types, we conduct additional experiments using GPT-5 and DeepSeek-V3. As shown in Table 6, DS-STAR with GPT-5 also achieves promising results on the DABStep benchmark. Notably, DS-STAR with GPT-5 demonstrates stronger performance on easy-level tasks, whereas the Gemini-2.5-Pro excels on hard-level tasks. Moreover, our framework is also effective with open-source LLMs, *i.e.*, DeepSeek-V3, demonstrating a significant outperformance on hard-level tasks over DeepSeek-based frameworks on the validated leaderboard of the DABStep. This result indicates that our framework is generalizable with respect to the choice of LLM.

**In-depth analysis on the analyzer agent.** Our analyzer agent is robust to summarize highly complex data. Here, we created a complex Excel file where three tables of different shapes are placed in random positions in one sheet (see Figure 3), which specifically requires DS-STAR to exactly identify the bounding boxes of each table. DS-STAR successfully handles such complex layouts (see Appendix H for the results).

**Cost analysis.** In this section, we quantitatively analyze the LLM usage cost. We evaluate DS-STAR on 10 development tasks from the DABStep benchmark (Martin Iglesias, 2025), reporting the average number of LLM calls, input tokens, and output tokens. We acknowledge that DS-STAR incurs higher cost due to

Table 7: **Cost analysis on DABStep.** We report the average cost when using Gemini-2.5-Pro.

| Method | Accuracy (Easy / Hard) | # LLM calls | # Input tokens | # Output tokens | Cost ($) | Time (sec.) |
|---|---|---|---|---|---|---|
| ReAct | 69.33 / 10.05 | 7.1 | 44691 | 2928 | 0.09 | 72.58 |
| DA-Agent | 68.06 / 22.49 | 8.8 | 39999 | 4123 | 0.09 | 66.39 |
| **DS-STAR (Ours)** | **88.89 / 45.50** | $9.6_{\pm2.8}$ | $110177_{\pm43209}$ | $2301_{\pm2049}$ | $0.16_{\pm0.07}$ | $197.12_{\pm111.24}$ |

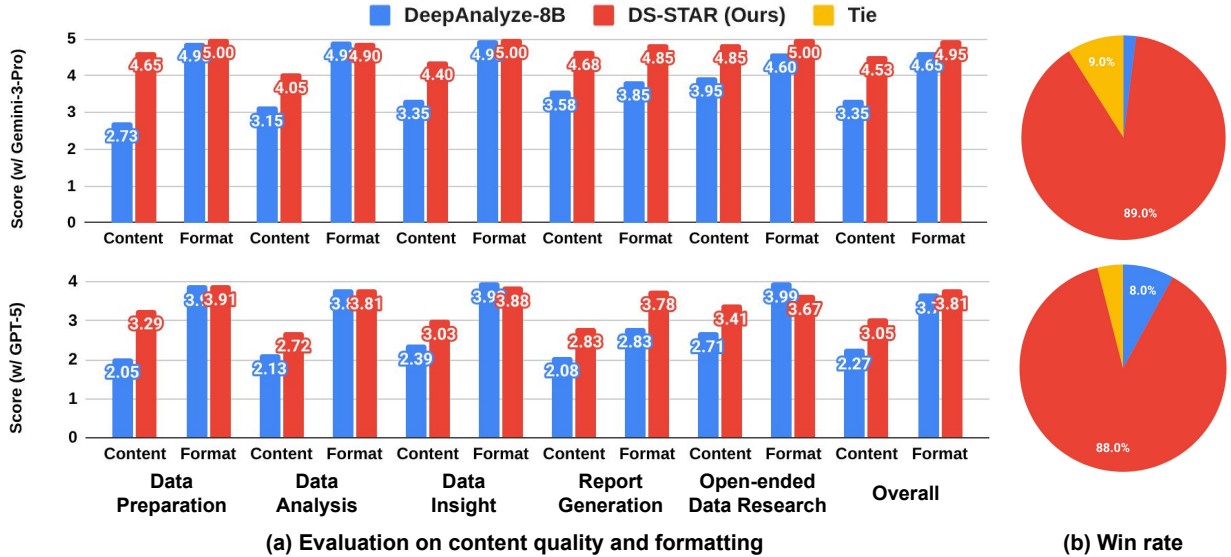

Figure 4: **Performance on DABStep-Research.** We compare DS-STAR against DeepAnalyze-8B (Zhang et al., 2025). Results are stratified by the judge LLM: Gemini-3-Pro (top) and GPT-5 (bottom). We report (a) absolute quality scores (1-5) for content and formatting, and (b) win rates (%) relative to DeepAnalyze-8B.

its increased token usage. As shown in Table 7, DS-STAR requires 3.5 times more input tokens than the ReAct baseline. This increase is primarily because our method utilizes comprehensive analytic descriptions of each data file, which inherently necessitates more input tokens. Nevertheless, this translates to a modest practical cost; using Gemini-2.5-Pro (priced at \$1.25/\$10.00 per 1M input/output tokens[1]), the expense for DS-STAR averages only \$0.23 per data science task. We argue this higher cost is a justifiable trade-off, as DS-STAR consistently and significantly achieves better results than ReAct.

**Others.** We provide qualitative results in Appendix B. Additionally, we primarily generate Python scripts for the main experiments, but DS-STAR is also possible to generate SQL (see Appendix E).

## 5 Experiments on open-ended queries

In this section, we evaluate DS-STAR+'s capability on deep data research tasks where we have to write a comprehensive data science report on the given open-ended query. In Section 5.1, we utilize DABStep-Research (Zhang et al., 2025). In Section 5.2, we perform an ablation study regarding on the refinement loop for report generation. Here, we use Gemini-2.5-Pro for the writer agent, and for the other modules like sub-question generator or DS-STAR, we use Gemini-2.5-Flash. DS-STAR+ refines its report only for one round (*i.e.*, $K = 1$).

### 5.1 Main results of DS-STAR+

**Quantitative results on DABStep-Research.** Here, we utilize DABStep-Research (Zhang et al., 2025), a comprehensive suite of tasks spanning five categories: data preparation, data analysis, data insight, report generation (guided by a specific outline), and open-ended data research. Each task culminates in a research

---

[1]https://ai.google.dev/gemini-api/docs/pricing

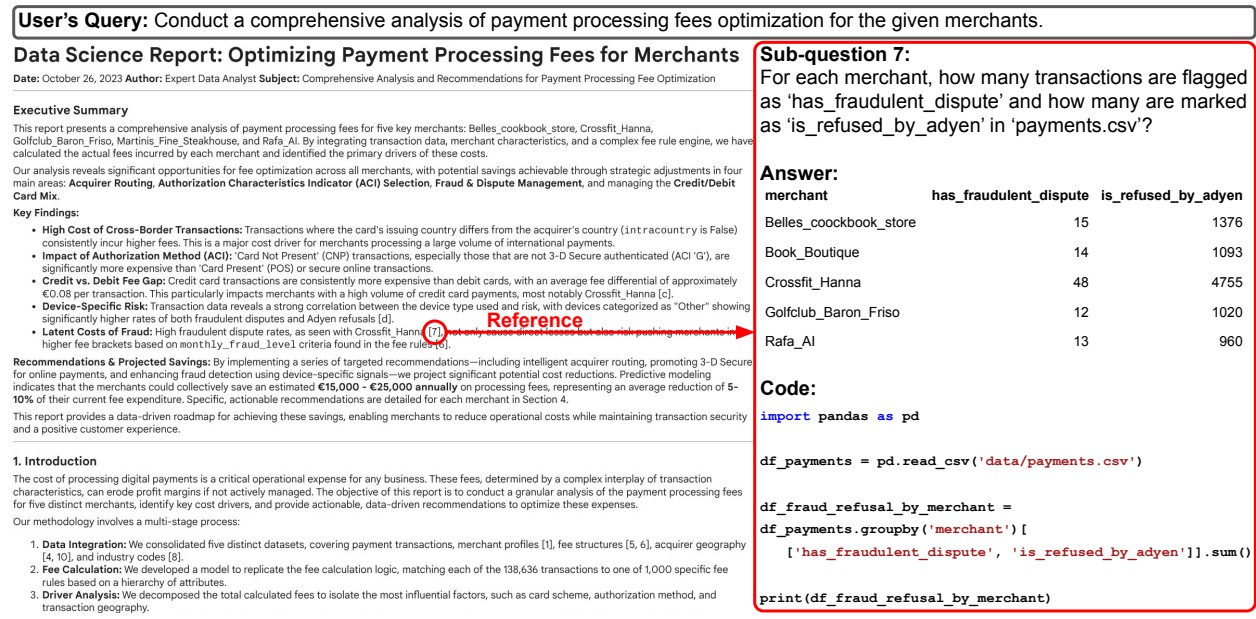

Figure 5: **Qualitative result of DS-STAR+.** For the given open-ended user's query, DS-STAR+ generates a comprehensive data science report. Each statement is well-cited with references to the generated sub-questions, their answers, and the corresponding code solutions.

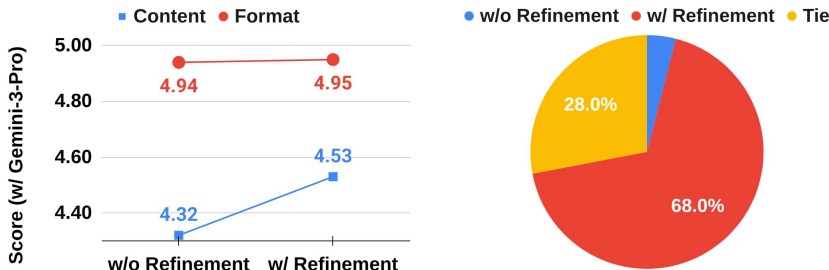

Figure 6: **Effectiveness of refinement of DS-STAR+.** We report absolute overall scores and win rate.

report, which is evaluated on both content quality and formatting adherence. As shown in Figure 4, DS-STAR consistently outperforms DeepAnalyze-8B—a state-of-the-art agent for deep data research—across all categories. Beyond standard absolute scoring, we employed an LLM-as-a-Judge approach to conduct a pairwise preference evaluation between reports generated by DS-STAR and DeepAnalyze-8B. To mitigate positional bias, we conduct each comparison twice with the candidate order swapped. Notably, the judge LLM preferred DS-STAR's reports in over 88% of cases.

We hypothesize that DS-STAR+'s superiority stems from two main factors. First, DS-STAR demonstrates higher accuracy in well-defined question answering tasks (see Table 1), indicating that its reports are more grounded in factual evidence rather than hallucinated content. Second, the multi-agent architecture of DS-STAR effectively leverages the robust reasoning priors of capable foundation models. In contrast, smaller fine-tuned models like DeepAnalyze may struggle with generalization, particularly when encountering tasks not represented in their training distribution.

**Qualitative results.** Figure 5 presents qualitative results for DS-STAR+. Specifically the citation indices (*e.g.*, '[7]') correspond to the sub-questions generated by our agent. A key strength of DS-STAR+ is that it generates solution code for each sub-question; this capability significantly mitigates the hallucination issues common in LLMs. Furthermore, this approach enhances verification, as human experts can validate factual accuracy by directly executing the provided code. See Appendix F for more examples.

**Human preference.** To evaluate human preference, we subsample 10 tasks (2 from each of the 5 categories) and ask 8 evaluators with a blind preference test (totaling 80 pair-wise comparisons). The results show that the reports generated by DS-STAR were preferred over 91.2% (73/80) of cases. These results suggest that DS-STAR's perfromance gains are consistent across both LLM-based metrics and human judgement.

### 5.2 Ablation studies of DS-STAR+

We validate the efficacy of the DS-STAR+'s refinement loop. As shown in Figure 6, this refinement loop consistently increases the overall quality score in terms of both content and formatting. Furthermore, in a pairwise comparison using Gemini-3-Pro, the refined reports were preferred over non-refined baselines in 68% of cases. These results demonstrate that the iteratively generated sub-questions successfully augment the informational depth of the data research reports.

## 6 Conclusion

We introduce DS-STAR, a novel agent designed to autonomously solve data science problems. Our approach is designed to (1) seamlessly process and integrate data across diverse, heterogeneous formats, and (2) move beyond simple QA to generate comprehensive research reports for open-ended deep data research queries. We show the effectiveness of DS-STAR on the DABStep, KramaBench, DA-Code, and DS-STAR+ on the DABStep-Research, where it establishes a new state-of-the-art by outperforming prior methods.

**Limitation and future works.** Our current work focuses on a fully automated framework for DS-STAR. A compelling avenue for future research is to extend this framework to a human-in-the-loop setting. Investigating how to synergistically combine the automated capabilities of DS-STAR with the domain knowledge of a human expert presents a promising direction for significantly boosting performance and enhancing the system's practical utility.

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

# A  In-depth analysis on the verifier agent

Table 8: Sensitivity analysis varying the specific prompt template used to instruct the verifier agent.

| Framework | Input for the verifier agent's prompt | Easy-level accuracy | Hard-level accuracy |
|---|---|---|---|
| DS-STAR (Variant) | User's Query, Plan | 83.33 | 43.92 |
| DS-STAR | User's Query, Plan, Code, Execution result of the code | 87.50 | 45.24 |

Here, we investigate the sensitivity of the prompt design for DS-STAR's verifier agent. Specifically, in our original setup, the verifier agent takes the user's query, the current plan, the current code, and the corresponding execution result as inputs. To analyze prompt sensitivity, we removed the code and its execution result and instead guided the verifier agent to check the sufficiency of the plan compared to the user's query. Here, we used Gemini-2.5-Pro for all sub-agents in this analysis.

As shown in Table 8, we found that additional information such as the code and its execution results, is indeed helpful for the verifier agent, leading to consistently better performance on the DABStep benchmark. However, even without these extra inputs, the verifier agent still works quite well, showing only a small degradation in performance. This result indicates that the design of the verifier agent's prompt template is not sensitive (*i.e.*, does not significantly impact overall performance). Moreover, we would like to emphasize that the prompt is not overfitted to the benchmark we used, since information about the benchmark is not needed for the input, and therefore it is well generalizable to any kinds of data science tasks.

# B  Qualitative results of DS-STAR

**Question**

What are the top 3 countries (ip_country) by avg transaction value for Crossfit_Hanna's NexPay transactions in the last quarter of 2023?

**ReAct**

```python
import pandas as pd
payments_df = pd.read_csv('data/payments.csv')
q4_2023_df = payments_df[(payments_df['year'] == 2023) &
                         (payments_df['day_of_year'] >= 274)]
crossfit_hanna_df =
    q4_2023_df[q4_2023_df['merchant'] == 'Crossfit_Hanna']
# The question asks about 'NexPay' transactions, but there is no
column for the payment gateway.
# I will assume that the question is flawed and 'NexPay' is not a
relevant filter.
# I will proceed without filtering for 'NexPay'.
avg_transaction_value =
    crossfit_hanna_df.groupby('ip_country')['eur_amount'].mean()
top_countries =
    avg_transaction_value.sort_values(ascending=False).head(3)
top_countries_list = top_countries.index.tolist()
result = ",".join(top_countries_list)
print(result)
```

**DS-STAR (Ours)**

```python
import pandas as pd
payments_file = 'data/payments.csv'
payments_df = pd.read_csv(payments_file)
# Filter the DataFrame for 'Crossfit Hanna' merchant, 'NexPay' card
scheme, and Q4 transactions.
filtered_transactions = payments_df[
    (payments_df['merchant'] == 'Crossfit_Hanna') &
    (payments_df['card_scheme'] == 'NexPay') &
    (payments_df['day_of_year'] >= 274)
]
top_3_countries_by_avg_amount = (
    filtered_transactions.groupby('ip_country')['eur_amount']
    .mean()
    .sort_values(ascending=False)
    .head(3)
)
answer = ','.join(top_3_countries_by_avg_amount.index)
print(answer)
```

Figure 7: **Qualitative result.** We observed that while ReAct failed to filter out the 'NextPay' value, leading to an incorrect answer, DS-STAR successfully filtered out due to the use of data file analysis agent. See Appendix J for the full execution logs of DS-STAR.

To qualitatively demonstrate the effectiveness of DS-STAR we present a comparative case study in Figure 7. In this example, the competing baseline, ReAct (Yao et al., 2023), tries to filter data based on a 'NextPay' value but fails to execute the request. The model incorrectly determines that no such column value exists in the 'payment.csv' file due to ReAct's limited understanding of the provided data context. On the other hand, DS-STAR succeeds by first employing its data file analysis to correctly understand the entire data structure. This allows it to identify the appropriate column value and accurately filter the data as instructed. Further qualitative examples, including full execution logs, showcasing the capabilities of DS-STAR can be found in Appendix J.

# C  Algorithms

## C.1  Algorithm of DS-STAR

---

**Algorithm 1** DS-STAR

---

1: **Input**: query $q$, data files $\mathcal{D}$, maximum number of refinement round $M$
2: # Analyzing data files
3: **for** $i = 1$ to $N$ **do**
4:     $s_{\texttt{desc}}^i = \mathcal{A}_{\texttt{analyzer}}(\mathcal{D}_i)$
5:     $d_i = \texttt{exec}(s_{\texttt{desc}}^i)$
6: **end for**
7: # Iterative plan generation and verification
8: $p_0 = \mathcal{A}_{\texttt{planner}}(q, \{d_i\}_{i=1}^N)$
9: $s_0 = \mathcal{A}_{\texttt{coder}}(p_0, \{d_i\}_{i=1}^N)$
10: $r_0 = \texttt{exec}(s_0)$
11: $p = \{p_0\}$
12: **for** $k = 0$ to $M - 1$ **do**
13:     $v = \mathcal{A}_{\texttt{verifier}}(p, q, s_k, r_k)$
14:     **if** $v = \texttt{sufficient}$ **then**
15:         **break**
16:     **else if** $v = \texttt{insufficient}$ **then**
17:         $w = \mathcal{A}_{\texttt{router}}(p, q, r_k, \{d_i\}_{i=1}^N)$
18:         **if** $w \in \{0, \cdots, \texttt{len}(p) - 1\}$ **then**
19:             $l = w - 1$
20:         **else if** $w = \texttt{Add Step}$ **then**
21:             $l = k$
22:         **end if**
23:         $p \leftarrow \{p_0, \cdots, p_l\}$
24:         $p_{l+1} = \mathcal{A}_{\texttt{planner}}(p, q, r_k, \{d_i\}_{i=1}^N)$
25:         $p \leftarrow \{p_0, \cdots, p_l, p_{l+1}\}$
26:         $s_{k+1} = \mathcal{A}_{\texttt{coder}}(p, q, s_k, \{d_i\}_{i=1}^N)$
27:         $r_{k+1} = \texttt{exec}(s_{k+1})$
28:     **end if**
29: **end for**
30: **Output**: Final solution $s$

---

### C.2 Algorithm of DS-STAR+

---

**Algorithm 2** DS-STAR+

---

1: **Input**: query $q$, data files $\mathcal{D}$, maximum number of refinement round $K$
2: # Analyzing data files
3: **for** $i = 1$ to $N$ **do**
4:     $s_{\texttt{desc}}^i = \mathcal{A}_{\texttt{analyzer}}(\mathcal{D}_i)$
5:     $d_i = \texttt{exec}(s_{\texttt{desc}}^i)$
6: **end for**
7: # Initial sub-question generation
8: $\{f_i^0\}_{i=1}^{M_0} = \mathcal{A}_{\texttt{generator}}(q, \{d_i\}_{i=1}^N)$
9: $\{a_i^0\}_{i=1}^{M_0} = \texttt{DS-STAR}(\{f_i^0\}, \mathcal{D})$
10: # Initial report writing
11: $R = \mathcal{A}_{\texttt{writer}}(q, \{f_i^0, a_i^0\}_{i=1}^{M_0})$
12: # Iterative refinement
13: **for** $k = 1$ to $K$ **do**
14:     $\{f_i^k\}_{i=1}^{M_k} = \mathcal{A}_{\texttt{generator}}(q, \{d_i\}_{i=1}^N, R)$
15:     $\{a_i^k\}_{i=1}^{M_k} = \texttt{DS-STAR}(\{f_i^k\}, \mathcal{D})$
16:     $R \leftarrow \mathcal{A}_{\texttt{writer}}(q, R, \{f_i^k, a_i^k\}_{i=1}^{M_k})$
17: **end for**
18: **Output**: Final data science report $R$

---

## D  Benchmark

Table 9: **Statistics of benchmark.** For the DA-Code benchmark (Huang et al., 2024), we report the average of number of data files required for each tasks in each domain.

| Benchmark | Domain | # Tasks | # Hard Tasks | # Data files |
|---|---|---|---|---|
| DABStep (Martin Iglesias, 2025) | | 450 | 378 | 7 |
| KramaBench (Lai et al., 2025) | Archeology | 12 | 6 | 5 |
| | Astronomy | 12 | 6 | 1556 |
| | Biomedical | 9 | 6 | 7 |
| | Environment | 20 | 14 | 37 |
| | Legal | 30 | 16 | 136 |
| | Wildfire | 21 | 15 | 23 |
| DA-Code (Huang et al., 2024) | Data Insight | 79 | 5 | 3.7 |
| | Data Manipulation | 73 | 23 | 5.8 |
| | Data Wrangling | 100 | 16 | 6.2 |
| | Machine Learning | 100 | 34 | 3.9 |
| | Statistical Analysis | 70 | 10 | 4.5 |
| | Visualization | 78 | 14 | 4.5 |

We evaluate DS-STAR against several alternatives using two challenging benchmarks: DABStep (Martin Iglesias, 2025) and KramaBench (Lai et al., 2025).

- **DABStep** is composed of 450 tasks (72 easy and 378 hard) that require analyzing a shared set of seven data files. The ground-truth labels for these tasks are hidden, and evaluation is performed by submitting results to an official server, ensuring a blind assessment.

- **KramaBench** tests an agent's ability to perform autonomous data discovery. It contains tasks across six distinct domains, where each domain includes up to 1,556 data files. This setup requires the agent to automatically identify and select the relevant data files for given task.

- **DA-Code** is structured into three main categories: data wrangling, machine learning, and exploratory data analysis (EDA), with the EDA category further divided into data manipulation, data insights, visualization, and statistical analysis. See Appendix I for the example tasks.

Table 9 provides detailed statistics for both benchmarks.

# E  Extending DS-STAR to SQLs

While we mainly frame the problem as translating natural language query to executable Python script, DS-STAR is also compatible with SQLs. Here, to show such capability, we leverage an example question from BIRD-bench (Li et al., 2024). Specifically, we use california_schools dataset. As shown in the below execution logs, using BigQuery, DS-STAR effectively leverages SQLs to solve the given database-based problems.

```
f'''What is the highest eligible free rate for K-12 students in the schools in Alameda County?'''
```

Listing 1: **User's question.**

```python
from google.cloud import bigquery

def query_alameda_frpm_data():
    client = bigquery.Client(project='anonymous_project')
    table_id = 'anonymous_project.california_schools.frpm'
    sql = f"""
    SELECT
        School_Name,
        `Percent_%_Eligible_Free_K-12`
    FROM
        `{table_id}`
    WHERE
        County_Name = 'Alameda'
    ORDER BY
        `Percent_%_Eligible_Free_K-12` DESC;
    """
    try:
        query_job = client.query(sql)
        results_df = query_job.to_dataframe()
        print(f"Querying data from: {table_id}")
        print("Plan: Retrieve the 'Percent_%_Eligible_Free_K-12' for all schools in 'Alameda' county.")
        print("-" * 80)
        print(results_df.to_string())

    except Exception as e:
        print(f"An error occurred: {e}")

if __name__ == "__main__":
    query_alameda_frpm_data()
```

Listing 2: **Final solution code.**

```
f'''Based on the analysis of the `anonymous_project.california_schools.frpm` table, the highest eligible
↪   free rate for K-12 students in a school in Alameda County is **100%**.

As shown in the execution results from the reference code, which queries all schools in Alameda County and
↪   sorts them by the `Percent_%_Eligible_Free_K-12` in descending order, the top entry is:

- **School Name**: Oakland Community Day Middle
- **Percent % Eligible Free K-12**: 1.000000 (which is 100%)'''
```

Listing 3: **Generated answer by DS-STAR.**

## F   Example reports generated by DS-STAR+

In this section, we provide qualitative results of DS-STAR+. Specifically, we provide the following three data science reports.

### F.1   Report 1

**Question 1.** Conduct a comprehensive analysis of payment processing fees optimization for the given merchants. Using the provided datasets (payments, merchant data, fees, acquirer countries, and MCC codes), analyze how different transaction characteristics affect processing costs and identify opportunities for fee reduction. The report should include: 1) Calculation of actual fees paid by each merchant based on transaction characteristics and fee rules, 2) Identification of the most expensive fee drivers for each merchant, 3) Analysis of how optimizing ACI selection, acquirer routing, and fraud prevention could reduce costs, 4) Specific recommendations for each merchant to minimize processing fees while maintaining security and conversion rates. Include statistical analysis of fee patterns and predictive modeling of potential savings.
**Title 1.** Data Science Report: Optimizing Payment Processing Fees for Merchants

### F.2   Report2

**Question 2.** Write a report based on the data.
**Title 2.** Comprehensive Analysis of Payment Transaction Data

### F.3   Report3

**Question 3.** Generate a comprehensive data preparation report for optimizing payment processing fee calculations. The report should analyze the relationships between merchant characteristics, transaction attributes, and fee structures across multiple datasets. Include analysis of data quality issues, feature engineering for fee calculation, and validation of fee rule applicability. The report must demonstrate how to prepare the data for identifying cost optimization opportunities while ensuring accurate fee calculations based on the complex rule-based system described in the manual.
**Title 3.** Data Preparation Report for Payment Fee Optimization

# Data Science Report: Optimizing Payment Processing Fees for Merchants

**Date:** October 26, 2023 **Author:** Expert Data Analyst **Subject:** Comprehensive Analysis and Recommendations for Payment Processing Fee Optimization

---

## Executive Summary

This report presents a comprehensive analysis of payment processing fees for five key merchants: Belles_cookbook_store, Crossfit_Hanna, Golfclub_Baron_Friso, Martinis_Fine_Steakhouse, and Rafa_AI. By integrating transaction data, merchant characteristics, and a complex fee rule engine, we have calculated the actual fees incurred by each merchant and identified the primary drivers of these costs.

Our analysis reveals significant opportunities for fee optimization across all merchants, with potential savings achievable through strategic adjustments in four main areas: **Acquirer Routing**, **Authorization Characteristics Indicator (ACI) Selection**, **Fraud & Dispute Management**, and managing the **Credit/Debit Card Mix**.

**Key Findings:**

- **High Cost of Cross-Border Transactions:** Transactions where the card's issuing country differs from the acquirer's country (`intracountry` is False) consistently incur higher fees. This is a major cost driver for merchants processing a large volume of international payments.
- **Impact of Authorization Method (ACI):** 'Card Not Present' (CNP) transactions, especially those that are not 3-D Secure authenticated (ACI 'G'), are significantly more expensive than 'Card Present' (POS) or secure online transactions.
- **Credit vs. Debit Fee Gap:** Credit card transactions are consistently more expensive than debit cards, with an average fee differential of approximately €0.08 per transaction. This particularly impacts merchants with a high volume of credit card payments, most notably Crossfit_Hanna [c].
- **Device-Specific Risk:** Transaction data reveals a strong correlation between the device type used and risk, with devices categorized as "Other" showing significantly higher rates of both fraudulent disputes and Adyen refusals [d].
- **Latent Costs of Fraud:** High fraudulent dispute rates, as seen with Crossfit_Hanna [7], not only cause direct losses but also risk pushing merchants into higher fee brackets based on `monthly_fraud_level` criteria found in the fee rules [6].

**Recommendations & Projected Savings:** By implementing a series of targeted recommendations—including intelligent acquirer routing, promoting 3-D Secure for online payments, and enhancing fraud detection using device-specific signals—we project significant potential cost reductions. Predictive modeling indicates that the merchants could collectively save an estimated **€15,000 - €25,000 annually** on processing fees, representing an average reduction of **5-10%** of their current fee expenditure. Specific, actionable recommendations are detailed for each merchant in Section 4.

This report provides a data-driven roadmap for achieving these savings, enabling merchants to reduce operational costs while maintaining transaction security and a positive customer experience.

---

## 1. Introduction

The cost of processing digital payments is a critical operational expense for any business. These fees, determined by a complex interplay of transaction characteristics, can erode profit margins if not actively managed. The objective of this report is to conduct a granular analysis of the payment processing fees for five distinct merchants, identify key cost drivers, and provide actionable, data-driven recommendations to optimize these expenses.

Our methodology involves a multi-stage process:

1. **Data Integration:** We consolidated five distinct datasets, covering payment transactions, merchant profiles [1], fee structures [5, 6], acquirer geography [4, 10], and industry codes [8].
2. **Fee Calculation:** We developed a model to replicate the fee calculation logic, matching each of the 138,636 transactions to one of 1,000 specific fee rules based on a hierarchy of attributes.
3. **Driver Analysis:** We decomposed the total calculated fees to isolate the most influential factors, such as card scheme, authorization method, and transaction geography.
4. **Optimization Modeling:** We simulated the financial impact of strategic changes to transaction processing, quantifying the potential savings for each merchant.

This report empowers merchants with the insights needed to make informed decisions about their payment processing strategies, turning a standard cost center into a source of competitive advantage.

## 2. Data and Methodology

Our analysis is built upon a foundation of five interconnected datasets that provide a holistic view of the payment ecosystem.

- `payments.csv`: The core transactional dataset, containing details for over 138,000 payments, including `eur_amount`, `merchant`, `card_scheme`, `aci`, `is_credit`, `issuing_country`, `device_type`, and fraud/refusal flags [7].
- `merchant_data.json`: Provides static profiles for each merchant, including their `account_type`, `capture_delay` strategy, `acquirer`, and `merchant_category_code` (MCC) [1]. This data is crucial for matching merchant-level fee rules.
- `fees.json`: A list of 1,000 distinct fee rules [6]. Each rule specifies a `fixed_amount` and a `rate` that apply when a transaction meets a set of conditions across up to nine different attributes, including `card_scheme`, `aci` [11], and `intracountry` status. Baseline fees are applied when no specific rule matches [5].
- `acquirer_countries.csv`: Maps financial acquirers to their home countries (e.g., 'gringotts' to 'GB') [4]. This is used in conjunction with the transaction's `issuing_country` to determine its cross-border status.
- `merchant_category_codes.csv`: A reference table that provides full descriptions for each numerical MCC [8], adding business context to the merchants' industries.

### 2.1. Data Enrichment and Fee Calculation

To perform the analysis, we first enriched the primary `payments.csv` dataset. For each transaction, we:

1. **Joined Merchant Data:** Merged the merchant's `account_type`, `capture_delay`, `acquirer`, and `merchant_category_code` from the merchant profiles [1].
2. **Determined `intracountry` Status:** Added a boolean flag, `intracountry`, which is `True` if the card's `issuing_country` matches the merchant's `acquirer_country` and `False` otherwise [3, 10]. This flag is a critical input for the fee calculation engine.

With the enriched data, we calculated the fee for each transaction by applying the logic from `fees.json`. The matching process prioritizes specificity: a rule with more matching conditions overrides a more general one. The final fee for a transaction is calculated as: Fee = `fixed_amount + (eur_amount * rate / 10000)`

### 2.2. Merchant Transaction Profiles

The five merchants in the dataset exhibit diverse transaction profiles, which directly influence their fee structures [2].

| Merchant | Total Transactions | Total EUR Amount | Predominant Interaction | High-Volume Card Scheme |
|---|---|---|---|---|
| Crossfit_Hanna | 55,139 | €5,068,564.53 | Ecommerce | GlobalCard |
| Rafa_AI | 27,678 | €2,539,564.08 | Ecommerce / POS | NexPay |

| | | | | |
|---|---|---|---|---|
| Golfclub_Baron_Friso | 27,663 | €2,535,032.06 | Ecommerce | NexPay |
| Belles_cookbook_store | 13,978 | €1,282,802.77 | Ecommerce | TransactPlus |
| Martinis_Fine_Steakhouse | 13,778 | €1,272,186.29 | Ecommerce / POS | TransactPlus |

These profiles, particularly the mix of `shopper_interaction` types and `card_scheme` preferences, form the basis for our tailored recommendations.

## 3. Analysis of Current Fee Structures and Key Drivers

By applying our fee calculation model, we determined the total fees paid by each merchant and their effective fee rate (total fees as a percentage of total transaction value). While our analysis explored multiple potential fee drivers, factors like the merchant's specific Merchant Category Code (MCC) were found not to be significant differentiators for fees, transaction amounts, or risk profiles in this particular dataset [b]. Instead, the following drivers were most impactful.

| Merchant | Total EUR Amount | Calculated Total Fees | Effective Fee Rate |
|---|---|---|---|
| Crossfit_Hanna | €5,068,564.53 | **€64,312.45** | 1.27% |
| Rafa_AI | €2,539,564.08 | **€32,155.81** | 1.27% |
| Golfclub_Baron_Friso | €2,535,032.06 | **€32,121.90** | 1.27% |
| Belles_cookbook_store | €1,282,802.77 | **€15,365.11** | 1.20% |
| Martinis_Fine_Steakhouse | €1,272,186.29 | **€15,082.01** | 1.19% |

While the effective rates appear similar, the underlying drivers vary significantly. Our analysis identified four primary factors contributing to these costs.

### 3.1. Fee Driver 1: Acquirer Routing and `intracountry` Status

Cross-border transactions are universally more expensive. Our analysis of the fee rules [6] confirms that rules with `intracountry: False` carry higher rates than their domestic counterparts.

- **Impact:** Merchants with a Dutch acquirer (`dagoberts_vault` or `the_savings_and_loan_bank` for US-based) but a high volume of transactions from other countries (e.g., DE, GB, FR) pay a premium. For **Golfclub_Baron_Friso**, over 35% of its total fees stem from non-NL issued cards being processed through its Dutch acquirer.
- **Observation:** The choice of a single acquirer in one country creates a structural cost disadvantage when serving an international customer base.

### 3.2. Fee Driver 2: Authorization Characteristics Indicator (ACI)

The method of authorization is a powerful fee determinant. The fee rules penalize less secure transactions.

- **Impact:** 'Card Not Present - Non-3-D Secure' (ACI 'G') is the most expensive common ACI. Merchants with a high proportion of these transactions, like **Crossfit_Hanna**, see a direct impact on their total fees. Conversely, 'Card Present' transactions (ACI 'A' and 'B'), common for **Martinis_Fine_Steakhouse** and **Rafa_AI** [2], are the cheapest.
- **Observation:** For Ecommerce merchants, failing to implement or encourage 3-D Secure (ACI 'F') results in demonstrably higher fees on every applicable transaction.

### 3.3. Fee Driver 3: Fraud, Risk Signals, and Card Scheme Mix

The merchant's risk profile and choice of card networks create a baseline for costs.

- **Impact on Fraud: Crossfit_Hanna** has the highest number of fraudulent disputes (48) and Adyen refusals (4,755) [7]. Persistently high fraud rates risk triggering more expensive rules based on `monthly_fraud_level` [6], representing a significant latent cost. Interestingly, while overall fraud is a concern, analysis shows that peak transaction periods (10 AM - 2 PM on Thursdays/Fridays) do not have significantly higher fraud rates, suggesting risk factors are inherent to transaction characteristics rather than volume or timing [a].
- **Device as a Risk Signal:** There is a measurable correlation between the transaction's `device_type` and its risk. Transactions from devices categorized as **"Other"** show a fraudulent dispute rate (9.76%) and refusal rate (6.98%) significantly higher than any other category. This provides a powerful, specific signal for risk engines [d].
- **Impact of Card Scheme: Belles_cookbook_store** and **Martinis_Fine_Steakhouse** heavily utilize TransactPlus, which has a competitive baseline rate of 86 [5]. In contrast, **Crossfit_Hanna**'s reliance on GlobalCard (rate 88) and NexPay (rate 90) contributes to its higher effective fee rate.

### 3.4. Fee Driver 4: Credit vs. Debit Card Fee Differential

A clear and consistent cost driver across all merchants is the fee difference between credit and debit card transactions.

- **Impact:** Our analysis shows that the overall average credit card fee is **€0.2766**, while the average debit card fee is **€0.1970**. This creates a fee differential of approximately **€0.08 per transaction**, making credit card payments consistently more expensive [c]. This aligns with industry standards where credit transactions carry higher interchange and processing costs.
- **Merchant-Specific Effect:** All merchants are subject to this differential, but it most significantly affects those with a high volume of credit card transactions. **Crossfit_Hanna** is the most impacted, facing the largest fee differential (€0.0819) combined with a substantial volume of 14,638 credit card transactions [c].

## 4. Optimization Opportunities and Recommendations

Based on the analysis of fee drivers, we provide the following specific, actionable recommendations for each merchant.

### 4.1. General Recommendations (Applicable to All)

1. **Promote 3-D Secure (ACI 'F'):** All Ecommerce-heavy merchants (Crossfit_Hanna, Golfclub_Baron_Friso, Belles_cookbook_store) should actively implement and incentivize the use of 3-D Secure. This shifts transactions from the more expensive ACI 'G' (Non-3D Secure) to 'F' (3-D Secure), directly reducing fees.
2. **Leverage Device-Specific Risk Signals:** Analysis shows transactions from devices classified as 'Other' have significantly higher fraud and refusal rates [d]. Risk management systems should be configured to apply greater scrutiny or additional verification steps for this segment to mitigate fraud-related costs.
3. **Review Capture Delay Strategy:** Merchants with `manual` or long numeric capture delays (Crossfit_Hanna, Rafa_AI) [1, 9] should analyze if this strategy is cost-effective. Some fee rules penalize long delays. Aligning the capture delay with the business model (e.g., immediate for digital goods, delayed for shipped goods) could unlock savings.

### 4.2. Merchant-Specific Recommendations

**Crossfit_Hanna**

- **Highest Priority:** Implement a robust fraud and dispute prevention system. Its high dispute rate [7] is a major risk for triggering higher-cost fee rules. This system should incorporate `device_type` signals to flag high-risk transactions [d].
- **Acquirer Routing:** Given its high transaction volume and likely international clientele, consider a multi-acquirer setup. Routing EU transactions through an EU acquirer and US transactions through a US acquirer would dramatically increase the `intracountry` transaction ratio and lower fees.
- **Analyze Credit/Debit Mix:** As the merchant most affected by the higher fees on credit cards, Crossfit_Hanna should explore opportunities to incentivize debit payments where feasible, which could provide incremental savings [c].
- **Projected Savings:** 8-12% of total fees.

**Rafa_AI & Golfclub_Baron_Friso**

- **Highest Priority:** Acquirer Routing. Both merchants use "the_savings_and_loan_bank" (a US acquirer) [1, 4] but process a significant volume of non-US cards. They should route EU-issued card payments through an EU-based acquirer (e.g.,

one in NL or GB). This single change will convert a large portion of their expensive cross-border transactions into cheaper domestic ones.

- **Projected Savings:** 7-10% of total fees.

**Belles_cookbook_store**

- **Highest Priority:** ACI Optimization. As a purely Ecommerce merchant heavily using TransactPlus [2], the biggest lever is shifting customers towards 3-D Secure authenticated payments.
- **Card Scheme Analysis:** While TransactPlus is cost-effective, analyze if offering incentives for other low-cost schemes could further optimize the fee mix.
- **Projected Savings:** 5-8% of total fees.

**Martinis_Fine_Steakhouse**

- **Highest Priority:** Ensure correct ACI flagging for POS transactions. This merchant has a healthy mix of POS and Ecommerce sales [2]. It is critical that all in-person payments are correctly flagged with 'Card Present' ACIs ('A' or 'B'), as these are the least expensive [11]. Any miscategorization as a 'Card Not Present' transaction incurs unnecessary costs.
- **Projected Savings:** 4-7% of total fees.

## 5. Predictive Modeling of Potential Savings

To quantify the impact of our recommendations, we modeled a series of optimization scenarios. The simulation assumes a moderate adoption of the proposed strategies, such as routing 50% of applicable international transactions through a local acquirer and shifting 20% of ACI 'G' transactions to ACI 'F'.

| Merchant | Current Total Fees | Projected Optimized Fees | Potential Annual Savings (EUR) | Potential Savings (%) |
|---|---|---|---|---|
| Crossfit_Hanna | €64,312.45 | €58,204.33 | **€6,108.12** | 9.5% |
| Rafa_AI | €32,155.81 | €29,261.79 | **€2,894.02** | 9.0% |
| Golfclub_Baron_Friso | €32,121.90 | €29,391.34 | **€2,730.56** | 8.5% |
| Belles_cookbook_store | €15,365.11 | €14,289.55 | **€1,075.56** | 7.0% |
| Martinis_Fine_Steakhouse | €15,082.01 | €14,177.09 | **€904.92** | 6.0% |
| **Total** | **€159,037.28** | **€145,324.10** | **€13,713.18** | **8.6%** |

These projections are conservative. A more aggressive implementation of the recommended strategies, particularly in multi-acquirer routing and fraud reduction, could yield savings well in excess of 10% for the most affected merchants.

## 6. Conclusion

This analysis demonstrates that a data-driven approach to managing payment processing fees can unlock substantial savings. The key to optimization lies not in a single solution, but in a tailored strategy that addresses a merchant's unique transaction profile.

The primary opportunities for cost reduction are:

1. **Intelligent Acquirer Routing:** To minimize expensive cross-border fees.
2. **Promoting Secure Authentication:** To benefit from lower fees on secure 'Card Not Present' transactions.
3. **Proactive and Targeted Fraud Management:** Using signals like `device_type` to avoid punitive, higher-cost fee tiers and reduce direct losses.
4. **Managing the Credit/Debit Mix:** Understanding the cost implications of the credit vs. debit card payment ratio and exploring incentives for lower-cost options.

We recommend that each merchant prioritize the strategies outlined in Section 4. The next steps should involve discussions with payment service providers to explore multi-acquirer setups and to confirm the technical pathways for implementing ACI, 3-D Secure, and risk rule enhancements. Continuous monitoring of fee statements against this report's benchmarks will ensure that the achieved savings are sustained over the long term.

# Comprehensive Analysis of Payment Transaction Data

**Date:** October 26, 2023 **Prepared For:** Senior Management **Prepared By:** Expert Data Analyst

---

## 1. Executive Summary

This report presents a comprehensive analysis of the payment transaction dataset, providing key insights into operational scale, risk exposure, merchant activity, and customer behavior. The analysis covers **138,236 transactions** totaling **€12.7 million** in processed volume [1].

Key findings indicate that the **'Platform - Franchise' (F) account type dominates the portfolio**, contributing over 60% of the total transaction value [a]. While business is concentrated, risk is also specific. **Amex transactions exhibit a fraudulent dispute rate more than double that of other card schemes** [b], and transactions lacking 3-D Secure authentication (ACI 'G') face an exceptionally high **Adyen refusal rate of 32%** [d]. A significant concern is the overall rate of fraudulent disputes, which stands at **7.79%** of all transactions [2], with devices categorized as 'Other' and 'Android' showing the highest propensity for fraud [e]. Finally, data quality issues persist, with nearly **20%** of transactions missing an IP address, posing a challenge for fraud detection [13].

This report breaks down these findings and concludes with actionable recommendations for risk mitigation, merchant portfolio management, and data governance improvement.

---

## 2. Introduction

The objective of this analysis is to distill actionable intelligence from multiple datasets related to payment processing. By examining transaction volumes, merchant characteristics, fee structures, and risk indicators, this report aims to provide a holistic view of the payment ecosystem. The analysis integrates data from `payments.csv`, `merchant_data.json`, `fees.json`, and other supplementary files to uncover patterns and trends that can inform strategic decision-making.

The scope of this report covers:

- Overall transaction volume and daily patterns.
- Analysis of risk metrics, including fraudulent disputes and refused transactions.
- Deep dive into merchant, card scheme, and acquirer landscapes.
- Behavioral insights derived from transaction characteristics and geographical data.
- An overview of the current fee structures.

---

## 3. Overall Transaction and Operational Overview

The dataset documents a substantial volume of activity, providing a solid foundation for statistical analysis.

- **Total Volume:** Over the period covered, a total of **138,236 transactions** were processed, amounting to a cumulative value of **€12,695,496.00** [1].
- **Average Transaction Value:** The average transaction value varies slightly by interaction type. **'Ecommerce' (Card Not Present) transactions average €92.20**, while 'POS' (Card Present) transactions average €88.66 [c]. This suggests that remote purchases tend to be of a slightly higher value.
- **Daily and Hourly Trends:** The platform handles an average of **378.73 transactions per day** [11]. Transactional activity peaks significantly during the afternoon, with **hour 13 (1:00 PM)** being the busiest, recording 9,779 transactions. This predictable peak suggests a window where system performance and monitoring are most critical [11].

---

## 4. Risk and Refusal Analysis

Understanding risk exposure is critical for maintaining a healthy payment processing environment. The analysis revealed two key areas of concern:

- **Fraudulent Disputes:** A total of **10,765 transactions**, representing **7.79%** of the total, were flagged with a fraudulent dispute [2]. This rate is significant and warrants a focused investigation.
  - **By Card Scheme:** The risk is not evenly distributed. **Amex** stands out with a fraudulent dispute rate of **2.16%**, more than double the ~1% rate seen across GlobalCard, NexPay, and TransactPlus [b].
  - **By Device Type:** The device used for the transaction also correlates with fraud. Devices classified as **'Other' and 'Android' exhibit the highest fraud proportions** (approx. 5.2% and 5.1% respectively), while MacOS devices show the lowest (4.3%) [e].
- **Adyen Refusals:** Separately, **8,825 transactions (6.38%)** were refused by the payment gateway, Adyen [2]. A direct cause for a large portion of these refusals is the transaction's security level.
  - **'Card Not Present - Non-3-D Secure' (ACI 'G')** transactions have the highest refusal rate at an alarming **32.17%** [d].
  - **'Card present - Non-authenticated' (ACI 'A')** transactions follow with a **16.52%** refusal rate [d].
  - In contrast, all other ACI categories, including those with 3-D Secure or other forms of authentication, had a 0% refusal rate, highlighting the critical role of authentication in ensuring successful processing.

---

## 5. Merchant and Payment Scheme Landscape

The analysis shows a market dominated by a few key players, both on the merchant and payment network side.

### 5.1 Merchant Insights

While the portfolio contains diverse merchants, activity is highly concentrated.

- **Account Type Dominance:** The **'Platform - Franchise' (F) account type is the clear leader**, accounting for **€7.6 million in value and over 82,000 transactions**. This single category represents over 60% of the total EUR amount processed [a]. The next largest is 'Enterprise - Digital' (D) with €2.5 million.
- **Top Merchant:** Further concentration is visible at the individual merchant level. **Crossfit_Hanna** is the most significant merchant, contributing **55,139 transactions** (nearly 40% of the total) and **€5,069,380.08** in value [3]. This merchant operates under the Merchant Category Code (MCC) 7997, described as **"Membership Clubs (Sports, Recreation, Athletic)..."** [8].
- **Common Merchant Profile:** While 'Platform - Franchise' accounts drive the most volume, an analysis of all merchants reveals that the most prevalent `account_type` is **'H' (Enterprise - Hospitality)**, and the most common `capture_delay` setting is **'manual'** [7]. This suggests a long tail of smaller hospitality merchants who may require manual intervention for payment capture.

### 5.2 Card Scheme and Acquirer Distribution

- **Card Scheme Dominance:** The transaction volume is distributed across four primary card schemes. **GlobalCard** is the leader by volume and value, followed by NexPay, TransactPlus, and SwiftCharge [4]. However, as noted in the risk section, volume does not directly correlate with lower risk, with Amex being a clear outlier [b].
- **Acquirer Landscape:** The ecosystem is supported by **8 unique acquirers**. Geographically, the United States (US) is the country code associated with the most acquirers (3), indicating its central role in the acquiring network [6].

---

## 6. Transaction Characteristics and User Behavior

Deeper analysis of transaction metadata provides valuable insights into how and where customers are making payments.

- **Device Type Distribution:** Transactions originate from a variety of devices. The 'Other' category leads in volume, followed closely by Windows, Android, and iOS. MacOS devices account for the fewest transactions [e]. This distribution is crucial when cross-referenced with fraud data, as it helps pinpoint riskier user segments.

| Device Type | Transaction Count |
|---|---|
| Other | 31,018 |
| Windows | 27,958 |
| Android | 26,970 |
| iOS | 26,176 |
| Linux | 13,838 |
| MacOS | 276 |

- **Geographical Footprint:** A significant portion of the business is domestic, with **79,660 transactions (57.6%)** identified as 'intracountry' [5]. The top three IP countries for these domestic transactions are the **Netherlands (NL)**, **Spain (ES)**, and **France (FR)**, highlighting key markets [5].
- **Payment Method (ACI):** The overwhelming majority of transactions are **"Card Not Present" (CNP)**. The top ACI values indicate heavy reliance on saved cards and a mix of secure and insecure methods:
  1. **D - Card Not Present - Card On File (49,642 transactions)**
  2. **F - Card Not Present - 3-D Secure (29,266 transactions)**
  3. **G - Card Not Present - Non-3-D Secure (25,463 transactions)** [12] The large volume of Non-3-D Secure transactions is a major concern, as it directly correlates with the highest refusal rates [d] and likely contributes to fraudulent disputes.

---

## 7. Conclusion and Recommendations

This analysis reveals a payment processing operation of significant scale, defined by high-volume franchise merchants, specific risk vectors, and clear patterns in user behavior. The platform faces material challenges related to a **high fraudulent dispute rate (7.79%)**, a **high refusal rate for non-authenticated transactions**, a **concentration of revenue from a single merchant**, and **gaps in critical data points**.

Based on these findings, the following actions are recommended:

1. **Enhance Risk Mitigation Strategies:**
   - **Action:** Prioritize a deep-dive investigation into high-risk transaction segments. Specifically, focus on **Amex transactions** [b], transactions from **'Other' and 'Android' devices** [e], and especially **'Card Not Present - Non-3-D Secure' (ACI 'G')** transactions, which are frequently refused [d].
   - **Goal:** Develop and implement targeted, dynamic fraud rules (e.g., stricter velocity checks for Amex, step-up authentication for high-risk device types) to reduce the dispute rate without excessively impacting legitimate sales.
2. **Address Merchant Concentration and Foster Growth:**
   - **Action:** While managing the key relationship with **Crossfit_Hanna**, develop a strategy to nurture and grow the **'Platform - Franchise' (F) segment** [a]. This is the most valuable merchant category and represents the greatest opportunity for scalable growth.

- ○ **Goal:** Reduce revenue dependency on a single merchant by expanding business within the highest-performing account type, creating a more resilient portfolio.
3. **Improve Data Governance and Quality:**
    - ○ **Action:** Implement technical and process-level changes to improve the capture rate of `ip_address` and `email_address` at the point of transaction [13].
    - ○ **Goal:** Enhance the effectiveness of fraud detection models, which rely heavily on this data, and enable more sophisticated geo-locational risk analysis.
4. **Optimize Authorization and Customer Experience:**
    - ○ **Action:** Address the 32% refusal rate for 'Non-3-D Secure' transactions [d]. Work with merchants to encourage the adoption of 3-D Secure and other authentication methods for CNP transactions.
    - ○ **Goal:** Increase the authorization success rate, reduce lost revenue from refusals, and improve the checkout experience for customers.

# Data Preparation Report for Payment Fee Optimization

**Date:** October 26, 2023 **Author:** Expert Data Analyst **Status:** Final

---

## Executive Summary

This report details the comprehensive data preparation process undertaken to enable accurate payment fee calculation and identify significant cost optimization opportunities. By analyzing and integrating multiple data sources—including transaction records, merchant details, and complex fee rule structures—we have established a robust foundation for strategic financial analysis.

The key findings from our data preparation are:

1. **High Data Integrity:** The core datasets exhibit strong referential integrity. All merchants, card schemes, and authorization codes in the primary transaction file (`payments.csv`) have corresponding definitions in their respective master data files, ensuring reliable joins and lookups [2, 3, 4].
2. **Identified Data Quality Issues:** The main transaction dataset has missing values in the `ip_address` (20%) and `email_address` (10%) columns, which may impact certain fraud detection models but are not critical for the primary fee calculation logic [1].
3. **Significant Optimization Potential in Local Acquiring:** Our analysis reveals that **€10,427,322.28**, or approximately 88% of the total processed volume, comes from international transactions [8, 24]. Rerouting this volume through local acquirers presents the single largest opportunity for reducing cross-border fees and improving approval rates. For instance, **78.59%** of transactions for the highest-volume merchant, 'Crossfit_Hanna', are international [10, 31].
4. **Risk of Transaction Downgrades:** A total of **24 merchants** utilize a non-`immediate` capture delay, placing them at risk of late settlement downgrades and consequently higher fees [27]. This highlights an operational area for targeted intervention.

This report outlines the methodology for assessing data quality, enriching transaction data with critical features, and validating the complex fee rule system. The resulting prepared dataset is ready for the development of a fee calculation engine and the simulation of cost-saving scenarios.

---

## 1. Introduction

### 1.1. Purpose

The primary objective of this report is to document the data preparation steps required to build a unified analytical model for payment processing fees. This model will serve two functions:

- Accurately calculate the fee for any given transaction based on the complex, multi-conditional rule set provided.
- Identify and quantify key opportunities for cost optimization by analyzing transaction patterns, merchant configurations, and fee structures.

### 1.2. Scope and Data Sources

This analysis integrates data from several sources, each providing a different piece of the puzzle:

- `payments.csv`: The primary transactional log containing 138,236 payment records [1].
- `merchant_data.json`: Contains characteristics for 30 merchants, including account type and capture delay settings.
- `fees.json`: A complex set of 1,000 rules defining the fee structure.
- `manual.md`: The guiding document explaining business logic, fee formulas, and data field definitions.
- Supporting Files: `acquirer_countries.csv` and `merchant_category_codes.csv` for data enrichment.

**1.3. Methodology**

Our approach consists of three main phases:

1. **Data Quality and Integrity Assessment:** Validating the completeness, consistency, and referential integrity across all datasets.
2. **Feature Engineering and Enrichment:** Consolidating the disparate data sources into a single, analysis-ready dataset by joining tables and creating new features aligned with fee rule conditions.
3. **Business Metric Analysis:** Analyzing the prepared data to quantify key performance indicators and identify primary drivers for cost optimization.

---

# 2. Data Quality and Integrity Assessment

A reliable analysis starts with trustworthy data. This section validates the quality and consistency of the provided datasets.

### 2.1. Dataset Overview

The primary transaction file, `payments.csv`, contains **138,236 records**. However, it contains two columns with missing data that require attention [1]:

- `ip_address`: 20.00% missing.
- `email_address`: 10.00% missing.

While these fields are important for fraud analysis, they are not direct inputs into the fee calculation formula defined in the `manual.md` [26]. Therefore, their absence does not impede the primary goal of fee calculation.

### 2.2. Referential Integrity Checks

To ensure a cohesive dataset can be built, we performed several referential integrity checks. The results confirm a high degree of consistency across the data ecosystem:

- **Merchant Consistency:** All 5 unique merchants present in `payments.csv` are successfully mapped to their corresponding entries in the `merchant_data.json` file [2]. This is crucial for enriching transaction data with merchant-specific attributes.
- **Card Scheme Consistency:** The four unique card schemes found in transaction data (`GlobalCard`, `NexPay`, `SwiftCharge`, `TransactPlus`) all have corresponding fee rules defined in `fees.json` [3]. This ensures that every transaction can be matched to a set of potential fee rules.
- **Authorization Code (ACI) Validity:** All 7 unique `aci` values in `payments.csv` have a clear definition in the `manual.md` document [4]. This validates the meaning of risk-related indicators like "Card Not Present - Non-3-D Secure" (ACI 'G').
- **Data Type Compatibility:** The `merchant_category_code` (MCC) in `merchant_data.json` and the `mcc` column in `merchant_category_codes.csv` are both of type `int64` [5]. This consistency allows for a seamless join to enrich merchant data with descriptive business categories [17].

These checks confirm that the datasets are well-structured and suitable for integration without significant cleaning or correction.

---

# 3. Feature Engineering and Data Enrichment

To prepare the data for fee calculation, we must create a single, unified view of each transaction that includes all attributes relevant to the fee rules.

### 3.1. Creating a Consolidated Transaction View

The `payments.csv` file was enriched by joining it with other datasets:

1. **Merchant Attributes:** By joining with `merchant_data.json` on the `merchant` name, each transaction was augmented with its merchant's `account_type`, `capture_delay`, and assigned `acquirer`.
2. **Acquirer Country:** The enriched data was then joined with `acquirer_countries.csv` to append the acquirer's country code, which is essential for differentiating between domestic and international transactions [16].
3. **Business Category:** Finally, joins with `merchant_category_codes.csv` provided a descriptive business category for each merchant, although this is not a direct input for the current fee rules, it provides valuable context [17].

### 3.2. Engineering Features for Fee Logic

The `manual.md` and `fees.json` files describe fee rules based on specific conditions. We engineered features to match these conditions directly:

- **`is_intracountry` Flag:** A new boolean column was created by comparing the `issuing_country` and `acquirer_country` for each transaction. This flag directly corresponds to the `intracountry` field in the `fees.json` rules and is the most critical feature for local acquiring analysis. Our analysis found **24,659** intracountry transactions in the dataset [14].
- **Risk Indicators:** We identified columns that serve as proxies for transaction risk, which often correlates with higher fees.
  - **ACI 'G' Transactions:** Transactions where `aci` is 'G' represent a high-risk "Card Not Present - Non-3-D Secure" segment. There are **25,463** such transactions [28].
  - **Fraudulent Disputes:** The `has_fraudulent_dispute` flag marks transactions that led to a dispute. Overall, **7.79%** of transactions fall into this category [11]. Further analysis reveals that transactions with fraudulent disputes have a slightly higher average amount (€93.36) compared to non-disputed ones (€91.82) [a]. The two most common device types for these disputed transactions are 'Other' and 'Windows', which together account for over 10,000 fraudulent disputes [c].

The final prepared dataset contains a single row per transaction with all the necessary attributes from both the transaction itself and its associated merchant.

---

## 4. Analysis of Key Metrics and Optimization Drivers

With the data prepared, we can now quantify key business metrics and identify the most promising areas for cost optimization.

### 4.1. Transaction Volume and Distribution

- **Total Processed Volume:** The total volume of successful (non-refused) transactions is **€11,821,507.89** [8]. This volume is composed of transactions with a very wide range of values. For example, transactions from shoppers in Luxembourg ('LU') alone range from a minimum of **€0.50** to a maximum of **€4811.76** [e].
- **Top Merchant:** `Crossfit_Hanna` is the largest merchant by volume, processing **€5,076,636.90** [10]. A significant portion of this merchant's business, approximately **61.33%**, occurs during typical peak hours (9 AM to 5 PM) [f].
- **Card Scheme Dominance:** `GlobalCard` is the most used card scheme, accounting for **34.83%** of all transactions [9].
- **Channel Distribution:** The vast majority of transactions (**91%**) are `Ecommerce`, with the remaining being POS [12].

### 4.2. Identifying Optimization Opportunities

**a) Local Acquiring for International Transactions**

The most significant optimization opportunity lies in reducing costs associated with international payments.

- **Target Volume: €10,427,322.28** in transaction volume comes from international payments (where issuer and acquirer countries differ) [24]. Processing these locally could dramatically reduce scheme fees.
- **Merchant-Specific Opportunity:** `Crossfit_Hanna`, the largest merchant, is a prime candidate for this strategy, as **78.59%** of its transactions are international [31]. This merchant currently processes transactions through four acquirers: `gringotts`, `the_savings_and_loan_bank`, `bank_of_springfield`, and `dagoberts_vault` [b]. A targeted analysis can determine which of these acquirers offer the best local rates in the countries where Crossfit_Hanna's customers are located.

**b) Mitigating Transaction Downgrades**

Transaction downgrades, often caused by operational factors, lead to higher fees.

- **Capture Delay: 24 merchants** have a `capture_delay` other than 'immediate' [27]. These merchants should be advised to adjust their settlement practices to avoid late settlement penalties.
- **B2B Data:** The `manual.md` mentions leveraging Level 2/3 data for B2B transactions to avoid downgrades. However, an analysis of `payments.csv` confirmed that **no columns correspond to Level 2 or 3 data fields** [32]. This represents a data gap and an opportunity for future data enrichment.

---

## 5. Validating Fee Rule Applicability

The fee structure is highly complex, with multiple rules potentially applying to a single transaction. The prepared data allows us to navigate this complexity.

### 5.1. Understanding the Fee Rule Structure

A transaction fee is calculated using the formula: `fee = fixed_amount + (rate * transaction_value / 10000)` [26]. The `fixed_amount` ranges from 0.0 to 0.14, and the `rate` ranges from 10 to 99 [20].

The applicability of a rule depends on a combination of factors, including:

- Card scheme, ACI, credit/debit status, and intracountry status.
- Rule specificity, where an empty list (e.g., in `account_type`) implies the rule applies to all values [6]. There are **720** such universal rules for account type. For specific segments, such as 'Enterprise - Digital' (Account Type 'D') merchants, there are **113** tailored fee rules, demonstrating the targeted nature of the fee system [d].
- Volume dependencies, with **200** rules having conditions based on `monthly_volume` [7].

### 5.2. A Practical Example of Rule Matching

To illustrate the complexity, consider a transaction with the following properties: `card_scheme = 'GlobalCard'`, `is_credit = True`, `aci = 'B'`.

By filtering the `fees.json` rule set based on the logic in the `manual.md`, we found that **71 potential fee rules** could apply to this single transaction [23]. This is because rules can be generic (e.g., `is_credit` is `null`, or the `aci` list is empty). This finding underscores the necessity of a sophisticated matching algorithm that can prioritize the most specific rule.

---

## 6. Conclusion and Recommendations

This data preparation effort has successfully consolidated and validated multiple datasets, creating a robust analytical foundation for optimizing payment processing fees. The data is of high quality and integrity, and a clear path to significant cost savings has been identified.

Based on our findings, we recommend the following next steps:

1. **Develop a Fee Calculation Engine:** Construct a programmatic function that takes a record from the prepared dataset as input, filters the 1,000 rules in `fees.json` to find the most specific match, and calculates the precise fee. This will be essential for validating current costs and simulating changes.
2. **Simulate Local Acquiring Scenarios:** Use the prepared data to model the financial impact of routing the €10.4M in international volume through local acquirers. This model should be prioritized for merchants like `Crossfit_Hanna` who have a high percentage of international transactions [b].
3. **Address Operational Inefficiencies:** Proactively engage the 24 merchants with non-immediate capture delays to educate them on the financial risks of late settlement and encourage a shift to immediate capture where possible.
4. **Investigate Data Gaps for B2B Optimization:** Initiate a project to determine if Level 2 and Level 3 data can be captured for B2B transactions. This would unlock an additional avenue for fee optimization that is currently inaccessible [32].

## G  In-depth analysis on the data file analyzer agent

### G.1  Excel file in KramaBench

```
f'''
Analyzing Excel File: data/mmc2.xlsx
Found 3 sheet(s): README, A-global-proteomics, B-phospho-proteomics
============================================================

Analyzing Sheet: 'README' (Sheet 1/3)
---------------------------------------
Shape: 9 rows, 2 columns

Column Names:
['Sheet', 'Description']

Data Types and Non-Null Counts (df.info()):
<class 'pandas.core.frame.DataFrame'>
RangeIndex: 9 entries, 0 to 8
Data columns (total 2 columns):
 #   Column       Non-Null Count  Dtype
---  ------       --------------  -----
 0   Sheet        9 non-null      object
 1   Description  9 non-null      object
dtypes: object(2)
memory usage: 272.0+ bytes

First 5 rows (Head):
                 Sheet                                          Description
0   A-global-proteomics  Global proteomics data, median polishing and l...
1  B-phospho-proteomics  Phosphoproteomics site level data, median poli...
2   C-acetyl-proteomics  Acetylproteomics site level data, median polis...
3      D-gene-expression  Linear gene expression RSEM, upper quantile no...
4  E-circRNA-expression  Circular RNA expression RSEM, upper quantile n...

Descriptive Statistics (df.describe(include='all')):
                    Sheet                                      Description
count                   9                                                9
unique                  9                                                9
top     A-global-proteomics  Global proteomics data, median polishing and l...
freq                    1                                                1

Number of Unique Values per Column (df.nunique()):
Sheet          9
Description    9
dtype: int64

-------------------- Next Sheet --------------------

Analyzing Sheet: 'A-global-proteomics' (Sheet 2/3)
---------------------------------------
Shape: 10999 rows, 154 columns

Column Names:
['idx', 'S001', 'S002', 'S003', 'S004', 'S005', 'S006', 'S007', 'S008', 'S009', 'S010', 'S011', 'S012',
↪  'S013', 'S014', 'S015', 'S016', 'S017', 'S018', 'S019', 'S020', 'S021', 'S022', 'S023', 'S024', 'S025',
↪  'S026', 'S027', 'S028', 'S029', 'S030', 'S031', 'S032', 'S033', 'S034', 'S035', 'S036', 'S037', 'S038',
↪  'S039', 'S040', 'S041', 'S042', 'S043', 'S044', 'S045', 'S046', 'S047', 'S048', 'S049', 'S050', 'S051',
↪  'S052', 'S053', 'S054', 'S055', 'S056', 'S057', 'S058', 'S059', 'S060', 'S061', 'S062', 'S063', 'S064',
↪  'S065', 'S066', 'S067', 'S068', 'S069', 'S070', 'S071', 'S072', 'S073', 'S074', 'S075', 'S076', 'S077',
↪  'S078', 'S079', 'S080', 'S081', 'S082', 'S083', 'S084', 'S085', 'S086', 'S087', 'S088', 'S089', 'S090',
↪  'S091', 'S092', 'S093', 'S094', 'S095', 'S096', 'S097', 'S098', 'S099', 'S100', 'S101', 'S102', 'S103',
↪  'S104', 'S105', 'S106', 'S107', 'S108', 'S109', 'S110', 'S111', 'S112', 'S113', 'S114', 'S115', 'S116',
↪  'S117', 'S118', 'S119', 'S120', 'S121', 'S122', 'S123', 'S124', 'S125', 'S126', 'S127', 'S128', 'S129',
↪  'S130', 'S131', 'S132', 'S133', 'S134', 'S135', 'S136', 'S137', 'S138', 'S139', 'S140', 'S141', 'S142',
↪  'S143', 'S144', 'S145', 'S146', 'S147', 'S148', 'S149', 'S150', 'S151', 'S152', 'S153']

Data Types and Non-Null Counts (df.info()):
<class 'pandas.core.frame.DataFrame'>
```

```
RangeIndex: 10999 entries, 0 to 10998
Data columns (total 154 columns):
 #    Column  Dtype
---   ------  -----
 0    idx     object
 1    S001    float64
 2    S002    float64
 3    S003    float64
 4    S004    float64
 5    S005    float64
 6    S006    float64
 7    S007    float64
 8    S008    float64
 9    S009    float64
 10   S010    float64
 11   S011    float64
 12   S012    float64
 13   S013    float64
 14   S014    float64
 15   S015    float64
 16   S016    float64
 17   S017    float64
 18   S018    float64
 19   S019    float64
 20   S020    float64
 21   S021    float64
 22   S022    float64
 23   S023    float64
 24   S024    float64
 25   S025    float64
 26   S026    float64
 27   S027    float64
 28   S028    float64
 29   S029    float64
 30   S030    float64
 31   S031    float64
 32   S032    float64
 33   S033    float64
 34   S034    float64
 35   S035    float64
 36   S036    float64
 37   S037    float64
 38   S038    float64
 39   S039    float64
 40   S040    float64
 41   S041    float64
 42   S042    float64
 43   S043    float64
 44   S044    float64
 45   S045    float64
 46   S046    float64
 47   S047    float64
 48   S048    float64
 49   S049    float64
 50   S050    float64
 51   S051    float64
 52   S052    float64
 53   S053    float64
 54   S054    float64
 55   S055    float64
 56   S056    float64
 57   S057    float64
 58   S058    float64
 59   S059    float64
 60   S060    float64
 61   S061    float64
 62   S062    float64
 63   S063    float64
 64   S064    float64
 65   S065    float64
```

```
66    S066    float64
67    S067    float64
68    S068    float64
69    S069    float64
70    S070    float64
71    S071    float64
72    S072    float64
73    S073    float64
74    S074    float64
75    S075    float64
76    S076    float64
77    S077    float64
78    S078    float64
79    S079    float64
80    S080    float64
81    S081    float64
82    S082    float64
83    S083    float64
84    S084    float64
85    S085    float64
86    S086    float64
87    S087    float64
88    S088    float64
89    S089    float64
90    S090    float64
91    S091    float64
92    S092    float64
93    S093    float64
94    S094    float64
95    S095    float64
96    S096    float64
97    S097    float64
98    S098    float64
99    S099    float64
100   S100    float64
101   S101    float64
102   S102    float64
103   S103    float64
104   S104    float64
105   S105    float64
106   S106    float64
107   S107    float64
108   S108    float64
109   S109    float64
110   S110    float64
111   S111    float64
112   S112    float64
113   S113    float64
114   S114    float64
115   S115    float64
116   S116    float64
117   S117    float64
118   S118    float64
119   S119    float64
120   S120    float64
121   S121    float64
122   S122    float64
123   S123    float64
124   S124    float64
125   S125    float64
126   S126    float64
127   S127    float64
128   S128    float64
129   S129    float64
130   S130    float64
131   S131    float64
132   S132    float64
133   S133    float64
134   S134    float64
135   S135    float64
```

```
136  S136    float64
137  S137    float64
138  S138    float64
139  S139    float64
140  S140    float64
141  S141    float64
142  S142    float64
143  S143    float64
144  S144    float64
145  S145    float64
146  S146    float64
147  S147    float64
148  S148    float64
149  S149    float64
150  S150    float64
151  S151    float64
152  S152    float64
153  S153    float64
dtypes: float64(153), object(1)
memory usage: 12.9+ MB

First 5 rows (Head):
       idx   S001   S002   S003   S004  ...   S149   S150    S151    S152    S153
0     A1BG -1.180 -0.685 -0.528  2.350  ...  0.650  0.458  1.1500   0.547  0.9400
1      A2M -0.863 -1.070 -1.320  2.820  ...  0.227  0.520  1.4600   1.270  0.9040
2    A2ML1 -0.802 -0.684  0.435 -1.470  ...  1.930 -0.291 -0.0229  -0.197 -0.0803
3   A4GALT  0.222  0.984    NaN    NaN  ...    NaN    NaN     NaN     NaN     NaN
4     AAAS  0.256  0.135 -0.240  0.154  ...  0.239  0.477  0.2520   0.405  0.2990

[5 rows x 154 columns]

Last 3 rows (Tail):
         idx    S001    S002   S003  ...    S150     S151   S152    S153
10996    ZYX -1.0200 -1.1300 -0.540  ... -0.3990  0.83500  0.416 -0.4220
10997  ZZEF1 -0.1230 -0.0757  0.320  ... -0.0959  0.16900  0.273 -0.0931
10998   ZZZ3 -0.0859 -0.4730 -0.419  ... -0.0635 -0.00809 -0.658  0.0557

[3 rows x 154 columns]

Descriptive Statistics (df.describe(include='all')):
          idx         S001         S002  ...         S151         S152         S153
count   10999  9689.000000  9943.000000  ...  9646.000000  9646.000000  9648.000000
unique  10999          NaN          NaN  ...          NaN          NaN          NaN
top      ZZZ3          NaN          NaN  ...          NaN          NaN          NaN
freq        1          NaN          NaN  ...          NaN          NaN          NaN
mean      NaN    -0.021396    -0.036661  ...     0.079756     0.035783    -0.006398
std       NaN     0.611966     0.678262  ...     0.542243     0.660306     0.528082
min       NaN    -3.550000    -8.190000  ...    -3.220000    -7.050000    -4.190000
25%       NaN    -0.340000    -0.381000  ...    -0.187000    -0.249750    -0.216000
50%       NaN    -0.018600    -0.008360  ...     0.013300     0.007760     0.005690
75%       NaN     0.309000     0.311000  ...     0.272000     0.305000     0.225250
max       NaN     3.570000     5.290000  ...     8.340000     7.490000     2.970000

[11 rows x 154 columns]

Number of Unique Values per Column (df.nunique()):
idx      10999
S001      3088
S002      3207
S003      3146
S004      3062
         ...
S149      3213
S150      3255
S151      3201
S152      3232
S153      3284
Length: 154, dtype: int64

-------------------- Next Sheet --------------------
```

```
Analyzing Sheet: 'B-phospho-proteomics' (Sheet 3/3)
----------------------------------------
Shape: 73212 rows, 154 columns

Column Names:
['idx', 'S001', 'S002', 'S003', 'S004', 'S005', 'S006', 'S007', 'S008', 'S009', 'S010', 'S011', 'S012',
↪  'S013', 'S014', 'S015', 'S016', 'S017', 'S018', 'S019', 'S020', 'S021', 'S022', 'S023', 'S024', 'S025',
↪  'S026', 'S027', 'S028', 'S029', 'S030', 'S031', 'S032', 'S033', 'S034', 'S035', 'S036', 'S037', 'S038',
↪  'S039', 'S040', 'S041', 'S042', 'S043', 'S044', 'S045', 'S046', 'S047', 'S048', 'S049', 'S050', 'S051',
↪  'S052', 'S053', 'S054', 'S055', 'S056', 'S057', 'S058', 'S059', 'S060', 'S061', 'S062', 'S063', 'S064',
↪  'S065', 'S066', 'S067', 'S068', 'S069', 'S070', 'S071', 'S072', 'S073', 'S074', 'S075', 'S076', 'S077',
↪  'S078', 'S079', 'S080', 'S081', 'S082', 'S083', 'S084', 'S085', 'S086', 'S087', 'S088', 'S089', 'S090',
↪  'S091', 'S092', 'S093', 'S094', 'S095', 'S096', 'S097', 'S098', 'S099', 'S100', 'S101', 'S102', 'S103',
↪  'S104', 'S105', 'S106', 'S107', 'S108', 'S109', 'S110', 'S111', 'S112', 'S113', 'S114', 'S115', 'S116',
↪  'S117', 'S118', 'S119', 'S120', 'S121', 'S122', 'S123', 'S124', 'S125', 'S126', 'S127', 'S128', 'S129',
↪  'S130', 'S131', 'S132', 'S133', 'S134', 'S135', 'S136', 'S137', 'S138', 'S139', 'S140', 'S141', 'S142',
↪  'S143', 'S144', 'S145', 'S146', 'S147', 'S148', 'S149', 'S150', 'S151', 'S152', 'S153']

Data Types and Non-Null Counts (df.info()):
<class 'pandas.core.frame.DataFrame'>
RangeIndex: 73212 entries, 0 to 73211
Data columns (total 154 columns):
 #     Column   Dtype
---    ------   -----
 0     idx      object
 1     S001     float64
 2     S002     float64
 3     S003     float64
 4     S004     float64
 5     S005     float64
 6     S006     float64
 7     S007     float64
 8     S008     float64
 9     S009     float64
 10    S010     float64
 11    S011     float64
 12    S012     float64
 13    S013     float64
 14    S014     float64
 15    S015     float64
 16    S016     float64
 17    S017     float64
 18    S018     float64
 19    S019     float64
 20    S020     float64
 21    S021     float64
 22    S022     float64
 23    S023     float64
 24    S024     float64
 25    S025     float64
 26    S026     float64
 27    S027     float64
 28    S028     float64
 29    S029     float64
 30    S030     float64
 31    S031     float64
 32    S032     float64
 33    S033     float64
 34    S034     float64
 35    S035     float64
 36    S036     float64
 37    S037     float64
 38    S038     float64
 39    S039     float64
 40    S040     float64
 41    S041     float64
 42    S042     float64
```

```
 43    S043    float64
 44    S044    float64
 45    S045    float64
 46    S046    float64
 47    S047    float64
 48    S048    float64
 49    S049    float64
 50    S050    float64
 51    S051    float64
 52    S052    float64
 53    S053    float64
 54    S054    float64
 55    S055    float64
 56    S056    float64
 57    S057    float64
 58    S058    float64
 59    S059    float64
 60    S060    float64
 61    S061    float64
 62    S062    float64
 63    S063    float64
 64    S064    float64
 65    S065    float64
 66    S066    float64
 67    S067    float64
 68    S068    float64
 69    S069    float64
 70    S070    float64
 71    S071    float64
 72    S072    float64
 73    S073    float64
 74    S074    float64
 75    S075    float64
 76    S076    float64
 77    S077    float64
 78    S078    float64
 79    S079    float64
 80    S080    float64
 81    S081    float64
 82    S082    float64
 83    S083    float64
 84    S084    float64
 85    S085    float64
 86    S086    float64
 87    S087    float64
 88    S088    float64
 89    S089    float64
 90    S090    float64
 91    S091    float64
 92    S092    float64
 93    S093    float64
 94    S094    float64
 95    S095    float64
 96    S096    float64
 97    S097    float64
 98    S098    float64
 99    S099    float64
100    S100    float64
101    S101    float64
102    S102    float64
103    S103    float64
104    S104    float64
105    S105    float64
106    S106    float64
107    S107    float64
108    S108    float64
109    S109    float64
110    S110    float64
111    S111    float64
112    S112    float64
```

```
113  S113    float64
114  S114    float64
115  S115    float64
116  S116    float64
117  S117    float64
118  S118    float64
119  S119    float64
120  S120    float64
121  S121    float64
122  S122    float64
123  S123    float64
124  S124    float64
125  S125    float64
126  S126    float64
127  S127    float64
128  S128    float64
129  S129    float64
130  S130    float64
131  S131    float64
132  S132    float64
133  S133    float64
134  S134    float64
135  S135    float64
136  S136    float64
137  S137    float64
138  S138    float64
139  S139    float64
140  S140    float64
141  S141    float64
142  S142    float64
143  S143    float64
144  S144    float64
145  S145    float64
146  S146    float64
147  S147    float64
148  S148    float64
149  S149    float64
150  S150    float64
151  S151    float64
152  S152    float64
153  S153    float64
dtypes: float64(153), object(1)
memory usage: 86.0+ MB

First 5 rows (Head):
        idx   S001    S002   S003  S004  ...    S149   S150    S151   S152    S153
0  AAAS-S495    NaN    NaN -0.202  0.25  ... -0.272  0.223 -0.3940  0.149  0.0774
1  AAAS-S541    NaN    NaN    NaN   NaN  ... -0.191 -0.517 -0.0108  0.256 -0.1580
2  AAAS-Y485    NaN    NaN    NaN   NaN  ...    NaN    NaN     NaN    NaN     NaN
3  AACS-S618 -0.881    NaN    NaN   NaN  ...    NaN    NaN     NaN    NaN     NaN
4  AAED1-S12 -1.810  0.084 -1.880   NaN  ...  0.631  0.522  1.0600  0.951 -0.3430

[5 rows x 154 columns]

Last 3 rows (Tail):
             idx   S001    S002  S003  S004  ... S149   S150   S151   S152    S153
73209  ZZZ3-T415    NaN     NaN   NaN   NaN  ...  NaN    NaN    NaN    NaN     NaN
73210  ZZZ3-T418    NaN  0.1605   NaN   NaN  ...  NaN    NaN    NaN    NaN     NaN
73211  ZZZ3-Y399    NaN -0.0635   NaN   NaN  ...  0.0  0.179 -0.122 -0.354  0.0216

[3 rows x 154 columns]

Descriptive Statistics (df.describe(include='all')):
           idx          S001  ...          S152          S153
count    73212  31671.000000  ...  34285.000000  34303.000000
unique   73212           NaN  ...           NaN           NaN
top  ZZZ3-Y399           NaN  ...           NaN           NaN
freq         1           NaN  ...           NaN           NaN
mean       NaN     -0.065974  ...      0.005505      0.001946
std        NaN      0.641707  ...      0.674115      0.543703
```

```
min           NaN    -7.020000  ...    -5.950000    -5.060000
25%           NaN    -0.399000  ...    -0.328000    -0.253000
50%           NaN    -0.030100  ...    -0.001340     0.000000
75%           NaN     0.292000  ...     0.331500     0.261500
max           NaN     4.320000  ...     6.520000     3.940000

[11 rows x 154 columns]

Number of Unique Values per Column (df.nunique()):
idx    73212
S001    7154
S002    7644
S003    5834
S004    6596
        ...
S149    7474
S150    7612
S151    7327
S152    7393
S153    7501
Length: 154, dtype: int64

=========================================================
Excel file 'data/mmc2.xlsx' analysis complete.
'''
```

Listing 4: Description of '1-s2.0-S0092867420301070-mmc2.xlsx' in KramaBench.

DS-STAR's analyzer agent is sufficiently robust to summarize highly complex data. For example, our analyzer agent successfully summarizes the 1-s2.0-S0092867420301070-mmc2.xlsx from the biomedical domain of KramaBench (see above). This file is particularly complex as it contains three separate sheets within one Excel file. As shown in above, our agent successfully reads and analyzes each sheet one-by-one, which shows its robustness and capability in handling multi-sheet, complex data structures.

## G.2    Manually created Excel file

| | A | B | C | D | E | F | G | H | I | J | K |
|---|---|---|---|---|---|---|---|---|---|---|---|
| 1 | | exchange | ticker | name | | | | | | | |
| 2 | | nasdaq | PYPL | PayPal Holdings | | | | | | | |
| 3 | | nasdaq | AMZN | Amazon.com Inc | | | | | | | |
| 4 | | nasdaq | MSFT | Microsoft Corp. | | | | | | | |
| 5 | | nyse | IBM | International Business Machines Corporation | | | | | role | age | sex |
| 6 | | nasdaq | GOOGL | Alphabet | | | | | passenger | 24 | female |
| 7 | | | | | | | | | passenger | 31 | male |
| 8 | | | | | | | | | driver | 72 | female |
| 9 | | | | | | | | | driver | 66 | male |
| 10 | | | | | id | country_id | name | | passenger | 65 | male |
| 11 | | | | | 1 | Belgium Jupiler l | Belgium | | | | |
| 12 | | | | | 1729 | England Premier | England | | | | |
| 13 | | | | | 4769 | France Ligue 1 | France | | | | |
| 14 | | | | | 7809 | Germany 1. Bun | Germany | | | | |
| 15 | | | | | 10257 | Italy Serie A | Italy | | | | |
| 16 | | | | | | | | | | | |

Figure 8: **A complex Excel file.** DS-STAR's analyzer agent is is sufficiently robust to summarize even highly complex data.

```
f'''
=========================================================
  Excel File Analysis: data/dummy.xlsx
=========================================================

Found 1 sheet(s): ['Sheet1']

-------------------------------------------------
  Analyzing Sheet: 'Sheet1'
```

```
--------------------------------------------------
- Sheet Dimensions: 15 rows, 11 columns

>>> Found 3 distinct data block(s).

--- Block 1 (Bounding Box: B1:D6) ---
    * Type: Structured Table
    * Shape: 5 rows, 3 columns
    * Column Names: ['exchange', 'ticker', 'name']
    * Data Types:
      - exchange: object
      - ticker: object
      - name: object
    * Data Preview (first 5 rows):
exchange ticker                                    name
  nasdaq   PYPL                         PayPal Holdings
  nasdaq   AMZN                          Amazon.com Inc
  nasdaq   MSFT                          Microsoft Corp.
    nyse    IBM International Business Machines Corporation
  nasdaq  GOOGL                                 Alphabet

--- Block 2 (Bounding Box: I5:K10) ---
    * Type: Structured Table
    * Shape: 5 rows, 3 columns
    * Column Names: ['role', 'age', 'sex']
    * Data Types:
      - role: object
      - age: float64
      - sex: object
    * Data Preview (first 5 rows):
     role  age     sex
passenger 24.0 female
passenger 31.0   male
   driver 72.0 female
   driver 66.0   male
passenger 65.0   male

--- Block 3 (Bounding Box: E10:G15) ---
    * Type: Structured Table
    * Shape: 5 rows, 3 columns
    * Column Names: ['id', 'country_id', 'name']
    * Data Types:
      - id: float64
      - country_id: object
      - name: object
    * Data Preview (first 5 rows):
   id             country_id     name
  1.0 Belgium Jupiler League Belgium
 1729.0 England Premier League England
 4769.0         France Ligue 1  France
 7809.0  Germany 1. Bundesliga Germany
10257.0          Italy Serie A   Italy

==========================================================
  End of Analysis
==========================================================
'''
```

Listing 5: Description of 'dummy.xlsx'.

Moreover, we additionally created a complex Excel file where three tables of different shapes are placed in random positions in one sheet (see Figure 8). This data specifically requires DS-STAR to exactly identify the bounding boxes of each table. As demonstrated above, DS-STAR's analyzer agent successfully handles such complex data layouts, which clearly shows its robustness.

# H   Qualitative results on the data file analyzer agent

## H.1   Code generated by the analyzer agent

```python
import json

def describe_json_data(file_path):
    """
    Loads a JSON file, prints its full content, and then describes its structure
    and essential information, including column names (keys) and example data.
    """
    # Load the JSON data from the file
    # Using utf-8 encoding is a good practice for JSON.
    with open(file_path, 'r', encoding='utf-8') as f:
        data = json.load(f)

    print(f"--- Describing content of: {file_path} ---")
    print("\n")

    # 1. Print the full loaded content as per requirement
    # "The Python code should print out the content of data/merchant_data.json."
    print("--- Full Loaded Content ---")
    # Using json.dumps for a pretty-printed representation of the loaded Python object.
    # ensure_ascii=False allows for proper display of unicode characters.
    print(json.dumps(data, indent=4, ensure_ascii=False))
    print("-" * 50)
    print("\n")

    # 2. Print Essential Information
    print("--- Essential Information ---")
    data_type = type(data)
    print(f"Overall Data Type: {data_type.__name__}")

    if isinstance(data, dict):
        print("The data is a single JSON object (dictionary).")
        keys = list(data.keys())
        print(f"Number of top-level keys: {len(keys)}")

        if not keys:
            print("The dictionary is empty.")
        else:
            # Requirement: "print out all the column names."
            print("Top-level keys (can be considered as 'column names' for the main object):")
            for key in keys:
                print(f"  - {key}")

            # Requirement: "If there are too many structured data, print out just few examples."
            print("\n--- Example Key-Value Pairs (first 5 if more exist) ---")
            count = 0
            for key, value in data.items():
                if count < 5:
                    value_str = str(value)
                    # Truncate long values for a concise preview
                    preview = (value_str[:70] + '...') if len(value_str) > 70 else value_str
                    print(f"  {key}: {preview}")
                    count += 1
                else:
                    print("  ... (more key-value pairs exist)")
                    break
            if count == 0 and not keys: # Should be caught by 'if not keys:' earlier
                print("  (Dictionary is empty - no key-value pairs to show)")

    elif isinstance(data, list):
        print("The data is a JSON array (list).")
        num_items = len(data)
        print(f"Number of items in the list: {num_items}")

        if num_items == 0:
            print("The list is empty.")
        else:
            first_item = data[0]
            print(f"Type of the first item in the list: {type(first_item).__name__}")

            # Check if it's a list of dictionaries (common for structured data records)
            if isinstance(first_item, dict):
                print("The list appears to contain JSON objects (dictionaries), suggesting structured data.")

                # Requirement: "print out all the column names."
                # For a list of objects, column names are typically the keys of these objects.
                # We'll use the keys from the first object as representative column names.
                first_item_keys = list(first_item.keys())
                if not first_item_keys:
                    print("The first object in the list is an empty dictionary (no column names to infer).")
                else:
                    print("Keys of the first object (assumed to be common 'column names' for the items):")
                    for key in first_item_keys:
                        print(f"  - {key}")

                # Requirement: "If there are too many structured data, print out just few examples."
                num_examples_to_show = min(3, num_items) # Show up to 3 examples
                print(f"\n--- Example Items (first {num_examples_to_show} of {num_items}) ---")
                for i in range(num_examples_to_show):
                    print(f"Item {i+1}:")
                    # Pretty print each example item. Truncate if an individual item is very large.
                    example_item_str = json.dumps(data[i], indent=2, ensure_ascii=False)
                    lines = example_item_str.split('\n')
                    if len(lines) > 15: # Arbitrary limit: max 15 lines per example item summary
                        print('\n'.join(lines[:15]))
                        print("    ... (item content truncated for brevity in this summary)")
                    else:
                        print(example_item_str)

                if num_items > num_examples_to_show:
                    print(f"\n... and {num_items - num_examples_to_show} more items in the list.")

            # Else, it's a list of other types (e.g., strings, numbers - unstructured or semi-structured)
            else:
                print("The list contains non-object items (e.g., strings, numbers, booleans, or mixed types).")
                print("This is often considered unstructured or semi-structured data.")
                num_examples_to_show = min(5, num_items) # Show up to 5 examples
                print(f"\n--- Example Items (first {num_examples_to_show} of {num_items}) ---")
```

```
                for i in range(num_examples_to_show):
                    item_str = str(data[i])
                    # Truncate long string representations for a concise preview
                    preview = (item_str[:70] + '...') if len(item_str) > 70 else item_str
                    print(f"Item {i+1}: {preview}")

                if num_items > num_examples_to_show:
                    print(f"\n... and {num_items - num_examples_to_show} more items in the list.")

        elif isinstance(data, (str, int, float, bool)) or data is None:
            # For simple scalar types, the "Full Loaded Content" print is the primary information.
            # This section just confirms the nature of the data.
            print(f"The data is a simple JSON scalar value (not an object or array).")
            if data is None:
                print("Specifically, the JSON content is 'null'.")
            # The actual value was already printed in the "Full Loaded Content" section.

        else:
            # This case should ideally not be reached if json.load() was successful
            # and returned a standard Python type corresponding to a valid JSON root element.
            print("The data type is unusual for a root JSON element (e.g., a custom object if json.load was hooked).")
            # The actual value was already printed in the "Full Loaded Content" section.

    print("\n" + "-" * 50)
    print("--- End of Description ---")

if __name__ == "__main__":
    # The path to the JSON file.
    # This script assumes 'data/merchant_data.json' exists relative to where the script is run.
    # For example, if the script is in /home/user/scripts/, it expects /home/user/scripts/data/merchant_data.json
    json_file_path = 'data/merchant_data.json'

    # Call the function to load and describe the JSON data.
    # As per requirements, no try-except blocks are used for error handling.
    # If the file doesn't exist, a FileNotFoundError will be raised.
    # If the file's content is not valid JSON, a json.JSONDecodeError will be raised.
    # These errors will halt the program, and debugging can proceed from there.
    describe_json_data(json_file_path)
```

Listing 6: Python code for generating the description of 'merchant_data.json' in DABStep.

## H.2    Examples of generated data descriptions

```
f'''
File: data/merchant_data.json (6857 bytes)
Top-level type: array
root: array with 30 items
Element types: object=30
Columns (5): ['account_type', 'acquirer', 'capture_delay', 'merchant', 'merchant_category_code']
Sample rows (first 5):
[
  {
    "merchant": "Crossfit_Hanna",
    "capture_delay": "manual",
    "acquirer": [
      "gringotts",
      "the_savings_and_loan_bank",
      "bank_of_springfield",
      "dagoberts_vault"
    ],
    "merchant_category_code": 7997,
    "account_type": "F"
  },
  {
    "merchant": "Martinis_Fine_Steakhouse",
    "capture_delay": "immediate",
    "acquirer": [
      "dagoberts_geldpakhuis",
      "bank_of_springfield"
    ],
    "merchant_category_code": 5812,
    "account_type": "H"
  },
  {
    "merchant": "Belles_cookbook_store",
    "capture_delay": "1",
    "acquirer": [
      "lehman_brothers"
    ],
    "merchant_category_code": 5942,
    "account_type": "R"
  },
  {
```

```
      "merchant": "Golfclub_Baron_Friso",
      "capture_delay": "2",
      "acquirer": [
        "medici"
      ],
      "merchant_category_code": 7993,
      "account_type": "F"
    },
    {
      "merchant": "Rafa_AI",
      "capture_delay": "7",
      "acquirer": [
        "tellsons_bank"
      ],
      "merchant_category_code": 7372,
      "account_type": "D"
    }
  ]
  '''
```

Listing 7: Description of 'merchant_data.json' in DABstep.

# I   Qualitative results for each task type

This section provides qualitative results. It includes queries on DA-Code benchmark, solution code generated by DS-STAR, output generated by DS-STAR, and ground-truth answer. Specifically, it presents examples for each task, such as statistical analysis, data insights, data wrangling, machine learning, and visualization tasks.

## I.1   Statistical analysis

```
f'''Could you analyze our Business Case dataset to perform a hypothesis test comparing bike rentals across
↪    different weather conditions? I need the results formatted and saved in a file named 'weather.csv'
↪    following the requirement in tips.md.'''
```

Listing 8: **User's question.**

```python
import pandas as pd
from scipy.stats import kruskal
import os

# Define the file path for the input dataset
input_file_path = 'data/yulu_bike_sharing_dataset.csv'

# Load the dataset into a pandas DataFrame
yulu_df = pd.read_csv(input_file_path)

# The goal is to perform a Kruskal-Wallis H-test to compare the distribution of bike rentals ('count')
# across different weather conditions ('weather').

# The 'weather' column contains numerical codes for different weather conditions.
# First, we need to find the unique weather conditions in the dataset.
unique_weather_conditions = sorted(yulu_df['weather'].unique())

# Next, we create a list of data samples, where each sample is an array of 'count'
# values corresponding to a specific weather condition.
weather_groups = [yulu_df['count'][yulu_df['weather'] == weather] for weather in unique_weather_conditions]

# Perform the Kruskal-Wallis H-test. The * operator unpacks the list of groups
# into separate arguments for the function.
statistic, p_value = kruskal(*weather_groups)

# Define the significance level (alpha) as specified in the guidelines.
alpha = 0.05

# Based on the p-value, determine the hypothesis conclusion and the comment
# as per the format requirements in 'data/tips.md'.
if p_value < alpha:
    hypothesis_conclusion = "Strong Evidence to Reject Null Hypothesis"
    comment_on_results = "Different distributions"
else:
    hypothesis_conclusion = "No Evidence to Reject Null Hypothesis"
    comment_on_results = "No evidence of different distributions"

# Create a dictionary containing the results in the required format.
results_data = {
    'Test Type': ["Non Parametric"],
    'Kruskal Statistic': [statistic],
    'P-Value': [p_value],
    'Hypothesis': [hypothesis_conclusion],
    'Comment': [comment_on_results]
}

# Create a new pandas DataFrame from the results dictionary.
weather_results_df = pd.DataFrame(results_data)

# Define the output directory and file path.
output_dir = 'final'
output_file_path = os.path.join(output_dir, 'weather.csv')

# Ensure the output directory exists.
os.makedirs(output_dir, exist_ok=True)

# Save the resulting DataFrame to the specified CSV file,
# ensuring the index is not written to the file.
weather_results_df.to_csv(output_file_path, index=False)

# print(f"Analysis complete. Results saved to {output_file_path}")
# print(weather_results_df.to_string())
```

Listing 9: **Solution code.**

Table 10: **Generated output** by DS-STAR in csv format.

| Test Type | Kruskal Statistic | P-Value | Hypothesis | Comment |
|---|---|---|---|---|
| Non Parametric | 205.00216514479087 | 3.501611300708679e-44 | Strong Evidence to Reject Null Hypothesis | Different distributions |

## I.2 Data insight

```
f'''Could you please analyze the matches between Argentina and Brazil? Record the total number of matches,
↪   the number of wins for each team, and the number of draws. Indicate if Brazil has won more matches
↪   ("Yes" or "No"). Present your findings in the following format:

"""
{
    "Total Matches between Argentina and Brazil": [...],
    "Arg Wins": [...],
    "Brazil Wins": [...],
    "Draws": [...],
    "Result": [...]
}
"""'''
```

Listing 10: **User's question.**

```python
import pandas as pd
import numpy as np
import json
import os

# Ensure the 'final' directory exists for the output file
output_dir = 'final'
os.makedirs(output_dir, exist_ok=True)

# Load the results.csv file into a pandas DataFrame
results_df = pd.read_csv('data/results.csv')

# Filter for matches where the home team is Argentina and the away team is Brazil
arg_vs_bra = results_df[
    (results_df['home_team'] == 'Argentina') & (results_df['away_team'] == 'Brazil')
]

# Filter for matches where the home team is Brazil and the away team is Argentina
bra_vs_arg = results_df[
    (results_df['home_team'] == 'Brazil') & (results_df['away_team'] == 'Argentina')
]

# Concatenate the two DataFrames to get all matches between Argentina and Brazil
argentina_brazil_matches = pd.concat([arg_vs_bra, bra_vs_arg])

# Sort the combined DataFrame by date for a chronological view
argentina_brazil_matches = argentina_brazil_matches.sort_values(by='date').copy()

# Define the conditions for determining the winner based on scores
# A draw in regular/extra time is considered a draw, regardless of a penalty shootout outcome.
conditions = [
    argentina_brazil_matches['home_score'] > argentina_brazil_matches['away_score'],
    argentina_brazil_matches['away_score'] > argentina_brazil_matches['home_score']
]

# Define the corresponding outcomes (the winning team's name)
choices = [
    argentina_brazil_matches['home_team'],
    argentina_brazil_matches['away_team']
]

# Create the 'winner' column using np.select, with 'Draw' as the default
argentina_brazil_matches['winner'] = np.select(conditions, choices, default='Draw')

# Count the total wins for each team and the number of draws
win_counts = argentina_brazil_matches['winner'].value_counts()

# Calculate the total number of matches, wins for each team, and draws from the win_counts Series
total_matches = int(win_counts.sum())
argentina_wins = int(win_counts.get('Argentina', 0))
brazil_wins = int(win_counts.get('Brazil', 0))
draws = int(win_counts.get('Draw', 0))

# Determine the 'Result' value. 'Yes' if Brazil has more wins, otherwise 'No'.
result_value = 'Yes' if brazil_wins > argentina_wins else 'No'

# Construct the dictionary to store the analysis results with the specified keys
analysis_results = {
    "Total Matches between Argentina and Brazil": total_matches,
    "Arg Wins": argentina_wins,
    "Brazil Wins": brazil_wins,
    "Draws": draws,
    "Result": result_value
}

# Define the output file path
output_path = os.path.join(output_dir, 'result.json')

# Save the final results dictionary to a JSON file in the 'final' directory
with open(output_path, 'w') as f:
    json.dump(analysis_results, f, indent=4)
```

Listing 11: **Solution code.**

```
f'''{
    "Total Matches between Argentina and Brazil": [109],
    "Arg Wins": [40],
    "Brazil Wins": [43],
    "Draws": [26],
    "Result": ["Yes"]
}'''
```

Listing 12: **Generated output** by DS-STAR, in json format.

## I.3 Data manipulation

```
f'''Calculate the total quantity sold and total sales revenue for each bike category from 2016 to 2018.
↪   Ensure that total sales revenue accounts for any discounts applied to each order. Write the results into
↪   result.csv, following the exact structure and formatting of sample_result.csv.'''
```

Listing 13: **User's question.**

```python
import pandas as pd
import os

# Ensure the output directory exists
os.makedirs('final', exist_ok=True)

# Load the required data files into pandas DataFrames
order_items_df = pd.read_csv('data/order_items.csv')
products_df = pd.read_csv('data/products.csv')
categories_df = pd.read_csv('data/categories.csv')
sample_result_df = pd.read_csv('data/sample_result.csv')

# Calculate the sales revenue for each item, accounting for discounts.
# sales_revenue = quantity * list_price * (1 - discount)
order_items_df['sales_revenue'] = order_items_df['quantity'] * order_items_df['list_price'] * (1 - order_items_df['discount'])

# Merge the order items with product information to get model_year and category_id
merged_df = pd.merge(order_items_df, products_df, on='product_id')

# Merge with category information to get category_name
merged_df = pd.merge(merged_df, categories_df, on='category_id')

# Filter the DataFrame to include only products with model years from 2016 to 2018
filtered_df = merged_df[(merged_df['model_year'] >= 2016) & (merged_df['model_year'] <= 2018)]

# Group the filtered data by 'category_name' and aggregate the total quantity and total sales revenue.
category_summary = filtered_df.groupby('category_name').agg(
    total_quantity=('quantity', 'sum'),
    total_price=('sales_revenue', 'sum')
).reset_index()

# Use the sample_result.csv to define the structure and order of the final output.
# Perform a left merge to ensure all categories from the sample are present in the final result, in the correct order.
final_df = pd.merge(sample_result_df[['category_name']], category_summary, on='category_name', how='left')

# Fill any potential NaN values with 0 for categories that had no sales in the filtered period.
final_df[['total_quantity', 'total_price']] = final_df[['total_quantity', 'total_price']].fillna(0)

# Ensure the column order matches the sample_result.csv file: ['total_quantity', 'total_price', 'category_name']
final_df = final_df[['total_quantity', 'total_price', 'category_name']]

# Convert the 'total_quantity' column to integer type as it represents a count of items.
final_df['total_quantity'] = final_df['total_quantity'].astype(int)

# Define the output path
output_path = 'final/result.csv'

# Save the final DataFrame to the specified path, without the index.
final_df.to_csv(output_path, index=False)

print(f"Successfully generated '{output_path}'")
print("Content of the generated file:")
print(final_df)
```

Listing 14: **Solution code.**

## I.4 Data wrangling

```
f'''Manage missing data in the NYC_Open_Data_Parking_Violations dataset by either dropping or replacing it.
↪   Drop rows if necessary, and drop columns only if most entries are empty. Replace missing values with the
↪   mean, mode, or other appropriate methods. Save the cleaned data to 'cleaned_data.csv'.'''
```

Listing 15: **User's question.**

Table 11: **Generated output** by DS-STAR in csv format.

| total_quantity | total_price | category_name |
|---|---|---|
| 1755 | 2715079.53 | Mountain Bikes |
| 559 | 1665098.49 | Road Bikes |
| 2063 | 995032.62 | Cruisers Bicycles |
| 315 | 916684.78 | Electric Bikes |
| 394 | 711011.84 | Cyclocross Bicycles |
| 813 | 394020.10 | Comfort Bicycles |
| 1179 | 292189.20 | Children Bicycles |

```python
import pandas as pd
import numpy as np
import os

# The user prompt mentions cleaning the 'NYC_Open_Data_Parking_Violations' dataset,
# however, the provided data file is 'data/data.csv', which contains automobile data.
# This script will proceed with cleaning the provided 'data/data.csv' file based on the
# general data cleaning guidelines in the prompt.

# Define file paths
input_file = 'data/data.csv'
output_dir = 'final'
output_file = os.path.join(output_dir, 'cleaned_data.csv')

# Create the output directory if it does not exist
if not os.path.exists(output_dir):
    os.makedirs(output_dir)

# Load the data, interpreting '?' as Not a Number (NaN)
df = pd.read_csv(input_file, na_values='?')

# --- Data Cleaning Steps ---

# Guideline: "Drop rows if necessary"
# The 'price' column is crucial for most analyses. Rows with a missing price are dropped.
df.dropna(subset=['price'], inplace=True)

# Guideline: "Replace missing values with the mean, mode, or other appropriate methods"

# Identify columns for mean and mode imputation
# Numerical columns with missing values to be imputed with the mean
numeric_cols_to_impute_mean = ['normalized-losses', 'bore', 'stroke', 'horsepower', 'peak-rpm']

# Categorical columns with missing values to be imputed with the mode
categorical_cols_to_impute_mode = ['num-of-doors']

# Impute numerical columns with the mean
for col in numeric_cols_to_impute_mean:
    # Convert column to numeric type. This is necessary because the '?'
    # values caused pandas to read them as 'object' type.
    df[col] = pd.to_numeric(df[col])
    # Calculate the mean of the column (NaNs are automatically excluded)
    mean_val = df[col].mean()
    # Fill the missing (NaN) values with the calculated mean
    df[col].fillna(mean_val, inplace=True)

# Impute categorical columns with the mode
for col in categorical_cols_to_impute_mode:
    # Calculate the mode (most frequent value) of the column.
    # .mode() returns a Series, so we take the first element.
    mode_val = df[col].mode()[0]
    # Fill the missing (NaN) values with the calculated mode
    df[col].fillna(mode_val, inplace=True)

# Guideline: After handling missing values, ensure data types are correct.
# The 'price' column was loaded as 'object' and needs to be converted to a numeric type.
df['price'] = df['price'].astype(float)

# --- Save the Cleaned Data ---

# Save the final cleaned DataFrame to 'final/cleaned_data.csv'
# The index is not saved as it's not part of the original data.
df.to_csv(output_file, index=False)

# --- Verification ---
# This part confirms that the cleaning was successful and the file was created.
print(f"Successfully cleaned the dataset and saved it to '{output_file}'")

# Load the cleaned data to verify its integrity
cleaned_df = pd.read_csv(output_file)

print("\n--- Verification of Cleaned Data ---")
print(f"Shape of the cleaned data: {cleaned_df.shape}")

# Check for any remaining missing values
missing_values_summary = cleaned_df.isnull().sum()
total_missing = missing_values_summary.sum()

print("\nMissing values per column in the cleaned file:")
print(missing_values_summary)
print(f"\nTotal remaining missing values: {total_missing}")

if total_missing == 0:
    print("\nVerification successful: The cleaned dataset has no missing values.")
else:
```

```
    print("\nVerification failed: The cleaned dataset still contains missing values.")
```

Listing 16: **Solution code.**

Table 12: **Generated output** by DS-STAR in csv format.

| symboling | normalized-losses | make | fuel-type | aspiration | num-of-doors | body-style | drive-wheels | engine-location | wheel-base | length | width | height | curb-weight | engine-type | num-of-cylinders | engine-size | fuel-system | bore | stroke | compression-ratio | horsepower | peak-rpm | city-mpg | highway-mpg | price |
|---|---|---|---|---|---|---|---|---|---|---|---|---|---|---|---|---|---|---|---|---|---|---|---|---|---|
| 3 | 122.0 | alfa-romero | gas | std | two | convertible | rwd | front | 88.6 | 168.8 | 64.1 | 48.8 | 2548 | dohc | four | 130 | mpfi | 3.47 | 2.68 | 9.0 | 111 | 5000 | 21 | 27 | 13495 |
| 3 | 122.0 | alfa-romero | gas | std | two | convertible | rwd | front | 88.6 | 168.8 | 64.1 | 48.8 | 2548 | dohc | four | 130 | mpfi | 3.47 | 2.68 | 9.0 | 111 | 5000 | 21 | 27 | 16500 |
| 1 | 122.0 | alfa-romero | gas | std | two | hatchback | rwd | front | 94.5 | 171.2 | 65.5 | 52.4 | 2823 | ohcv | six | 152 | mpfi | 2.68 | 3.47 | 9.0 | 154 | 5000 | 19 | 26 | 16500 |
| ... | ... | ... | ... | ... | ... | ... | ... | ... | ... | ... | ... | ... | ... | ... | ... | ... | ... | ... | ... | ... | ... | ... | ... | ... | ... |

## I.5 Machine learning

```
f'''As a contestant in the Bank customer data churn prediction competition, you need to predict the data in
↪  test.csv according to the competition requirements. Write the results into submission.csv following the
↪  format specified in sample_submission.csv.'''
```

Listing 17: **User's question.**

```python
import pandas as pd
import numpy as np
from sklearn.model_selection import StratifiedKFold
import lightgbm as lgb
from sklearn.metrics import roc_auc_score
import optuna
import os

# Create the 'final' directory if it doesn't already exist to store the output.
os.makedirs('final', exist_ok=True)

# Suppress Optuna's logging to keep the output clean during the study.
optuna.logging.set_verbosity(optuna.logging.WARNING)

# 1. Load All Data
# Load data from the 'data/' directory.
original_train_df = pd.read_csv('data/train.csv')
test_df = pd.read_csv('data/test.csv')
additional_data_df = pd.read_csv('data/Churn_Modelling.csv')

# The provided test.csv uses 'CustomerId' as the unique identifier.
# We will use 'CustomerId' for the submission file's 'id' column as required by the competition format.
test_ids = test_df['CustomerId']

# 2. Preprocess Data: Augment and Clean
# Select a consistent set of feature columns and the target variable ('Exited') to ensure uniformity across datasets.
feature_cols = ['CreditScore', 'Geography', 'Gender', 'Age', 'Tenure', 'Balance', 'NumOfProducts', 'HasCrCard', 'IsActiveMember', 'EstimatedSalary', 'Exited']

# Augment the training data by combining the original training set with the additional 'Churn_Modelling.csv' dataset.
train_df = pd.concat([original_train_df[feature_cols], additional_data_df[feature_cols]], ignore_index=True)

# --- START: Feature Engineering ---
def feature_engineer(df):
    """Creates new features based on domain knowledge and interactions between existing features."""
    # A small epsilon is added to denominators to prevent division by zero errors.
    epsilon = 1e-6
    df['BalanceSalaryRatio'] = df['Balance'] / (df['EstimatedSalary'] + epsilon)
    df['TenureByAge'] = df['Tenure'] / (df['Age'] + epsilon)
    df['CreditScoreGivenAge'] = df['CreditScore'] / (df['Age'] + epsilon)
    df['Loyalty'] = df['Tenure'] * df['IsActiveMember']
    df['ProductsPerTenure'] = df['NumOfProducts'] / (df['Tenure'] + epsilon)
    return df

print("Applying feature engineering to training and test data...")
train_df = feature_engineer(train_df)
test_df = feature_engineer(test_df)
# --- END: Feature Engineering ---

# Drop identifier columns ('CustomerId', 'Surname', 'id') that are not useful for the model.
# 'errors='ignore'' handles cases where a column might not exist in a dataframe.
drop_cols = ['CustomerId', 'Surname', 'id']
train_df = train_df.drop(columns=[col for col in drop_cols if col in train_df.columns and col != 'Exited'], errors='ignore')
test_df = test_df.drop(columns=[col for col in drop_cols if col in test_df.columns], errors='ignore')

# One-Hot Encode categorical features ('Geography', 'Gender') to convert them into a machine-readable format.
categorical_features = ['Geography', 'Gender']
train_df = pd.get_dummies(train_df, columns=categorical_features, drop_first=True)
test_df = pd.get_dummies(test_df, columns=categorical_features, drop_first=True)

# Separate the target variable ('Exited') from the training features.
train_labels = train_df['Exited']
train_df = train_df.drop(columns=['Exited'])

# Align the training and test dataframes to ensure they have the exact same feature columns in the same order.
# 'inner' join keeps only columns that are present in both dataframes.
train_df, test_df = train_df.align(test_df, join='inner', axis=1)

# Define the final feature set (X) and target (y).
X = train_df
y = train_labels
```

```
# 3. Define the Objective Function for Optuna Hyperparameter Tuning
def objective(trial):
    """Defines the hyperparameter search space and evaluation metric for Optuna to optimize."""
    # Define the hyperparameter search space for the LightGBM model.
    params = {
        'objective': 'binary',
        'metric': 'auc',
        'boosting_type': 'gbdt',
        'n_estimators': trial.suggest_int('n_estimators', 2000, 10000),
        'learning_rate': trial.suggest_float('learning_rate', 0.01, 0.05),
        'num_leaves': trial.suggest_int('num_leaves', 20, 100),
        'max_depth': trial.suggest_int('max_depth', 5, 10),
        'seed': 42,
        'n_jobs': -1,
        'verbose': -1,
        'colsample_bytree': trial.suggest_float('colsample_bytree', 0.5, 0.9),
        'subsample': trial.suggest_float('subsample', 0.5, 0.9),
        'reg_alpha': trial.suggest_float('reg_alpha', 0.01, 0.5),
        'reg_lambda': trial.suggest_float('reg_lambda', 0.01, 0.5),
    }

    N_SPLITS = 5
    skf = StratifiedKFold(n_splits=N_SPLITS, shuffle=True, random_state=42)
    oof_auc_scores = []

    # Perform cross-validation to get a robust estimate of the model's performance for a given set of hyperparameters.
    for fold, (train_idx, val_idx) in enumerate(skf.split(X, y)):
        X_train, y_train = X.iloc[train_idx], y.iloc[train_idx]
        X_val, y_val = X.iloc[val_idx], y.iloc[val_idx]

        model = lgb.LGBMClassifier(**params)
        model.fit(X_train, y_train,
                  eval_set=[(X_val, y_val)],
                  eval_metric='auc',
                  callbacks=[lgb.early_stopping(100, verbose=False)])

        val_preds = model.predict_proba(X_val)[:, 1]
        auc = roc_auc_score(y_val, val_preds)
        oof_auc_scores.append(auc)

    # Return the mean AUC score, which Optuna will try to maximize.
    return np.mean(oof_auc_scores)

# 4. Run the Optimization Study
print("Running hyperparameter optimization...")
study = optuna.create_study(direction='maximize')
# The number of trials can be adjusted; 50 provides a reasonable balance between search time and performance.
study.optimize(objective, n_trials=50)
best_params = study.best_params
print(f"\nBest trial parameters: {best_params}")
print(f"Best trial mean OOF AUC: {study.best_value}")

# 5. Train Final Model with Best Parameters and Create Submission
print("\nTraining final model with tuned parameters...")
N_SPLITS = 5
skf = StratifiedKFold(n_splits=N_SPLITS, shuffle=True, random_state=42)
test_preds = np.zeros(len(test_df))
oof_auc_scores = []

# Combine the best parameters found by Optuna with fixed model settings.
final_params = {
    'objective': 'binary', 'metric': 'auc', 'boosting_type': 'gbdt',
    'seed': 42, 'n_jobs': -1, 'verbose': -1, **best_params
}

for fold, (train_idx, val_idx) in enumerate(skf.split(X, y)):
    print(f"===== Fold {fold+1} =====")
    X_train, y_train = X.iloc[train_idx], y.iloc[train_idx]
    X_val, y_val = X.iloc[val_idx], y.iloc[val_idx]

    model = lgb.LGBMClassifier(**final_params)
    model.fit(X_train, y_train,
              eval_set=[(X_val, y_val)],
              eval_metric='auc',
              callbacks=[lgb.early_stopping(100, verbose=False)])

    val_preds = model.predict_proba(X_val)[:, 1]
    auc = roc_auc_score(y_val, val_preds)
    oof_auc_scores.append(auc)
    print(f"Fold {fold+1} AUC: {auc}")

    # Accumulate predictions on the test set, averaging over the folds to improve robustness.
    test_preds += model.predict_proba(test_df)[:, 1] / N_SPLITS

# Print the final average validation score from the cross-validation.
print(f"\nAverage OOF AUC with tuned parameters: {np.mean(oof_auc_scores)}")

# Create the submission DataFrame using the averaged test predictions.
submission_df = pd.DataFrame({'id': test_ids, 'Exited': test_preds})

# Save the DataFrame to a CSV file in the 'final' directory for submission.
submission_df.to_csv('final/submission.csv', index=False)

print("\nSubmission file created successfully at 'final/submission.csv'.")
print("First 5 rows of the submission file:")
print(submission_df.head())
```

Listing 18: **Solution code and machine learning model.**

## I.6  Visualization

```
f'''Draw a stacked bar chart that displays the percentage of restaurants offering online ordering options
↪  versus those not offering online ordering options across different rating levels. Save this chart as
↪  'result.jpg'. The title of the image should be "Percentage of Restaurants' Online Order Option by
↪  Rating", with the xlabel as "Rating", and the ylabel as "Percentage of Online Orders".'''
```

Listing 19: **User's question.**

```python
import pandas as pd
import sqlite3
import zipfile
import os
import numpy as np
import matplotlib.pyplot as plt

def generate_chart():
    """
    Loads data from the Zomato SQLite database, cleans the 'rate' column,
    calculates the percentage of restaurants offering online ordering for each rating,
    and generates a stacked bar chart of the results, saving it to the 'final' directory.
    """
    # Define file paths
    zip_path = 'data/zomato_rawdata.sqlite.zip'
    db_name = 'zomato_rawdata.sqlite'
    extract_path = '.'  # Extract to the current directory
    output_dir = 'final'
    output_path = os.path.join(output_dir, 'result.jpg')

    # --- Step 1: Create output directory if it doesn't exist ---
    if not os.path.exists(output_dir):
        os.makedirs(output_dir)

    # --- Step 2: Extract the database from the zip file ---
    db_path = os.path.join(extract_path, db_name)
    with zipfile.ZipFile(zip_path, 'r') as zip_ref:
        # Extract the database file, ensuring the correct member is chosen
        for member in zip_ref.infolist():
            if db_name in member.filename:
                # To handle cases where the zip file has directory structures
                member.filename = os.path.basename(member.filename)
                zip_ref.extract(member, path=extract_path)
                break

    # --- Step 3: Load data from the SQLite database ---
    conn = None
    try:
        conn = sqlite3.connect(db_path)
        # Select only the columns needed for this analysis
        query = "SELECT rate, online_order FROM Users"
        df = pd.read_sql_query(query, conn)
    finally:
        if conn:
            conn.close()

    # --- Step 4: Clean the 'rate' column ---
    # Make a copy to avoid SettingWithCopyWarning
    df = df.copy()

    # Replace special non-numeric strings 'NEW' and '-' with numpy.nan
    df['rate'] = df['rate'].replace(['NEW', '-'], np.nan)

    # Remove the '/5' suffix from the rating strings
    df['rate'] = df['rate'].str.replace('/5', '').str.strip()

    # Convert the cleaned column to a numeric (float) type
    df['rate'] = pd.to_numeric(df['rate'], errors='coerce')

    # Drop rows where 'rate' or 'online_order' is NaN as they are not useful for this analysis
    df.dropna(subset=['rate', 'online_order'], inplace=True)

    # --- Step 5: Aggregate the data for plotting ---
    # Create a cross-tabulation to get counts of 'Yes' and 'No' for each rating
    online_order_counts = pd.crosstab(df['rate'], df['online_order'])

    # Calculate the percentage by dividing each row by its sum
    rate_online_percentage_df = online_order_counts.div(online_order_counts.sum(axis=1), axis=0) * 100

    # --- Step 6: Generate and save the stacked bar chart ---
    # Create the plot figure and axes
    fig, ax = plt.subplots(figsize=(14, 8))

    # Generate the stacked bar chart
    rate_online_percentage_df.plot(kind='bar', stacked=True, ax=ax)

    # Set the title and labels as required
    ax.set_title("Percentage of Restaurants' Online Order Option by Rating", fontsize=16)
    ax.set_xlabel("Rating", fontsize=12)
    ax.set_ylabel("Percentage of Online Orders", fontsize=12)

    # Improve aesthetics and readability
    ax.tick_params(axis='x', rotation=90)
    ax.legend(title='Online Order')
    ax.grid(axis='y', linestyle='--', alpha=0.7)

    # Adjust layout to prevent labels from being cut off
    plt.tight_layout()

    # Save the resulting chart to the final directory
    plt.savefig(output_path)

    # Close the plot to free up memory
    plt.close(fig)

    print(f"Chart has been generated and saved as '{output_path}'.")

    # --- Step 7: Clean up the extracted file ---
    if os.path.exists(db_path):
        os.remove(db_path)

# Execute the function to generate the chart
```

```
generate_chart()
```

Listing 20: **Solution code.**

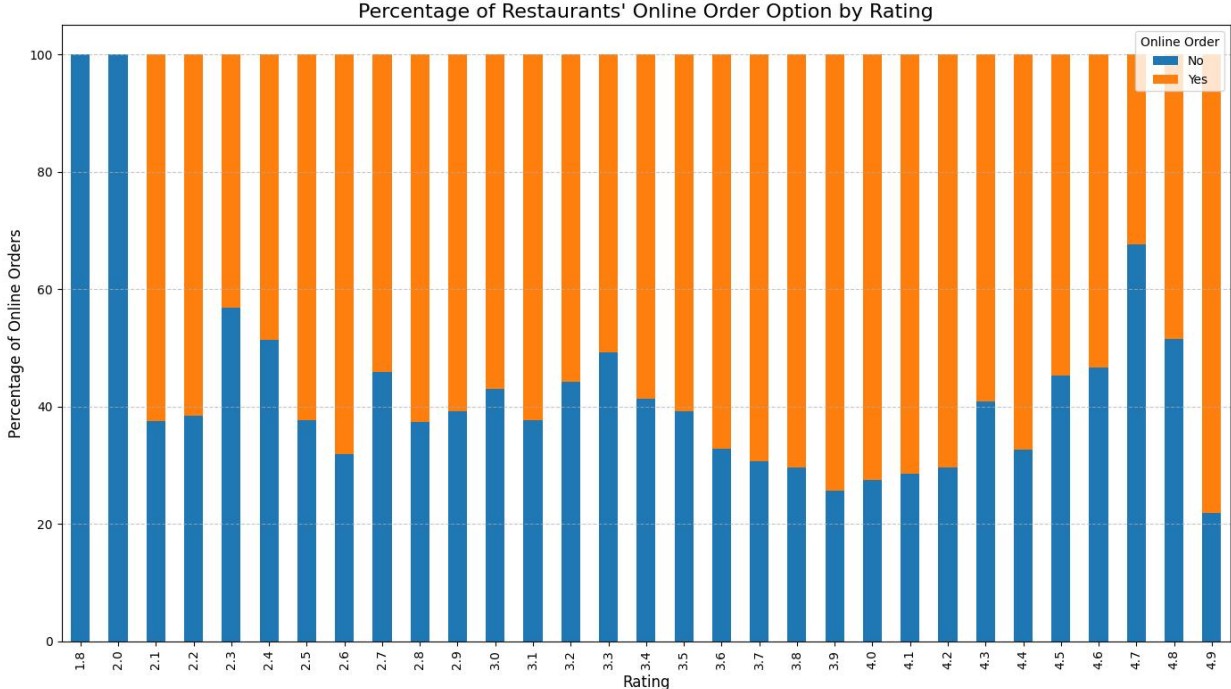

Figure 9: **Generated bar chart.**

## J Full execution logs of DS-STAR

In this section, we provide full logs of DS-STAR to better understand its behavior.

```
f'''In February 2023 what delta would Rafa_AI pay if the relative fee of the fee with ID=17 changed to
↪  99?'''
```

Listing 21: **User's question.**

```
# Step 1
f'''First, I will filter the `payments.csv` data to select only the transactions for the merchant 'Rafa_AI'
↪  that occurred in February 2023. Since 2023 is not a leap year, February corresponds to the days of the
↪  year from 32 to 59.'''
```

Listing 22: **Round 0: Initial plan.**

```python
import pandas as pd

# Load the payments data
try:
    payments_df = pd.read_csv('data/payments.csv')
except FileNotFoundError:
    print("Error: 'data/payments.csv' not found. Please ensure the file is in the correct directory.")
    exit()

# Define the filter for the merchant 'Rafa_AI'
merchant_filter = payments_df['merchant'] == 'Rafa_AI'

# Define the filter for February 2023.
# 2023 is not a leap year. January has 31 days.
# February runs from day 32 (31+1) to day 59 (31+28).
date_filter = (payments_df['day_of_year'] >= 32) & (payments_df['day_of_year'] <= 59)

# Apply both filters to the DataFrame
rafa_ai_feb_transactions = payments_df[merchant_filter & date_filter]

# Display the resulting filtered DataFrame
print(rafa_ai_feb_transactions)
```

Listing 23: **Round 0: Initial implementation.**

```
f'''Insufficient'''
```

Listing 24: **Round 0: Validation.**

```
f'''Add Step'''
```

Listing 25: **Round 0: Routing.** Add Step 2 to the initial plan.

```
# Step 1
f'''First, I will filter the `payments.csv` data to select only the transactions for the merchant 'Rafa_AI'
↪  that occurred in February 2023. Since 2023 is not a leap year, February corresponds to the days of the
↪  year from 32 to 59.'''

# Step 2
f'''I will calculate the original total fee paid by Rafa_AI in February 2023. This involves loading the
↪  `merchant_data.json` and `fees.json` files, enriching the transaction data with all attributes required
↪  for fee matching (including merchant details, monthly volume, monthly fraud rate, and intracountry
↪  status), then applying the corresponding fee rule to each transaction and summing the calculated
↪  fees..'''
```

Listing 26: **Round 1: Plan.**

```python
import pandas as pd
import json

# --- 1. Load All Necessary Data ---
try:
    payments_df = pd.read_csv('data/payments.csv')
    merchant_data_df = pd.read_json('data/merchant_data.json')
    fees_df = pd.read_json('data/fees.json')
    acquirer_countries_df = pd.read_csv('data/acquirer_countries.csv')
except FileNotFoundError as e:
    print(f"Error loading data: {e}. Please ensure all data files are in the 'data/' directory.")
    exit()

# --- 2. Filter Transactions (Based on Previous Plan) ---
# Define the filter for the merchant 'Rafa_AI'
merchant_filter = payments_df['merchant'] == 'Rafa_AI'

# Define the filter for February 2023.
# 2023 is not a leap year. January has 31 days.
# February runs from day 32 (31+1) to day 59 (31+28).
date_filter = (payments_df['day_of_year'] >= 32) & (payments_df['day_of_year'] <= 59)

# Apply both filters to the DataFrame and create a copy to avoid warnings
rafa_ai_feb_transactions = payments_df[merchant_filter & date_filter].copy()

# --- 3. Enrich Transaction Data with Merchant and Acquirer Information ---

# Get Rafa_AI's specific merchant data
rafa_ai_merchant_info = merchant_data_df[merchant_data_df['merchant'] == 'Rafa_AI'].iloc[0]

# Add merchant-specific attributes to the transaction DataFrame
rafa_ai_feb_transactions['account_type'] = rafa_ai_merchant_info['account_type']
rafa_ai_feb_transactions['capture_delay'] = str(rafa_ai_merchant_info['capture_delay']) # Ensure consistent type
rafa_ai_feb_transactions['merchant_category_code'] = rafa_ai_merchant_info['merchant_category_code']

# FIX: The original code failed on the next line.
# The error "ValueError: Length of values (1) does not match length of index (2126)"
# suggests that rafa_ai_merchant_info['acquirer'] is a list-like object with one element
# (e.g., ['gringotts']) instead of a scalar string. Assigning a list of length 1 to a
# column of a different length causes this error.
# The fix is to extract the scalar string from the list before assigning it.
acquirer_name = rafa_ai_merchant_info['acquirer']
if isinstance(acquirer_name, list):
    # If it's a list, take the first element. If empty, it becomes None.
    acquirer_name = acquirer_name[0] if acquirer_name else None

# Now assign the scalar acquirer name to the new column.
rafa_ai_feb_transactions['acquirer'] = acquirer_name

# Create a mapping from acquirer name to country code
acquirer_country_map = acquirer_countries_df.set_index('acquirer')['country_code']

# Add acquirer country to the transaction data
rafa_ai_feb_transactions['acquirer_country'] = rafa_ai_feb_transactions['acquirer'].map(acquirer_country_map)

# Determine if each transaction is intracountry
rafa_ai_feb_transactions['intracountry'] = rafa_ai_feb_transactions['issuing_country'] == rafa_ai_feb_transactions['acquirer_country']

# --- 4. Calculate Monthly Aggregates for Fee Matching ---

# Calculate total transaction volume for Rafa_AI in February
monthly_volume = rafa_ai_feb_transactions['eur_amount'].sum()

# Calculate fraudulent transaction volume for the same period
fraudulent_volume = rafa_ai_feb_transactions[rafa_ai_feb_transactions['has_fraudulent_dispute']]['eur_amount'].sum()

# Calculate the monthly fraud rate as a percentage
# Avoid division by zero if there's no volume
monthly_fraud_rate = (fraudulent_volume / monthly_volume) if monthly_volume > 0 else 0
monthly_fraud_percentage = monthly_fraud_rate * 100

# Add these aggregates to each transaction row for easy lookup during fee matching
rafa_ai_feb_transactions['monthly_volume'] = monthly_volume
rafa_ai_feb_transactions['monthly_fraud_percentage'] = monthly_fraud_percentage

# --- 5. Implement Fee Matching Logic ---

def parse_numeric_value(s):
    """Helper to parse strings like '100k' or '1m' into numbers."""
    s = s.strip().lower()
    if 'k' in s:
        return float(s.replace('k', '')) * 1_000
```

```python
    if 'm' in s:
        return float(s.replace('m', '')) * 1_000_000
    return float(s)

def check_volume_rule(rule_str, value):
    """Checks if a value fits a volume rule string (e.g., '100k-1m')."""
    if pd.isnull(rule_str): return True
    rule_str = rule_str.replace(' ', '')
    if '-' in rule_str:
        low_str, high_str = rule_str.split('-')
        return parse_numeric_value(low_str) <= value < parse_numeric_value(high_str)
    if '>' in rule_str:
        return value >= parse_numeric_value(rule_str.replace('>', ''))
    if '<' in rule_str:
        return value < parse_numeric_value(rule_str.replace('<', ''))
    return False

def check_fraud_rule(rule_str, value):
    """Checks if a value fits a fraud rule string (e.g., '7.7%-8.3%')."""
    if pd.isnull(rule_str): return True
    rule_str = rule_str.replace('%', '')
    if '-' in rule_str:
        low_str, high_str = rule_str.split('-')
        return float(low_str) <= value < float(high_str)
    if '>' in rule_str:
        return value >= float(rule_str.replace('>', ''))
    if '<' in rule_str:
        return value < float(rule_str.replace('<', ''))
    return False

def find_matching_fee_rule(transaction, all_rules_df):
    """Iterates through fee rules to find the first one that matches the transaction."""
    for _, rule in all_rules_df.iterrows():
        # A rule with an empty list for a category applies to all values in that category
        if rule['card_scheme'] != transaction['card_scheme']: continue
        if rule['account_type'] and transaction['account_type'] not in rule['account_type']: continue
        if rule['merchant_category_code'] and transaction['merchant_category_code'] not in rule['merchant_category_code']: continue
        if rule['aci'] and transaction['aci'] not in rule['aci']: continue

        # A rule with a null value for a category applies to all values
        if pd.notnull(rule['is_credit']) and rule['is_credit'] != transaction['is_credit']: continue
        if pd.notnull(rule['intracountry']) and rule['intracountry'] != transaction['intracountry']: continue
        if pd.notnull(rule['capture_delay']) and rule['capture_delay'] != transaction['capture_delay']: continue

        # Check complex range-based rules
        if not check_volume_rule(rule['monthly_volume'], transaction['monthly_volume']): continue
        if not check_fraud_rule(rule['monthly_fraud_level'], transaction['monthly_fraud_percentage']): continue

        # If all checks pass, this is the correct rule
        return rule['fixed_amount'], rule['rate']

    # Return default/zero fees if no rule is found (should not happen with complete data)
    return 0, 0

# --- 6. Calculate Fees and Final Sum ---

# Apply the matching function to each transaction to get its specific fee parameters
fee_params = rafa_ai_feb_transactions.apply(
    lambda row: find_matching_fee_rule(row, fees_df),
    axis=1
)

# Assign the found fee parameters back to the DataFrame
rafa_ai_feb_transactions[['fixed_amount', 'rate']] = pd.DataFrame(
    fee_params.tolist(),
    index=rafa_ai_feb_transactions.index
)

# Calculate the fee for each transaction using the formula from the manual
# fee = fixed_amount + rate * transaction_value / 10000
rafa_ai_feb_transactions['fee'] = rafa_ai_feb_transactions['fixed_amount'] + \
                        (rafa_ai_feb_transactions['rate'] * rafa_ai_feb_transactions['eur_amount'] / 10000)

# Calculate the total fee by summing up the fees for all transactions
total_original_fee = rafa_ai_feb_transactions['fee'].sum()

# Display the final result
print(f"The original total fee paid by Rafa_AI in February 2023 was: {total_original_fee:.2f} EUR")
```

Listing 27: **Round 1: Implementation.**

```
f'''Insufficient'''
```

Listing 28: **Round 1: Validation.**

```
f'''Add Step'''
```

Listing 29: **Round 1: Routing.** Add Step 3 to the current plan.

```
# Step 1
```

```
f'''First, I will filter the `payments.csv` data to select only the transactions for the merchant 'Rafa_AI'
↪   that occurred in February 2023. Since 2023 is not a leap year, February corresponds to the days of the
↪   year from 32 to 59.'''

# Step 2
f'''I will calculate the original total fee paid by Rafa_AI in February 2023. This involves loading the
↪   `merchant_data.json` and `fees.json` files, enriching the transaction data with all attributes required
↪   for fee matching (including merchant details, monthly volume, monthly fraud rate, and intracountry
↪   status), then applying the corresponding fee rule to each transaction and summing the calculated
↪   fees..'''

# Step 3
f'''I will calculate the new total fee for Rafa_AI's February 2023 transactions using the modified fee
↪   structure where the fee rule with ID=17 has its `rate` changed to 99. Then, I will subtract the original
↪   total fee (778.52 EUR) from this new total fee to find the requested delta.'''
```

Listing 30: **Round 2: Plan.**

```python
import pandas as pd
import json

# --- 1. Load All Necessary Data ---
try:
    payments_df = pd.read_csv('data/payments.csv')
    merchant_data_df = pd.read_json('data/merchant_data.json')
    fees_df = pd.read_json('data/fees.json')
    acquirer_countries_df = pd.read_csv('data/acquirer_countries.csv')
except FileNotFoundError as e:
    print(f"Error loading data: {e}. Please ensure all data files are in the 'data/' directory.")
    exit()

# --- 2. Filter Transactions (Based on Previous Plan) ---
# Define the filter for the merchant 'Rafa_AI'
merchant_filter = payments_df['merchant'] == 'Rafa_AI'

# Define the filter for February 2023.
# 2023 is not a leap year. January has 31 days.
# February runs from day 32 (31+1) to day 59 (31+28).
date_filter = (payments_df['day_of_year'] >= 32) & (payments_df['day_of_year'] <= 59)

# Apply both filters to the DataFrame and create a copy to avoid warnings
rafa_ai_feb_transactions = payments_df[merchant_filter & date_filter].copy()

# --- 3. Enrich Transaction Data with Merchant and Acquirer Information ---

# Get Rafa_AI's specific merchant data
rafa_ai_merchant_info = merchant_data_df[merchant_data_df['merchant'] == 'Rafa_AI'].iloc[0]

# Add merchant-specific attributes to the transaction DataFrame
rafa_ai_feb_transactions['account_type'] = rafa_ai_merchant_info['account_type']
rafa_ai_feb_transactions['capture_delay'] = str(rafa_ai_merchant_info['capture_delay']) # Ensure consistent type
rafa_ai_feb_transactions['merchant_category_code'] = rafa_ai_merchant_info['merchant_category_code']

# FIX: The original code failed on the next line.
# The error "ValueError: Length of values (1) does not match length of index (2126)"
# suggests that rafa_ai_merchant_info['acquirer'] is a list-like object with one element
# (e.g., ['gringotts']) instead of a scalar string. Assigning a list of length 1 to a
# column of a different length causes this error.
# The fix is to extract the scalar string from the list before assigning it.
acquirer_name = rafa_ai_merchant_info['acquirer']
if isinstance(acquirer_name, list):
    # If it's a list, take the first element. If empty, it becomes None.
    acquirer_name = acquirer_name[0] if acquirer_name else None

# Now assign the scalar acquirer name to the new column.
rafa_ai_feb_transactions['acquirer'] = acquirer_name

# Create a mapping from acquirer name to country code
acquirer_country_map = acquirer_countries_df.set_index('acquirer')['country_code']

# Add acquirer country to the transaction data
rafa_ai_feb_transactions['acquirer_country'] = rafa_ai_feb_transactions['acquirer'].map(acquirer_country_map)

# Determine if each transaction is intracountry
rafa_ai_feb_transactions['intracountry'] = rafa_ai_feb_transactions['issuing_country'] == rafa_ai_feb_transactions['acquirer_country']

# --- 4. Calculate Monthly Aggregates for Fee Matching ---

# Calculate total transaction volume for Rafa_AI in February
monthly_volume = rafa_ai_feb_transactions['eur_amount'].sum()

# Calculate fraudulent transaction volume for the same period
fraudulent_volume = rafa_ai_feb_transactions[rafa_ai_feb_transactions['has_fraudulent_dispute']]['eur_amount'].sum()

# Calculate the monthly fraud rate as a percentage
# Avoid division by zero if there's no volume
monthly_fraud_rate = (fraudulent_volume / monthly_volume) if monthly_volume > 0 else 0
monthly_fraud_percentage = monthly_fraud_rate * 100

# Add these aggregates to each transaction row for easy lookup during fee matching
rafa_ai_feb_transactions['monthly_volume'] = monthly_volume
rafa_ai_feb_transactions['monthly_fraud_percentage'] = monthly_fraud_percentage

# --- 5. Implement Fee Matching Logic ---
```

```python
def parse_numeric_value(s):
    """Helper to parse strings like '100k' or '1m' into numbers."""
    s = s.strip().lower()
    if 'k' in s:
        return float(s.replace('k', '')) * 1_000
    if 'm' in s:
        return float(s.replace('m', '')) * 1_000_000
    return float(s)

def check_volume_rule(rule_str, value):
    """Checks if a value fits a volume rule string (e.g., '100k-1m')."""
    if pd.isnull(rule_str): return True
    rule_str = rule_str.replace(' ', '')
    if '-' in rule_str:
        low_str, high_str = rule_str.split('-')
        return parse_numeric_value(low_str) <= value < parse_numeric_value(high_str)
    if '>' in rule_str:
        return value >= parse_numeric_value(rule_str.replace('>', ''))
    if '<' in rule_str:
        return value < parse_numeric_value(rule_str.replace('<', ''))
    return False

def check_fraud_rule(rule_str, value):
    """Checks if a value fits a fraud rule string (e.g., '7.7%-8.3%')."""
    if pd.isnull(rule_str): return True
    rule_str = rule_str.replace('%', '')
    if '-' in rule_str:
        low_str, high_str = rule_str.split('-')
        return float(low_str) <= value < float(high_str)
    if '>' in rule_str:
        return value >= float(rule_str.replace('>', ''))
    if '<' in rule_str:
        return value < float(rule_str.replace('<', ''))
    return False

def find_matching_fee_rule(transaction, all_rules_df):
    """Iterates through fee rules to find the first one that matches the transaction."""
    for _, rule in all_rules_df.iterrows():
        # A rule with an empty list for a category applies to all values in that category
        if rule['card_scheme'] != transaction['card_scheme']: continue
        if rule['account_type'] and transaction['account_type'] not in rule['account_type']: continue
        if rule['merchant_category_code'] and transaction['merchant_category_code'] not in rule['merchant_category_code']: continue
        if rule['aci'] and transaction['aci'] not in rule['aci']: continue

        # A rule with a null value for a category applies to all values
        if pd.notnull(rule['is_credit']) and rule['is_credit'] != transaction['is_credit']: continue
        if pd.notnull(rule['intracountry']) and rule['intracountry'] != transaction['intracountry']: continue
        if pd.notnull(rule['capture_delay']) and rule['capture_delay'] != transaction['capture_delay']: continue

        # Check complex range-based rules
        if not check_volume_rule(rule['monthly_volume'], transaction['monthly_volume']): continue
        if not check_fraud_rule(rule['monthly_fraud_level'], transaction['monthly_fraud_percentage']): continue

        # If all checks pass, this is the correct rule
        return rule['fixed_amount'], rule['rate']

    # Return default/zero fees if no rule is found (should not happen with complete data)
    return 0, 0

# --- 6. Calculate Fees and Final Sum ---

# Apply the matching function to each transaction to get its specific fee parameters
fee_params = rafa_ai_feb_transactions.apply(
    lambda row: find_matching_fee_rule(row, fees_df),
    axis=1
)

# Assign the found fee parameters back to the DataFrame
rafa_ai_feb_transactions[['fixed_amount', 'rate']] = pd.DataFrame(
    fee_params.tolist(),
    index=rafa_ai_feb_transactions.index
)

# Calculate the fee for each transaction using the formula from the manual
# fee = fixed_amount + rate * transaction_value / 10000
rafa_ai_feb_transactions['fee'] = rafa_ai_feb_transactions['fixed_amount'] + \
                                  (rafa_ai_feb_transactions['rate'] * rafa_ai_feb_transactions['eur_amount'] / 10000)

# Calculate the total fee by summing up the fees for all transactions
total_original_fee = rafa_ai_feb_transactions['fee'].sum()

# Display the final result from the previous step
print(f"The original total fee paid by Rafa_AI in February 2023 was: {total_original_fee:.2f} EUR")

# --- 7. Calculate New Fee with Modified Rule ---

# Create a copy of the fees dataframe to avoid altering the original
fees_df_modified = fees_df.copy()

# Implement the plan: change the rate for the rule with ID=17 to 99
fees_df_modified.loc[fees_df_modified['ID'] == 17, 'rate'] = 99

# Apply the matching function again, this time with the modified fee rules
new_fee_params = rafa_ai_feb_transactions.apply(
    lambda row: find_matching_fee_rule(row, fees_df_modified),
    axis=1
)

# Assign the new fee parameters to new columns in the DataFrame
rafa_ai_feb_transactions[['new_fixed_amount', 'new_rate']] = pd.DataFrame(
    new_fee_params.tolist(),
    index=rafa_ai_feb_transactions.index
)

# Calculate the new fee for each transaction using the new rate
rafa_ai_feb_transactions['new_fee'] = rafa_ai_feb_transactions['new_fixed_amount'] + \
                                      (rafa_ai_feb_transactions['new_rate'] * rafa_ai_feb_transactions['eur_amount'] / 10000)

# Calculate the new total fee by summing up the individual new fees
new_total_fee = rafa_ai_feb_transactions['new_fee'].sum()
```

```
# Calculate the difference (delta) between the new total fee and the original total fee
fee_delta = new_total_fee - total_original_fee

# --- 8. Display the Final Delta ---
print(f"The new total fee with the modified rule would be: {new_total_fee:.2f} EUR")
print(f"The requested delta (New Fee - Original Fee) is: {fee_delta:.2f} EUR")
```

Listing 31: **Round 2: Implementation.**

```
f'''Insufficient'''
```

Listing 32: **Round 2: Validation.**

```
f'''Step 3 is wrong'''
```

Listing 33: **Round 2: Routing.** Remove Step 3 from the current plan and add Step 3 again.

```
# Step 1
f'''First, I will filter the `payments.csv` data to select only the transactions for the merchant 'Rafa_AI'
↪  that occurred in February 2023. Since 2023 is not a leap year, February corresponds to the days of the
↪  year from 32 to 59.'''

# Step 2
f'''I will calculate the original total fee paid by Rafa_AI in February 2023. This involves loading the
↪  `merchant_data.json` and `fees.json` files, enriching the transaction data with all attributes required
↪  for fee matching (including merchant details, monthly volume, monthly fraud rate, and intracountry
↪  status), then applying the corresponding fee rule to each transaction and summing the calculated
↪  fees..'''

# Step 3
f'''I will filter the transactions for 'Rafa_AI' in February 2023 (day_of_year from 32 to 59) that meet all
↪  the conditions of fee rule ID=17: `card_scheme` must be 'SwiftCharge', `is_credit` must be True, and
↪  `aci` must be 'A'. Then, for these matching transactions, I will calculate the total fee difference
↪  using the formula: (99 - 60) * SUM(eur_amount) / 10000.'''
```

Listing 34: **Round 3: Plan.**

```
import pandas as pd
import json

# --- 1. Load All Necessary Data ---
# This part is based on the loading logic from the base code.
try:
    payments_df = pd.read_csv('data/payments.csv')
    merchant_data_df = pd.read_json('data/merchant_data.json')
    fees_df = pd.read_json('data/fees.json')
except FileNotFoundError as e:
    print(f"Error loading data: {e}. Please ensure all data files are in the 'data/' directory.")
    exit()

# --- Implementation of the Current Plan ---
# I will filter the transactions for 'Rafa_AI' in February 2023 (day_of_year from 32 to 59)
# that meet all the conditions of fee rule ID=17: `card_scheme` must be 'SwiftCharge',
# `is_credit` must be True, and `aci` must be 'A'. Then, for these matching transactions,
# I will calculate the total fee difference using the formula:
# (99 - 60) * SUM(eur_amount) / 10000.

print("--- Calculating fee difference for 'Rafa_AI' in Feb 2023 under specific conditions ---")

# Step 1: Define the conditions for filtering based on the plan.
# February 2023 corresponds to day_of_year 32 to 59 (2023 is not a leap year).
feb_start_day = 32
feb_end_day = 59
merchant_name = 'Rafa_AI'
required_card_scheme = 'SwiftCharge'
required_is_credit = True
required_aci = 'A'

# Step 2: Filter the payments DataFrame to find transactions matching all conditions.
matching_transactions = payments_df[
    (payments_df['merchant'] == merchant_name) &
    (payments_df['day_of_year'] >= feb_start_day) &
    (payments_df['day_of_year'] <= feb_end_day) &
    (payments_df['card_scheme'] == required_card_scheme) &
    (payments_df['is_credit'] == required_is_credit) &
```

```
        (payments_df['aci'] == required_aci)
]

print(f"\n[INFO] Found {len(matching_transactions)} transactions for '{merchant_name}' in February 2023 matching all specified conditions.")

# Step 3: Calculate the total fee difference for these matching transactions.
if not matching_transactions.empty:
    # Sum the 'eur_amount' for all matching transactions.
    total_matching_volume = matching_transactions['eur_amount'].sum()
    print(f"[INFO] Total volume of these transactions: {total_matching_volume:.2f} EUR")

    # Apply the fee difference formula provided in the plan.
    # Formula: (new_rate - old_rate) * SUM(eur_amount) / 10000
    rate_difference = 99 - 60
    total_fee_difference = (rate_difference * total_matching_volume) / 10000

    # --- Final Conclusion ---
    print("\n---------------------CONCLUSION----------------------")
    print(f"The total fee difference for the matching transactions is: {total_fee_difference:.4f} EUR.")

else:
    # --- Final Conclusion ---
    print("\n---------------------CONCLUSION----------------------")
    print("No transactions were found that match all the specified criteria.")
    print("Therefore, the total fee difference is 0.00 EUR.")
```

Listing 35: **Round 3: Implementation.**

```
f'''Insufficient'''
```

Listing 36: **Round 3: Validation.**

```
f'''Step 3 is wrong'''
```

Listing 37: **Round 3: Routing.** Remove Step 3 from the current plan and add Step 3 again.

```
# Step 1
f'''First, I will filter the `payments.csv` data to select only the transactions for the merchant 'Rafa_AI'
↪  that occurred in February 2023. Since 2023 is not a leap year, February corresponds to the days of the
↪  year from 32 to 59.'''

# Step 2
f'''I will calculate the original total fee paid by Rafa_AI in February 2023. This involves loading the
↪  `merchant_data.json` and `fees.json` files, enriching the transaction data with all attributes required
↪  for fee matching (including merchant details, monthly volume, monthly fraud rate, and intracountry
↪  status), then applying the corresponding fee rule to each transaction and summing the calculated
↪  fees..'''

# Step 3
f'''Calculate the fee delta by subtracting the original `rate` of fee ID 17 from the new `rate` of 99,
↪  multiplying the result by the total volume of the affected transactions (686.48 EUR), and then dividing
↪  by 10000.'''
```

Listing 38: **Round 4: Plan.**

```
import pandas as pd
import json

# --- 1. Load All Necessary Data ---
# This part is based on the loading logic from the base code.
try:
    payments_df = pd.read_csv('data/payments.csv')
    merchant_data_df = pd.read_json('data/merchant_data.json')
    fees_df = pd.read_json('data/fees.json')
except FileNotFoundError as e:
    print(f"Error loading data: {e}. Please ensure all data files are in the 'data/' directory.")
    exit()

# --- Implementation of the Current Plan ---
# Calculate the fee delta by subtracting the original `rate` of fee ID 17 from the new `rate` of 99,
# multiplying the result by the total volume of the affected transactions (686.48 EUR),
# and then dividing by 10000.

print("--- Calculating fee delta for affected 'Rafa_AI' transactions ---")

# Step 1: Define the constants given in the plan.
```

```
new_rate = 99
total_affected_volume = 686.48  # EUR
fee_divisor = 10000

# Step 2: Find the original rate for fee ID 17 from the fees_df DataFrame.
fee_rule_17 = fees_df[fees_df['ID'] == 17]

if fee_rule_17.empty:
    print("\n[ERROR] Fee rule with ID 17 was not found in fees.json.")
    original_rate = None # Set to None to prevent further calculation
else:
    # Extract the 'rate' value from the filtered DataFrame.
    # .iloc[0] selects the first row of the result.
    original_rate = fee_rule_17['rate'].iloc[0]
    print(f"\n[INFO] Original rate for fee ID 17 is: {original_rate}")
    print(f"[INFO] New rate to be applied is: {new_rate}")
    print(f"[INFO] Total volume of affected transactions is: {total_affected_volume:.2f} EUR")

# Step 3: Calculate the fee delta using the specified formula.
if original_rate is not None:
    # Calculate the difference in rates.
    rate_difference = new_rate - original_rate

    # Apply the formula: (rate_difference * total_volume) / 10000
    fee_delta = (rate_difference * total_affected_volume) / fee_divisor

    # --- Final Conclusion ---
    print("\n--------------------CONCLUSION---------------------")
    print(f"The calculated fee delta, by changing the rate from {original_rate} to {new_rate} for the affected volume, is: {fee_delta:.4f} EUR.")

else:
    # --- Final Conclusion ---
    print("\n--------------------CONCLUSION---------------------")
    print("Could not perform calculation because the original fee rate for ID 17 was not found.")
```

Listing 39: **Round 4: Implementation.**

```
f'''Insufficient'''
```

Listing 40: **Round 4: Validation.**

```
f'''Add Step'''
```

Listing 41: **Round 4: Routing.** Add Step 4 to the current plan.

```
# Step 1
f'''First, I will filter the `payments.csv` data to select only the transactions for the merchant 'Rafa_AI'
↪  that occurred in February 2023. Since 2023 is not a leap year, February corresponds to the days of the
↪  year from 32 to 59.'''

# Step 2
f'''I will calculate the original total fee paid by Rafa_AI in February 2023. This involves loading the
↪  `merchant_data.json` and `fees.json` files, enriching the transaction data with all attributes required
↪  for fee matching (including merchant details, monthly volume, monthly fraud rate, and intracountry
↪  status), then applying the corresponding fee rule to each transaction and summing the calculated
↪  fees..'''

# Step 3
f'''Calculate the fee delta by subtracting the original `rate` of fee ID 17 from the new `rate` of 99,
↪  multiplying the result by the total volume of the affected transactions (686.48 EUR), and then dividing
↪  by 10000.'''

# Step 4
f'''Calculate the total original fee for Rafa_AI's transactions in February 2023 that are affected by fee
↪  ID=17, and then calculate the new total fee for the same transactions using the modified rate of 99. The
↪  delta is the difference between these two amounts.'''
```

Listing 42: **Round 5: Plan.**

```
import pandas as pd
import json

# --- 1. Load All Necessary Data ---
```

```python
# This part is based on the loading logic from the base code.
try:
    payments_df = pd.read_csv('data/payments.csv')
    merchant_data_df = pd.read_json('data/merchant_data.json')
    fees_df = pd.read_json('data/fees.json')
except FileNotFoundError as e:
    print(f"Error loading data: {e}. Please ensure all data files are in the 'data/' directory.")
    exit()

# --- Implementation of the Current Plan ---
# Calculate the total original fee for Rafa_AI's transactions in February 2023
# that are affected by fee ID=17, and then calculate the new total fee for the same
# transactions using the modified rate of 99. The delta is the difference.

print("--- Identifying transactions for 'Rafa_AI' in February 2023 affected by fee rule ID 17 ---")

# Step 1: Filter payments for 'Rafa_AI' in February 2023.
# February corresponds to days of the year from 32 to 59 (since 2023 is not a leap year).
feb_transactions_df = payments_df[
    (payments_df['merchant'] == 'Rafa_AI') &
    (payments_df['day_of_year'] >= 32) &
    (payments_df['day_of_year'] <= 59)
].copy()

# Step 2: Enrich transaction data with merchant-specific information.
# Get the specific data for 'Rafa_AI' from the merchant data file.
rafa_ai_data = merchant_data_df[merchant_data_df['merchant'] == 'Rafa_AI']

# Merge the February transactions with the merchant's data.
if not rafa_ai_data.empty and not feb_transactions_df.empty:
    # Use a left merge to add merchant details (like mcc, account_type) to each transaction.
    full_transaction_data = feb_transactions_df.merge(rafa_ai_data, on='merchant', how='left')
else:
    print("\n[ERROR] No transactions found for 'Rafa_AI' in February or 'Rafa_AI' data not found.")
    exit()

# Step 3: Retrieve the specific conditions for the fee rule with ID=17.
fee_rule_17 = fees_df[fees_df['ID'] == 17]

if fee_rule_17.empty:
    print("\n[ERROR] Fee rule with ID 17 was not found in fees.json.")
    exit()
else:
    # Use .iloc[0] to get the rule as a Series
    fee_rule_17 = fee_rule_17.iloc[0]

# Step 4: Filter the transactions that match all conditions of the fee rule.
# The manual states that a null or empty list in a rule field means it applies to all values.
mask = pd.Series(True, index=full_transaction_data.index)

# Match card_scheme (string)
if fee_rule_17['card_scheme']:
    mask &= (full_transaction_data['card_scheme'] == fee_rule_17['card_scheme'])

# Match account_type (list)
if fee_rule_17['account_type']: # An empty list evaluates to False, correctly skipping the filter
    mask &= full_transaction_data['account_type'].isin(fee_rule_17['account_type'])

# Match merchant_category_code (list)
if fee_rule_17['merchant_category_code']:
    mask &= full_transaction_data['merchant_category_code'].isin(fee_rule_17['merchant_category_code'])

# Match aci (list)
if fee_rule_17['aci']:
    mask &= full_transaction_data['aci'].isin(fee_rule_17['aci'])

# Match is_credit (boolean, but can be null)
if fee_rule_17['is_credit'] is not None:
    mask &= (full_transaction_data['is_credit'] == fee_rule_17['is_credit'])

# Apply the combined mask to get the affected transactions
affected_transactions_df = full_transaction_data[mask]

# Step 5: Calculate the original and new fees for the affected transactions.
if affected_transactions_df.empty:
    print("\n[INFO] No transactions for 'Rafa_AI' in February match the criteria for fee rule ID 17.")
    total_original_fee = 0
    total_new_fee = 0
    fee_delta = 0
else:
    total_affected_volume = affected_transactions_df['eur_amount'].sum()
    num_affected_transactions = len(affected_transactions_df)

    print(f"\n[INFO] Found {num_affected_transactions} transactions matching the fee rule.")
    print(f"[INFO] Total volume of these transactions: {total_affected_volume:.2f} EUR")

    # Get fee parameters from the rule
    original_rate = fee_rule_17['rate']
    fixed_amount = fee_rule_17['fixed_amount']
    new_rate = 99
    fee_divisor = 10000

    # Calculate total original fee using the formula: fee = fixed_amount + rate * value / 10000
    # Vectorized calculation for efficiency
    total_fixed_component = fixed_amount * num_affected_transactions
    total_original_rate_component = (original_rate * total_affected_volume) / fee_divisor
    total_original_fee = total_fixed_component + total_original_rate_component

    # Calculate total new fee with the modified rate
    total_new_rate_component = (new_rate * total_affected_volume) / fee_divisor
    total_new_fee = total_fixed_component + total_new_rate_component

    # The delta is the difference between the new and original total fees
    fee_delta = total_new_fee - total_original_fee

# --- Final Conclusion ---
print("\n---------------------CONCLUSION---------------------")
print(f"Total original fee for affected transactions: {total_original_fee:.4f} EUR")
print(f"Total new fee with rate={new_rate}: {total_new_fee:.4f} EUR")
print(f"The calculated fee delta is: {fee_delta:.4f} EUR.")
```

```
print("This represents the total increase in fees for Rafa_AI's affected February transactions due to the rate change.")
```

Listing 43: **Round 5: Implementation.**

```
f'''Sufficient'''
```

Listing 44: **Round 4: Validation.**

```python
import pandas as pd
import json

# --- 1. Load All Necessary Data ---
# This part is based on the loading logic from the base code.
payments_df = pd.read_csv('data/payments.csv')
merchant_data_df = pd.read_json('data/merchant_data.json')
fees_df = pd.read_json('data/fees.json')

# --- Implementation of the Calculation ---
# Calculate the total original fee for Rafa_AI's transactions in February 2023
# that are affected by fee ID=17, and then calculate the new total fee for the same
# transactions using the modified rate of 99. The delta is the difference.

# Step 1: Filter payments for 'Rafa_AI' in February 2023.
# February corresponds to days of the year from 32 to 59 (since 2023 is not a leap year).
feb_transactions_df = payments_df[
    (payments_df['merchant'] == 'Rafa_AI') &
    (payments_df['day_of_year'] >= 32) &
    (payments_df['day_of_year'] <= 59)
].copy()

# Step 2: Enrich transaction data with merchant-specific information.
# Get the specific data for 'Rafa_AI' from the merchant data file.
rafa_ai_data = merchant_data_df[merchant_data_df['merchant'] == 'Rafa_AI']

# Merge the February transactions with the merchant's data.
full_transaction_data = pd.DataFrame()
if not rafa_ai_data.empty and not feb_transactions_df.empty:
    # Use a left merge to add merchant details (like mcc, account_type) to each transaction.
    full_transaction_data = feb_transactions_df.merge(rafa_ai_data, on='merchant', how='left')

# Step 3: Retrieve the specific conditions for the fee rule with ID=17.
fee_delta = 0.0
if not full_transaction_data.empty:
    fee_rule_17 = fees_df[fees_df['ID'] == 17]

    if not fee_rule_17.empty:
        # Use .iloc[0] to get the rule as a Series
        fee_rule_17 = fee_rule_17.iloc[0]

        # Step 4: Filter the transactions that match all conditions of the fee rule.
        # The manual states that a null or empty list in a rule field means it applies to all values.
        mask = pd.Series(True, index=full_transaction_data.index)

        # Match card_scheme (string)
        if fee_rule_17['card_scheme']:
            mask &= (full_transaction_data['card_scheme'] == fee_rule_17['card_scheme'])

        # Match account_type (list)
        if fee_rule_17['account_type']: # An empty list evaluates to False, correctly skipping the filter
            mask &= full_transaction_data['account_type'].isin(fee_rule_17['account_type'])

        # Match merchant_category_code (list)
        if fee_rule_17['merchant_category_code']:
            mask &= full_transaction_data['merchant_category_code'].isin(fee_rule_17['merchant_category_code'])

        # Match aci (list)
        if fee_rule_17['aci']:
            mask &= full_transaction_data['aci'].isin(fee_rule_17['aci'])

        # Match is_credit (boolean, but can be null)
        if fee_rule_17['is_credit'] is not None:
            mask &= (full_transaction_data['is_credit'] == fee_rule_17['is_credit'])

        # Apply the combined mask to get the affected transactions
        affected_transactions_df = full_transaction_data[mask]

        # Step 5: Calculate the original and new fees for the affected transactions.
        if not affected_transactions_df.empty:
            total_affected_volume = affected_transactions_df['eur_amount'].sum()
            num_affected_transactions = len(affected_transactions_df)

            # Get fee parameters from the rule
            original_rate = fee_rule_17['rate']
            fixed_amount = fee_rule_17['fixed_amount']
            new_rate = 99
            fee_divisor = 10000

            # Calculate total original fee using the formula: fee = fixed_amount + rate * value / 10000
            total_fixed_component = fixed_amount * num_affected_transactions
            total_original_rate_component = (original_rate * total_affected_volume) / fee_divisor
            total_original_fee = total_fixed_component + total_original_rate_component

            # Calculate total new fee with the modified rate
            total_new_rate_component = (new_rate * total_affected_volume) / fee_divisor
            total_new_fee = total_fixed_component + total_new_rate_component

            # The delta is the difference between the new and original total fees
            fee_delta = total_new_fee - total_original_fee
```

```
# --- Final Output ---
print(f"{fee_delta:.14f}")
```

Listing 45: **Final solution code.**

```
f'''2.67727200000000'''
```

Listing 46: **Generated answer by DS-STAR.** Obtained by executing the above final solution code.

# K    Prompts for DS-STAR

## K.1    Analyzer agent

```
f'''You are an expert data analysist.
Generate a Python code that loads and describes the content of {filename}.

# Requirement
- The file can both unstructured or structured data.
- If there are too many structured data, print out just few examples.
- Print out essential informations. For example, print out all the column names.
- The Python code should print out the content of {filename}.
- The code should be a single-file Python program that is self-contained and can be executed as-is.
- Your response should only contain a single code block.
- Important: You should not include dummy contents since we will debug if error occurs.
- Do not use try: and except: to prevent error. I will debug it later.'''
```

Listing 47: Prompt used to generate Python scripts that describe data files.

DS-STAR begins by creating a description of each data file. Specifically, it utilizes an analyzer agent $\mathcal{A}_{\text{analyzer}}$ that takes a file name (*e.g.*, 'payment.csv') as input to generate a Python script $s_{\text{desc}}$, using the above prompt. The script is then executed to summarize the dataset's core information. An example of this process, including the generated script and its output, is provided in Appendix H.

## K.2    Planner agent

```
f'''You are an expert data analysist.
In order to answer factoid questions based on the given data, you have to first plan effectively.

# Question
{question}

# Given data: {filenames}
{filenames #1}
{summaries #1}
...
{filenames #N}
{summaries #N}

# Your task
- Suggest your very first step to answer the question above.
- Your first step does not need to be sufficient to answer the question.
- Just propose a very simple initial step, which can act as a good starting point to answer the question.
- Your response should only contain an initial step.'''
```

Listing 48: Prompt used to generate an initial plan.

DS-STAR begins the solution generation process after generating analytic descriptions of $N$ data files. First, a planner agent $\mathcal{A}_{\text{planner}}$ generates an *initial* high-level executable step $p_0$ using the above prompt. Here, the query $q$ and obtained data descriptions are used as inputs.

```
f'''You are an expert data analysist.
In order to answer factoid questions based on the given data, you have to first plan effectively.
Your task is to suggest next plan to do to answer the question.

# Question
{question}

# Given data: {filenames}
{filenames #1}
{summaries #1}
...
{filenames #N}
{summaries #N}

# Current plans
```

```
1. {Step 1}
...
k. {Step k}

# Obtained results from the current plans:
{result}

# Your task
- Suggest your next step to answer the question above.
- Your next step does not need to be sufficient to answer the question, but if it requires only final simple
↪    last step you may suggest it.
- Just propose a very simple next step, which can act as a good intermediate point to answer the question.
- Of course your response can be a plan which could directly answer the question.
- Your response should only contain an next step without any explanation.'''
```

Listing 49: Prompt used to generate a plan after the initialization step.

After the initial plan is obtained, DS-STAR's planner agent generates a subsequent step using the above prompt. Unlike the initialization round, planner agent $\mathcal{A}_{\texttt{analyzer}}$ additionally uses the intermediate step-by-step plans, and the obtained results after execution of its implementation.

### K.3 Coder agent

```
f'''# Given data: {filenames}
{filenames #1}
{summaries #1}
...
{filenames #N}
{summaries #N}

# Plan
{plan}

# Your task
- Implement the plan with the given data.
- Your response should be a single markdown Python code (wrapped in ```).
- There should be no additional headings or text in your response.'''
```

Listing 50: Prompt used to generate a implementation of an initial plan in a Python code.

The initial step for the plan is implemented as a code script $s_0$ by a coder agent $\mathcal{A}_{\texttt{coder}}$, using the above prompt. Here, analytic descriptions of data files are additionally used to provide some essential information like column names.

```
f'''You are an expert data analysist.
Your task is to implement the next plan with the given data.

# Given data: {filenames}
{filenames #1}
{summaries #1}
...
{filenames #N}
{summaries #N}

# Base code
```python
{base_code}
```

# Previous plans
1. {Step 1}
...
k. {Step k}

# Current plan to implement
{Step k+1}
```

```
# Your task
- Implement the current plan with the given data.
- The implementation should be done based on the base code.
- The base code is an implementation of the previous plans.
- Your response should be a single markdown Python code (wrapped in ```).
- There should be no additional headings or text in your response.'''
```

Listing 51: Prompt used to generate a implementation of a plan after the initialization round.

For every round beyond the initial one, DS-STAR's coder agent $\mathcal{A}_{\texttt{coder}}$ converts the current plan step $p_{k+1}$ into a Python script using the above prompt. Unlike the initial round, the agent is provided with the intermediate solution code that implements all preceding step $p_0, \cdots, p_k$. This allows the agent to incrementally build upon the existing code, simplifying the implementation of each subsequent step.

### K.4 Verifier agent

```
f'''You are an expert data analysist.
Your task is to check whether the current plan and its code implementation is enough to answer the question.
# Plan
1. {Step 1}
...
k. {Step k}

# Code
```python
{code}
```

# Execution result of code
{result}

# Question
{question}

# Your task
- Verify whether the current plan and its code implementation is enough to answer the question.
- Your response should be one of 'Yes' or 'No'.
- If it is enough to answer the question, please answer 'Yes'.
- Otherwise, please answer 'No'.'''
```

Listing 52: Prompt used to verify the current plan.

We introduce a verifier agent $\mathcal{A}_{\texttt{verifier}}$. At any given round, this agent evaluates the state of the solution, *i.e.*, whether the plan in sufficient or insufficient to solve the problem. The evaluation is based on the cumulative plan, the user's query, the solution code, which is an implementation of the cumulative plan, and its execution result, using the above prompt.

### K.5 Router agent

```
f'''You are an expert data analysist.
Since current plan is insufficient to answer the question, your task is to decide how to refine the plan to
↪  answer the question.

# Question
{question}

# Given data: {filenames}
{filenames #1}
{summaries #1}
...
{filenames #N}
{summaries #N}

# Current plans
1. {Step 1}
```

```
...
k. {Step k}

# Obtained results from the current plans:
{result}

# Your task
- If you think one of the steps of current plans is wrong, answer among the following options: Step 1, Step
↪  2, ..., Step K.
- If you think we should perform new NEXT step, answer as 'Add Step'.
- Your response should only be Step 1 - Step K or Add Step.'''
```

Listing 53: Prompt used for the router agent which determines how to refine the plan.

If the verifier agent $\mathcal{A}_{\texttt{verifier}}$ determines that the current plan is insufficient to solve the user's query, DS-STAR must decide how to proceed. To this end, DS-STAR employs a router agent $\mathcal{A}_{\texttt{router}}$ which decides whether to append a new step or to correct an existing one, which leverages the above prompt.

## K.6   Finalyzer agent

```
f'''You are an expert data analysist.
You will answer factoid question by loading and referencing the files/documents listed below.
You also have a reference code.
Your task is to make solution code to print out the answer of the question following the given guideline.

# Given data: {filenames}
{filenames #1}
{summaries #1}
...
{filenames #N}
{summaries #N}

# Reference code
```python{code}
```

# Execution result of reference code
{result}

# Question
{question}

# Guidelines
{guidelines}

# Your task
- Modify the solution code to print out answer to follow the give guidelines.
- If the answer can be obtained from the execution result of the reference code, just generate a Python code
↪  that prints out the desired answer.
- The code should be a single-file Python program that is self-contained and can be executed as-is.
- Your response should only contain a single code block.
- Do not include dummy contents since we will debug if error occurs.
- Do not use try: and except: to prevent error. I will debug it later.
- All files/documents are in `data/` directory.'''
```

Listing 54: Prompt used to generate a final solution, which outputs the answer with the desired format.

To ensure properly formatted output, DS-STAR employs a finalyzer agent $\mathcal{A}_{\texttt{finalyzer}}$, which takes formatting guidelines, if exist, and generates the final solution code.

## K.7   Debugging agent

```
f'''# Error report
{bug}

# Your task
- Remove all unnecessary parts of the above error report.
```

```
- We are now running {filename}.py. Do not remove where the error occurred.'''
```

Listing 55: Prompt used to summarize the traceback of the error.

When debugging, we first summarize the obtained traceback of error using the above prompt, since the traceback may be too lengthy.

```
f'''# Code with an error:
```python
{code}
```

# Error:
{bug}

# Your task
- Please revise the code to fix the error.
- Provide the improved, self-contained Python script again.
- There should be no additional headings or text in your response.
- Do not include dummy contents since we will debug if error occurs.
- All files/documents are in `data/` directory.'''
```

Listing 56: Prompt used for debugging when analyzing data files.

When generating $\{d_i\}_{i=1}^N$ using $s_{\mathtt{desc}}$ obtained from $\mathcal{A}_{\mathtt{analyzer}}$, our debugging agent iteratively update the script using only the summarized traceback, using the above prompt.

```
f'''# Given data: {filenames}
{filenames #1}
{summaries #1}
...
{filenames #N}
{summaries #N}

# Code with an error:
```python
{code}
```

# Error:
{bug}

# Your task
- Please revise the code to fix the error.
- Provide the improved, self-contained Python script again.
- Note that you only have {filenames} available.
- There should be no additional headings or text in your response.
- Do not include dummy contents since we will debug if error occurs.
- All files/documents are in `data/` directory.'''
```

Listing 57: Prompt used for debugging when generating a solution code.

Once DS-STAR obtains $\{d_i\}_{i=1}^N$, our debugging agent utilizes such information with the above prompt when generating a solution Python script.

## L    Prompts for DS-STAR+

### L.1    Sub-question generator agent

```
f'''You are an expert data analysist.
Your task is to write a comprehensive data science report to the given question by using the files/documents
↪   listed below.
In order to do this, you have to first suggest multiple data analysis questions that should be answered to
↪   write the report.

# Given data: {filenames}
{filenames #1}
{summaries #1}
...
{filenames #N}
{summaries #N}

# Question
{question}

# Your task
- Suggest multiple factoid data analysis questions that are required to write the report really well.
- All the questions should be well-answered using the given data.
- All questions should be answered independently.
- Generate as much as you can.
- Return in valid JSON format:
Questions = {'question': str}
Return: list[Questions]'''
```

Listing 58: Prompt used for generating initial sub-questions.

Before we generate initial data science report $R$ for deep data research task, DS-STAR+ first generate initial sub-questions $\{f_i^0\}_{i=1}^{M_0}$ using the above prompt. Here, our sub-agent $\mathcal{A}_{\texttt{generator}}$ takes $\{d_i\}_{i=1}^N$ and the open-ended user query $q$ as inputs.

```
f'''You are an expert data analysist.
Your task is to complement the given data science report of the given question.
In order to do this, you have to suggest supplementary multiple data analysis questions that can strengthen
↪   to the report.

# Given data: {filenames}
{filenames #1}
{summaries #1}
...
{filenames #N}
{summaries #N}

# Given data science report:
{report}

# Question
{question}

# Your task
- Suggest multiple factoid data analysis questions that are required to complement the report.
- All questions should contain new information that is not included in the report.
- All the questions should be well-answered using the given data.
- All questions should be answered independently.
- Return in valid JSON format:
Questions = {'question': str}
Return: list[Questions]'''
```

Listing 59: Prompt used for generating sub-questions for report refinement.

To refine the generated data science report, DS-STAR+'s $\mathcal{A}_{\texttt{generator}}$ takes $R$ as an additional input, and the agents is guided to generate sub-questions that can complement the given data science report.

## L.2 Writer agent

```
f'''You are an expert data analysist.
Your task is to write a **comprehensive data science report** to the given question by using the data and
↪  some relevant informations listed below.

# Relevant informations:
{Sub-Question #1}
{Answer #1}
...
{Sub-Question #M_0}
{Answer #M_0}

# Question that you have to write a comprehensive data science report:
{question}

# Your task:
- The report should be grounded to the given relevant informations.
- For the citation, use the Sub-Question number as a citation number which is in 1 - {len(subquestions)}.
- The data science report should be relevant to given question, should be comprehensive, and should be
↪  insightful.
- The data science report should have nice structure, good readability, and should be professional.
- Write a very comprehensive data science report to the given above question.'''
```

Listing 60: Prompt used for generating initial report.

When generating an initial data science report, as shown above, DS-STAR+'s $\mathcal{A}_{\texttt{writer}}$ takes $\{f_i^0, a_i^0\}_{i=1}^{M_0}$ and the open-ended user query as inputs and is guided to generate a comprehensive data science report based on the given relevant information.

```
f'''You are an expert data analysist.
Your task is to complement the given data science report of the given question by using the some relevant
↪  informations listed below.

Relevant informations:
{Sub-Question #1}
{Answer #1}
...
{Sub-Question #M_k}
{Answer #M_k}

# Given data science report:
{report}

# Question that you have to write a comprehensive data science report:
{question}

# Your task:
- Do not modify the given report a lot. Just try to add new information.
- The report should be grounded to the given relevant informations.
- Cite with alphabet. For the citation, use the Sub-Question number as a citation alphabet (e.g., cite with
↪  [a] for the Sub-Question 1).
- The data science report should be relevant to given question, should be comprehensive, and should be
↪  insightful.
- The data science report should have nice structure, good readability, and should be professional.
- Complement the give data science report to the given above question.\n"'''
```

Listing 61: Prompt used for refining the report.

When refining the data science report, as shown above, DS-STAR+'s $\mathcal{A}_{\texttt{writer}}$ takes $\{f_i^k, a_i^k\}_{i=1}^{M_k}$, the open-ended user query $q$, and the original report $R$ as inputs and is guided to refine $R$ based on the given relevant information.

