# OpenReview forum: "DS-STAR: Data Science Agent for Solving Diverse Tasks across Heterogeneous Formats and Open-Ended Queries"
_TMLR — Decision pending for TMLR_

### Review · Reviewer_1sYm · 2026-04-28

**Summary Of Contributions:**

DS-STAR is a multi-agent LLM framework for automating end-to-end data science workflows. The paper presents two variants: DS-STAR (for well-defined queries with exact answers) and DS-STAR+ (for open-ended, exploratory research tasks that produce comprehensive reports).

## Strengths

* The authors provide a reasonably large selection of baselines.
* The selection of benchmarks in the paper is appropriate, although none of the three is well-established as of this writing.
* The ablations of the DS-Star method are thorough and convincing.
* Kudos to the authors for providing the prompts in Appendix K and L and the algorithms in Appendix C, which helps somewhat with reproducibility.
* Although appendix J is not particularly readable as an artifact, it is still useful in helping the reader to understand the system.

**Audience:**

Yes

**Audience Explanation:**

* The topic is highly relevant and would be of great interest to the TMLR community, were it successfully implemented.
* That said, the scientific value of the submission is shaky because of the aforementioned limitations in the empirical results, and the engineering relevance is limited because there is no artifact release, nor is there a release of an evaluation framework suitable for these benchmarks; for TMLR in particular, which emphasizes reproducibility, this is a serious flaw.

**Broader Impact Concerns:**

None.

**Claims And Evidence:**

No

**Claims Explanation:**

* Many of the baseline results in the main paper are sourced from leaderboards. While I acknowledge that it may not be possible to reproduce all of the results on those leaderboards, this fact makes it extremely difficult to assess what the actual contribution of DS-Star is, in isolation. The authors would like to argue that it is the design of their agent that drives the gains; but could it be better execution environments? A stronger LLM model as a backbone? A more appropriate set of abstractions that happen to fit the constraints of these benchmarks? The lack of independent reproductions of the baselines with proper experimental controls make it impossible to know for certain.
* There is no comparison to Claude Code, GitHub Copilot Workspace, Gemini's built-in code execution, Codex, Cursor, or any production-grade agentic coding tool. This is a very serious limitation in this paper, because the paper's claimed contributions are highly practical, and these are the actual tools most use in industry and most agentic benchmarks report. It makes little sense to have a class of "data scientist specialist" agents if a common coding agent can accomplish the same goals.
* The paper depends on Gemini-2.5-Pro's API, which is a moving target and itself already a generation behind Google's latest frontier release, which itself somewhat trails competitors. This issue harms credibility and reproducibility.
* The verifier agent appears superficially more novel in this work than it really is because of limited recognition of key prior work; Reflexion (Shinn et al. 2023), Self-Refine (Madaan et al. 2023) and TextGrad (Yuksekgonul et al, 2024) are important prior work and should be cited here.

**Requested Changes:**

At a minimum, I would request that the authors address the core weaknesses of the work outlined above.

---

> ### Author Response · Authors · 2026-05-22
> **Official Comment by Authors (1/2)**
>
> Dear Reviewer 1sYm,
>
> We sincerely appreciate the time and effort you dedicated to reviewing our manuscript and providing insightful comments. Below, we address each of your comments individually. The revised parts of the manuscript are marked in red.
>
> ---
> **[R1] Lack of experimental control for baselines.**
>
> We appreciate the reviewer’s focus on experimental rigor. To ensure a fair and controlled comparison between DS-STAR and existing baselines, we implemented the following measures.
>
> First of all, to isolate the impact of architectural design, we compared DS-STAR against the top-performing commercial baseline (Amity DA Agent), which also utilizes Gemini-2.5-Pro. This ensures that the performance gains are attributable to the DS-STAR framework rather than the underlying LLM.
>
> Secondly, to address concerns regarding the testing environment, we independently re-implemented the strongest open baseline (DA-Agent) within our own infrastructure using Gemini-2.5-Pro. As shown in Table 3, our reproduction of DA-Agent actually exceeded its original GPT-4 results.
>
> By evaluating both DS-STAR and the re-implemented DA-Agent under identical conditions, we carefully believe that we have effectively isolated the algorithmic contributions of our approach. We hope these controlled experiments provide the independent verification requested and demonstrate the effectiveness of DS-STAR’s abstractions for these benchmarks.
>
> ---
> **[R2] Missing baselines, specifically production-level agentic coding tools.**
>
> Thank you for this insightful and constructive feedback. We agree that production-grade agentic coding tools have become highly capable. To directly address your concern and validate the necessity of data science-specialized agents, we conducted an additional experiment by evaluating Claude Code and Gemini CLI on the DABStep benchmark.
>
> As shown in the below table, while Claude Code and Gemini CLI demonstrate impressive performance, even outperforming the strongest baseline (Amity DA Agent), it still falls short of our proposed framework. Notably, DS-STAR achieves higher accuracy despite using an older generation foundation model (Gemini-2.5-Pro) compared to Claude Code (Claude-4.5-Sonnet); note that ReAct framework with Gemini-2.5-Pro shows lower performance compared to ReAct with Claude-4-Sonnet (see Table 1 of our manuscript). Moreover, with the same base model (Gemini-2.5-Pro), we found that Gemini CLI is much worse than DS-STAR. This experiment demonstrates that while general-purpose coding agents are powerful, specialized data science agent architectures provide distinct, practical advantages for data science tasks, thereby validating our paper’s core contribution. We have incorporated these results in Table 1 and Section 4.1 of our revised manuscript.
>
> \begin{array}{lccc}
> \hline
> \text{Framework} & \text{Model} & \text{Easy} & \text{Hard}\newline\hline
> \text{Claude Code} & \text{Claude-4.5-Sonnet} & 84.72 & 41.80\newline
> \text{Gemini CLI} & \text{Gemini-2.5-Pro} & 76.39 & 27.78\newline
> \text{Gemini CLI} & \text{Gemini-3-Flash} & \textbf{88.89} & 42.33\newline
> \textbf{DS-STAR (Ours)} & \text{Gemini-2.5-Pro} & \textbf{88.89} & \textbf{45.50}\newline\hline
> \end{array}
>
> ---
> **[R3] The paper depends on Gemini-2.5-Pro.**
>
> Thanks for the insightful question. We understand the concern regarding the evolving nature of API-based models. To alleviate your concern, we additionally compare Gemini-2.5-Flash, Gemini-2.5-Pro, and Gemini-3.1-Pro-Preivew. As shown in the below table, as the underlying model’s reasoning capabilities increase, the effectiveness of our framework improves proportionally. This empirical evidence suggests that our method is model-agnostic and will likely yield even better results on upcoming frontier models. We have incorporated such results in our revised manuscript (see Table 1).
>
> \begin{array}{lccc}
> \hline
> \text{Framework} & \text{Model} & \text{Easy} & \text{Hard}\newline\hline
> \textbf{DS-STAR (Ours)} & \text{Gemini-2.5-Flash} & 80.56 & 29.37\newline
> \textbf{DS-STAR (Ours)} & \text{Gemini-2.5-Pro} & \textbf{88.89} & 45.50\newline
> \textbf{DS-STAR (Ours)} & \text{Gemini-3.1-Pro-Preview} & 87.50 & \textbf{54.23}\newline\hline
> \end{array}

---

> ### Author Response · Authors · 2026-05-22
> **Official Comment by Authors (2/2)**
>
> **[R4] Insufficient literature review on verifiers.**
>
> We thank the reviewer for the insightful suggestion. We agree that acknowledging these key works is crucial for properly positioning our verifier agent. In the revised manuscript, we have added a dedicated subsection to the Related Works (see Section 2) to discuss how our approach relates to and differs from these methods. Specifically, we highlight how our verifier mitigates reasoning problems in long-horizon data science tasks through iterative self-correction and plan sufficiency checks.
>
> **Updated Related Works Content:** Our work builds upon the literature on iterative refinement and self-correction. Reflexion [1] introduced a framework for agents to learn from linguistic feedback through trial and error, while Self-Refine [2] demonstrated that LLMs can improve their own outputs by generating and acting upon multi-step feedback. More recently, TextGrad [3] proposed treating textual feedback as gradients to optimize system components. Our verifier agent differentiates itself by specifically evaluating the sufficiency of a dynamically evolving analysis plan grounded in intermediate code execution results across heterogeneous data formats, enabling the agent to navigate complex, open-ended data science tasks where ground-truth labels are absent.
>
> [1] Shinn et al., Reflexion: Language Agents with Verbal Reinforcement Learning, 2023.\
> [2] Madaan et al., Self-Refine: Iterative Refinement with Self-Feedback, 2023.\
> [3] Yuksekgonul et al., TextGrad: Automatic “Differentiation” via Text, 2024.
>
> ---
> **[R5] Lack of artifacts.**
>
> First, we would like to clarify that all the evaluations were conducted strictly following the original benchmark protocols. Specifically, the evaluation protocols for KramaBench, DA-Code, and DABStep-Research are publicly available. For the DABStep benchmark, our submission has been officially validated by the Hugging Face leaderboard. Furthermore, we commit to open-sourcing the DS-STAR project upon acceptance.

---

### Review · Reviewer_ivHv · 2026-05-08

**Summary Of Contributions:**

This paper argues that real data analysis involves heterogeneous data formats and increasingly open queries two challenges, and the prior agents either assume structured data format or only handle specific, well-defined queries. To address the gap, this paper proposes a two-tier system DS-STA (with a analyzer→planer→coder→verifier→router loop) for QA, and DS-STAR+ on top of it (sub-question generator →DS-STAR per sub-question→ write with citations → optional refinement). Empirical comparison studies on DABStep, KramaBench, DA-Code are reported.

**Audience:**

Yes

**Audience Explanation:**

The submission addresses a problem that is of interest to TMLR's audience working on LLM agents and applied data science, and at least one component is novel and practically valuable enough to stand on its own.

**Claims And Evidence:**

No

**Claims Explanation:**

1. The paper's own appendix ablation contradicts its central methodological claim about the verifier.

Section 3.1 explicitly argues: "By conditioning the judgement on $s_k$ and its execution output $r_k$, $\mathcal{A}_\text{verifier}$ can provide more grounded feedback since it assesses whether $s_k$ is well-implemented following the plan, and whether $r_k$contains the information needed to fully address the query." This grounded-feedback property is positioned as a key feature distinguishing DS-STAR from execution success only verification.

Appendix Table 8 directly tests this claim: removing the code and its execution result from the verifier's input drops hard-level accuracy only from 45.24 → 43.92, a 1.3-point gap. By the paper's own measurement, grounding on execution contributes marginally; the verifier is judging plan sufficiency almost entirely from {plan, query} alone. Thus the paper overstates what the data show.

in addition, the verifier's reliability is never directly evaluated. Namely there is no precision/recall analysis of its sufficient/insufficient classifications against ground truth, despite the verifier being the framework's central feedback signal and termination criterion. Therefore it is difficult to tell whether the verifier is approximating correctness, rewarding code that runs cleanly regardless of conceptual validity, or doing something else entirely.

2. DS-STAR+'s contribution is generating analyst-grade research reports for human consumption, yet the supporting evidence is a pairwise win-rate against a single baseline (DeepAnalyze-8B) judged by Gemini-3-Pro and GPT-5. Also, the generator (Gemini-2.5-Pro/Flash) and one of the two judges (Gemini-3-Pro) come from the same model family, exposing the comparison to documented self-preference bias. Also, for a contribution whose output is consumed by humans, the absence of even a small human pairwise study is a clear soundness gap.

**Requested Changes:**

1. To strengthen the claim in Section 3.1, either (a)  add a direct evaluation of the verifier as a classifier: on tasks with known ground truth, report precision/recall of the sufficient/insufficient judgments. If the verifier reliably separates correct from incorrect plans, the grounded-feedback claim is supported, or (b) revise Section 3.1 to match Table 8: clarify that grounding on execution contributes marginally and that the verifier's main value comes from step-by-step (vs. full-plan-at-once) judgment, regardless of execution context.

2. To strengthen the DS-STAR+ evaluation, add a small human pairwise study, where the central artifact is a research report consumed by humans, also add a third-family judge to mitigate same-family self-preference bias.

---

> ### Author Response · Authors · 2026-05-22
>
> Dear Reviewer ivHv,
>
> We sincerely appreciate the time and effort you dedicated to reviewing our manuscript and providing insightful comments. Below, we address each of your comments individually. The revised parts of the manuscript are marked in red.
>
> ---
> **[R1] Contradiction in methodological claims.**
>
> We thank the reviewer for the insightful analysis of our ablation study. We appreciate the opportunity to clarify the primary motivation behind the DS-STAR’s verifier. While we initially emphasized “grounded feedback” in Section 3.1, the fundamental value of our verifier agent lies in its ability to explicitly evaluate plan sufficiency at a granular, step-by-step level.
>
> Here, the marginal drop when removing $s_k$ and $r_k$ indicates that the verifier’s strength is derived primarily from its role in decomposing the query-plan relationship, rather than just observing execution traces. However, we would also like to mention that even though the gain is not that large, there are still consistent improvements with these two additional contexts for both easy and hard-level tasks. This justifies their inclusion as a design choice for maximizing performance.
>
> Following the reviewer’s suggestion, we have revised Section 3.1 to better align the empirical results in Table 8. We have toned down the claim regarding the execution-based grounding and instead position $s_k$ and $r_k$ as supplementary context that aids the verifier. The revised text has emphasized that the verifier’s primary utility is providing a step-wise assessment of plan sufficiency, which remains a novelty from execution-only verification.
>
> ---
> **[R2] Lack of verifier performance metrics.**
>
> We appreciate this insightful question. First of all, the verifier’s reliability is implicitly validated through the final task accuracy. As shown in Figure 2(b), the positive correlation between the maximum number of refinements and overall performance suggests that the verifier effectively identifies deficiencies, enabling the model to generate sufficient plans. Secondly, we have addressed the verifier’s reliability through sensitivity analyses in Table 5 and Appendix A. These results demonstrate that the verifier’s performance remains consistent across various configurations.
>
> To further alleviate your concern regarding precision and recall, we conducted an additional experiment on the DABStep development set using DS-STAR with a maximum of two refinement rounds. Specifically, we evaluated the verifier’s classifications against the ground truth outcomes. We define the verifier as “correct” if it labels a plan as “Sufficient” and the task is subsequently solved correctly, or if it labels a plan as “Insufficient” and the task fails. We have included this precision/recall analysis (see the below table) in the revised manuscript (see Section 4.2).
>
> \begin{array}{lccc}
> \hline
> & \text{Accuracy} & \text{Precision} & \text{Recall}\newline\hline
> \text{DS-STAR’s verifier} & 70.00 & 63.89 & 68.33\newline\hline
> \end{array}
>
> ---
> **[R3] Evaluation methodology for open-ended tasks.**
>
> First of all, we would like to clarify that we have already measured the win rate of DS-STAR+ against DeepAnalyze-8B using a third-family judge, specifically GPT-5 (see lower part of Figure 4).
>
> In addition, we would like to clarify that the DABStep-Research benchmark utilizes a specific, human-generated checklist for evaluation. By using these human-authored rubrics to guide the LLM-based scoring and preference judging (see Figure 4), we ensure that the automated assessment remains grounded in human standards of quality and accuracy. You can find the  full details and rubrics for this benchmark at https://github.com/ruc-datalab/DeepAnalyze/blob/main/playground/DABStep-Research/gpt_eval.py.
>
> To further alleviate your concern, we conducted an experiment regarding human preference. Specifically, we subsampled 10 tasks (2 from each of the 5 categories) and asked 8 evaluators with a blind preference test (totaling 80 pair-wise comparisons). The results show that the reports generated by DS-STAR were preferred over 91.2% (73/80) of cases. These results suggest that DS-STAR’s performance gains are consistent across both LLM-based metrics and human judgement. We have incorporated this result in Section 5.2 of the revised manuscript.

---

### Review · Reviewer_83KA · 2026-05-09

**Summary Of Contributions:**

This paper introduces DS-STAR, a collection of LLM-based agents designed to automatically answer data science questions from collections of heterogeneous data files. At its core, DS-STAR comprises specialized agents that iteratively (1) plan what analysis needs to be done, (2) write code to perform the analysis (3) verify whether the analysis answers the question and (4) decide whether to modify or extend the existing analysis if it does not yet answer the question. For open-ended questions, the paper introduces DS-STAR+, which breaks open-ended questions into a set of well-defined queries which can be answered by DS-STAR before being recombined into a complete answer.

DS-STAR is evaluated on a range of existing data science LLM benchmarks, and is compared to a variety of baselines. Across these benchmarks, it outperforms these baselines, sometimes by considerable margin.

**Audience:**

Yes

**Audience Explanation:**

There is no discernible ML contribution in this paper; It strings together existing LLMs with novel prompts. It should nevertheless be interesting to data science practitioners within the TMLR audience.

**Broader Impact Concerns:**

Broader impact concerns are not addressed, but appear in line with established concerns around large language models.

**Claims And Evidence:**

Yes

**Claims Explanation:**

DS-STAR is evaluated on a wide variety of benchmarks, and is compared to a selection of recent alternatives. These results are accompanied by extensive ablation studies showing that all proposed components of DS-STAR contribute to its performance, and a cost analysis (both in terms of tokens and time) comparison between DS-STAR and the baselines.

**Requested Changes:**

Generally, I have no concerns that I would call critical. I do however, think that several things could make the paper stronger:
- No variance or standard deviation is reported on any of the results in the paper. Furthermore, no statistical testing is done. I understand that evaluating methods like DS-STAR is already prohibitively expensive, but not even the cost analysis measures the variance in the cost.
- For open-ended questions, DS-STAR+ was compared to DABStep-Research by asking another LLM which output it thought was best. Though I can see why this would be a good starting point for comparison, I do not think this is a complete substitute for human evaluation of these outputs. Humans have very different cognitive capacities and biases from LLMs and might therefore respond differently on these kinds of comparisons.

---

> ### Author Response · Authors · 2026-05-22
>
> Dear Reviewer 83KA,
>
> We sincerely appreciate the time and effort you dedicated to reviewing our manuscript and providing insightful comments. Below, we address each of your comments individually. The revised parts of the manuscript are marked in red.
>
> ---
> **[R1] DS-STAR strings together existing LLMs with novel prompts. But, no discernible ML contribution.**
>
> We appreciate the reviewer’s observation that DS-STAR features novel prompts and would be of interest to data science practitioners within the TMLR audience. We believe the paper’s primary contribution lies in the effective orchestration of these components to solve the critical, yet underexplored, challenge of handling heterogeneous and unstructured data formats (beyond standard CSVs) to automate data science tasks.
>
> In addition, we would like to also emphasize that the novelty of DS-STAR is evidenced by its unique architectural components and their empirical impact (see Table 4). Specifically, our data file analyzer is a novel preprocessing step that identifies file structures; removing it causes hard-level accuracy to drop significantly from 45.24% to 26.98%. Furthermore, our verifier agent provides a novel mechanism for evaluating plan sufficiency, while the router agent introduces a non-trivial decision-making step that allows the system to backtrack and fix errors.
>
> ---
> **[R2] Lack of statistical rigor.**
>
> We thank the reviewer for highlighting the importance of statistical rigor. We agree that reporting variance and conducting statistical tests provide a more complete picture of a method’s robustness. However, we also clarify that our evaluation strictly follows the established protocols of the benchmarks. To ensure a direct and fair comparison with existing baselines, we reported scores in the same format as the published literature, which typically omits standard deviation due to the high evaluation costs.
>
> To further alleviate your concern, we conduct an additional experiment to measure the standard deviation of accuracy on the DABStep benchmark. We report mean and standard deviation (in parentheses) in the below table. We have also included the result in Table 1, 4, 5, 6 of the revised manuscript.
>
> \begin{array}{lccc}
> \hline
> \text{Framework} & \text{Model} & \text{Easy} & \text{Hard}\newline\hline
> \text{DS-STAR (Ours)} & \text{Gemini-2.5-Pro} & 88.89 (1.39) & 45.50 (1.22)\newline\hline
> \end{array}
>
> Moreover, we also measure the mean and standard deviation (in parentheses) in computational complexity on the development set of DABStep benchmark (see the table below). We have incorporated these variance measures in Table 7 of the revised manuscript.
>
> \begin{array}{lccccc}
> \hline
> \text{Method} & \text{\\# LLM calls} & \text{\\# Input tokens} & \text{\\# Output tokens} & \text{Cost (\\$)} & \text{Time (sec.)}\newline\hline
> \text{DS-STAR (Ours)} & 9.6 (2.8) & 110177 (43029) & 2301 (2049) & 0.16 (0.07) & 197.12 (111.24)\newline\hline
> \end{array}
>
> ---
> **[R3] Evaluation methodology for open-ended tasks.**
>
> We would like to clarify that the DABStep-Research benchmark utilizes a specific, human-generated checklist for evaluation. By using these human-authored rubrics to guide the LLM-based scoring and preference judging (see Figure 4), we ensure that the automated assessment remains grounded in human standards of quality and accuracy. You can find the  full details and rubrics for this benchmark at https://github.com/ruc-datalab/DeepAnalyze/blob/main/playground/DABStep-Research/gpt_eval.py.
>
> However, to further alleviate your concern, we conducted an experiment regarding human preference. Specifically, we subsampled 10 tasks (2 from each of the 5 categories) and asked 8 evaluators with a blind preference test (totaling 80 pair-wise comparisons). The results show that the reports generated by DS-STAR were preferred over 91.2% (73/80) of cases. These results suggest that DS-STAR’s performance gains are consistent across both LLM-based metrics and human judgement. We have incorporated this result in Section 5.2 of the revised manuscript.

---

### Decision · Action_Editor_LoWG · 2026-06-23

**Recommendation:** Accept with minor revision

**Additional Comments:**

The authors addressed the main concerns of the three reviewers adequately and conducted extra experiments which strengthen the contribution. I recommend to add these results to the new version of the paper ( these can be added to appendix if there is not enough space but refer to them in the main if possible), specifically:
- Add standard deviations to results tables.
  - Include the human preference study results.
  - Include controlled comparisons: re-implementation of DA-Agent in their environment and add Claude Code and Gemini CLI baselines on DABStep.
  - Add results with other Gemini models, such as DS-STAR with Gemini-3.1-Pro-Preview.
  - As mentioned by reviewer ivHv, the revised verifier claims in Section 3.1 should match the ablation evidence and soften the initial claims about the verifier.
  - Add missing related work citations as mentioned by reviewer 1sYm.
  - Open-sourcing code upon acceptance if possible.
For this reason, I am recommending acceptance with minor revisions.

**Audience:**

Yes

**Audience Explanation:**

The contribution is of interest to those working on LLM agents and applied data science within the TMLR audience.

**Claims And Evidence:**

Yes

**Claims Explanation:**

The authors took significant steps during the revision to strengthen the empirical evidence in order to support their method's claims: they (1) demonstrated the generality of the framework by evaluating DS-STAR across multiple model versions (Gemini-2.5-Flash, Gemini-2.5-Pro, and Gemini-3.1-Pro-Preview), showing consistent improvements as the underlying model's capabilities increase; (2) conducted controlled comparisons against production-grade agentic tools (Claude Code and Gemini
  CLI) , showing that DS-STAR outperforms general-purpose coding agents on hard-level tasks even when using non-SotA LLMs; (3) added standard deviations and cost analysis tables to address the lack of statistical rigor concern; (4) provided a human preference study that confirming that DS-STAR+ reports are preferred in 91.2% of cases; and (5) revised the verifier claims in Section 3.1 to better align with the ablation evidence. These additions adequately address the main concerns raised by all reviewers.

---

> ### Author Response · Authors · 2026-07-20
>
> Dear Action Editor LoWG,
>
> We sincerely appreciate the time and effort you dedicated to reviewing our manuscript and providing insightful comments. Following your recommendations, we have revised our manuscript as:
>
> - Added standard deviations: Table 1, Table 4, Table 5, Table 7.
> - Included the human preference study results: Section 5.1.
> - Included additional baselines (e.g., Claude Code) & using other Gemini models (e.g., Gemini-3.1-Pro-Preview): Table 1.
> - Revised Section 3.1 to match the ablation evidence and to soften the claims about the verifier.
> - Revised Section 2 to add missing related work citations.
> - Open-sourced the code at https://github.com/google-research/ds-star.
>
> Thanks for the helpful recommendations!
>
> Sincerely,
>
> Authors